# Crystal and melt inclusion timescales reveal the evolution of magma migration before eruption

Dawn C.S. Ruth [1,2], Fidel Costa [2,3], Caroline Bouvet de Maisonneuve [2,3], Luis Franco [4], Joaquin A. Cortés [5,6] & Eliza S. Calder[6]

Volatile element concentrations measured in melt inclusions are a key tool used to understand magma migration and degassing, although their original values may be affected by different re-equilibration processes. Additionally, the inclusion-bearing crystals can have a wide range of origins and ages, further complicating the interpretation of magmatic processes. To clarify some of these issues, here we combined olivine diffusion chronometry and melt inclusion data from the 2008 eruption of Llaima volcano (Chile). We found that magma intrusion occurred about 4 years before the eruption at a minimum depth of approximately 8 km. Magma migration and reaction became shallower with time, and about 6 months before the eruption magma reached 3–4 km depth. This can be linked to reported seismicity and ash emissions. Although some ambiguities of interpretation still remain, crystal zoning and melt inclusion studies allow a more complete understanding of magma ascent, degassing, and volcano monitoring data.

[1] Department of Geology, State University of New York at Buffalo, 126 Cooke Hall, Buffalo, NY 14260, USA. [2] Earth Observatory of Singapore, Nanyang Technological University 50 Nanyang Avenue, Singapore 639798, Singapore. [3] Asian School of the Environment, Nanyang Technological University, 50 Nanyang Avenue, Singapore 639798, Singapore. [4] Observatorio Volcanológico de los Andes del Sur, Servicio Nacional de Geología y Minería, Rudecindo Ortega #03850, Temuco, Chile. [5] Department of Geography, Edge Hill University, Ormskirk L39 4QP, UK. [6] School of GeoSciences, University of Edinburgh, The King's Building, James Hutton Road, Edinburgh EH9 3FE, Scotland. Correspondence and requests for materials should be addressed to D.C.S.R. (email: sdawn@ntu.edu.sg)

Magmatic volatiles are the fuel that drives volcanic eruptions. How volatile elements (e.g., C, H, S) exsolve from the magma critically affects eruption intensity. The transition between explosive and effusive eruptions depends on whether magma behaves as a closed or an open system during ascent. In closed system behaviour, volatiles fail to escape, causing magma expansion and leading to fragmentation[1,2], while volatiles in an open system can escape and ascend separately from the magma[3,4]. Measuring the volatile content and composition in melt inclusions trapped in phenocrysts is one of the key techniques used to infer magma storage depth and degassing modes[5–7]. However, recent studies have identified several processes that likely modify the volatile contents of melt inclusions including: dehydration and rehydration due to $H^+$ diffusion through the host crystal[8–12], $CO_2$ partitioning into vapour bubbles[13–15], and $CO_2$ loss via melt inclusion rupture[16]. In addition, in the last two decades, chemical zoning studies of minerals have shown that a single batch of magma can carry a large variety of crystals of different origins and ages[17,18]. In particular, studying the chemical gradients of crystals and modelling them with diffusion laws has provided evidence for crystals being sourced from different parts of the plumbing system with timescales ranging from a few hours to years within a single eruption deposit[19–26]. Thus, it is currently not straightforward how melt inclusion data can be related to the processes of magma migration and evolution that may precede the eruption.

We focus on the zoning patterns and melt inclusions from olivine crystals sourced from the 2008 eruption of Llaima volcano (Southern Chile). Melt inclusions from this and other eruptions at Llaima volcano were the subject of three previous studies focusing on magma chamber and conduit processes related to violent Strombolian activity[25,27,28]. These investigations reveal that despite consistently narrow bulk rock compositions (52–54 wt% $SiO_2$), the melt inclusions indicate the presence of a wider compositional range of magmas (48–56 wt% $SiO_2$) within the plumbing system. This plumbing system likely consists of a plexus of vertically oriented crystal-rich bodies that extend from approximately 1–14 km beneath the summit. A more developed zone of magma storage is present at around 2–4 km according to the apparent melt inclusion pressures[27,28]. Prolonged magma injections and mixing with the shallow resident melt (some months up to a few years) prior to eruption has been proposed based on crystal zoning studies of the 1957 and older eruptions[25]. Additionally, the classic open system degassing trends observed in the melt inclusion dataset from the 2008 eruption are likely a record of the background persistent degassing rather than a record of magmatic processes specifically related to the 2008 eruption[28].

Here we use the chemical diffusion laws in the host crystals as a proxy to estimate the minimum time between the melt inclusions' last re-equilibration depth and the eruption. This provides a first-order approximation of the progressive ascent, mixing, and overall shallow accumulation of magma prior to eruption.

## Results

**Crystal classification and chemical zoning.** Based on crystal morphology, melt inclusion abundance, and melt inclusion morphology (Fig. 1 with additional images in Supplementary Fig. 1 and ref. [29]; classification schema in Supplementary Data 1) we identified three groups of crystals. Group A are subhedral to anhedral with one to three centrally located melt inclusions. Group B crystals are subhedral to anhedral and have multiple melt inclusions, including large ones in the crystal centre and abundant smaller, round melt inclusions near the rims. These textures suggest a more complex history that includes dynamic,

disequilibrium conditions, including dissolution. The multiple small melt inclusions might reflect that they were once melt channels formed during dissolution, which then closed during regrowth. Group C are subhedral to euhedral crystals, sometimes exhibiting dendritic growth features (see br1-3 in Supplementary Fig. 1[29]). Melt inclusions in this category are round to elongate and are generally aligned with the host crystal faces.

Fe–Mg X-ray maps of these crystals show broad concentric chemical zoning (Figs. 1 and 2a, complete element profiles provided in Supplementary Data 2 and in ref. [30]). We also observe a very thin (<3 μm) Fe-rich rim in 13 of 22 crystals. These rims are potentially growth-induced and likely related to final quenching of the magma, which we will not describe further. Olivine crystals have reverse, normal and complex Fe–Mg zoning, but most (15 out of 22 crystals), show increasing Mg towards the rims (i.e., reverse %Fo zoning, where %Fo (mol%) = 100 × Mg/[Fe + Mg]; Fig. 2a). Two crystals exhibit normal zoning, and five crystals show complex zoning with one alternation (hook or trough) between normal and reverse zoning (see br1-4 in Fig. 1). Olivine core compositions cluster into two groups, one of %$Fo_{70–76}$ and another of %$Fo_{80–82}$ (Fig. 2a, b); there is no observed relationship between zoning and core composition. Rim compositions are also bimodal, but over a narrower range between %$Fo_{77–79}$ and %$Fo_{80–83}$ (Fig. 2a, c). Hook compositions from the complexly zoned crystals are %$Fo_{84–85}$ (Fig. 2a, d). Llaima has erupted remarkably consistent lava compositions over multiple historic eruptions, with olivine crystals exhibiting similar compositions and zoning patterns[25,27] to the ones we report here, and thus, we posit that the crystals and melt inclusions in this study are representative of Llaima products in general.

Phosphorous (P) X-ray maps (Fig. 1) show that elevated P regions do not necessarily correlate with the shift of %Fo content. This is expected because any coherence between Fe–Mg and P zoning would be removed via Fe–Mg diffusion. Reversely zoned crystals have the most abundant P bands, especially near the crystal rims. Near-rim P bands were not observed for the normally zoned crystals. Normal and reverse zoned crystals show elevated P concentrations in the core. Some melt inclusions are bounded by high relative P concentration bands; these are often parallel to crystal faces, especially near the crystal rims (for example see br5–1, Fig. 1). These data suggest that olivine crystals had an early episode of fast growth, as recorded in the high-P core zones, and then subsequent melt inclusion entrapment[31,32]. At some point, intrusion of more primitive magma occurred into a crystal-rich mush that contained the more evolved crystals. This led to partial dissolution and diffusion of various elements (Fe–Mg, Mn, etc.) between the crystals and the intruding melts. Such series of events are also observed by the combined X-Ray map and trace element study of olivine crystals from older Llaima eruptions[25]. Normal and complexly zoned crystals have higher %Fo content in the core than reversely zoned crystals (Fig. 2a), consistent with the former originating from a primitive recharge magma, and the latter from the evolved resident magma. The wide range of rim compositions (%$Fo_{77–79}$, %$Fo_{80–83}$) probably reflects varying proportions of mixing between the incoming and resident melt, and potentially incomplete melt homogenization owing to mixing taking place in a crystal-rich mush. Similar variability in olivine and melt inclusion textures have been observed in scoria from other mafic volcanic systems and were attributed to similar processes, like dissolution (e.g., our group B) and rapid crystallization (our group C) (for example refs. [33–35]).

**Diffusion timescales from olivine zoning.** We modelled the %Fo concentrations of 22 profiles in olivine and obtained 27 timescales. We interpret the hook and trough features in the complex

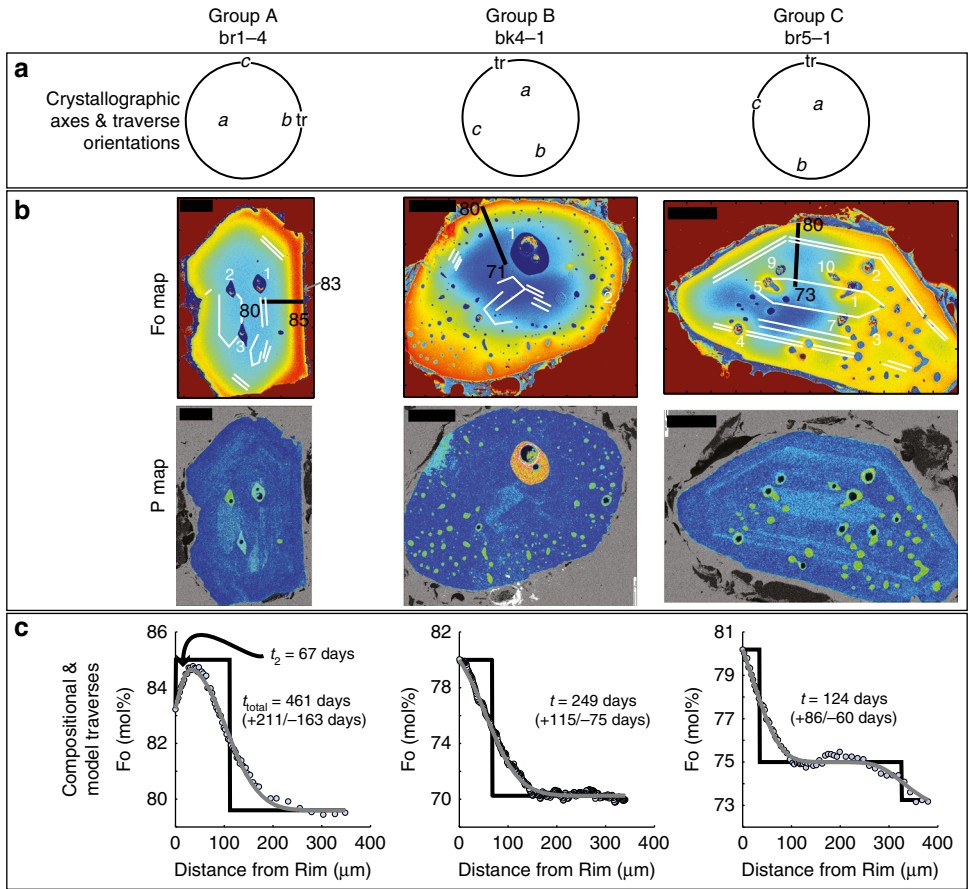

**Fig. 1** Key datasets for representative crystals from each crystal group. **a** EBSD results showing the crystallographic and traverse orientation projected on the lower hemisphere (a, b, c, crystallographic axes, tr = traverse). **b** X-ray maps of %Fo (numbers in the map; red colour is higher %Fo than blue) and phosphorous abundance (brighter zones are higher concentrations). In the %Fo maps black lines indicate the traverse location, and white lines indicate regions of elevated phosphorous; these crystals were imbedded in indium. The outlined numbers identify specific melt inclusions. Some melt inclusions are open to the surrounding matrix melt (e.g., crystal br 5-1) as evidenced by the host olivine around some melt inclusions that shows patchy zoning. We refer to these as open melt inclusions. **c** Electron microprobe and calibrated BSE traverses of Fo (light dots), initial conditions used for the diffusion model (black), and models profiles (grey). We note these crystals are tilted, but profile locations were placed perpendicular to the tilt direction to minimize this effect. The times are shown in each crystal

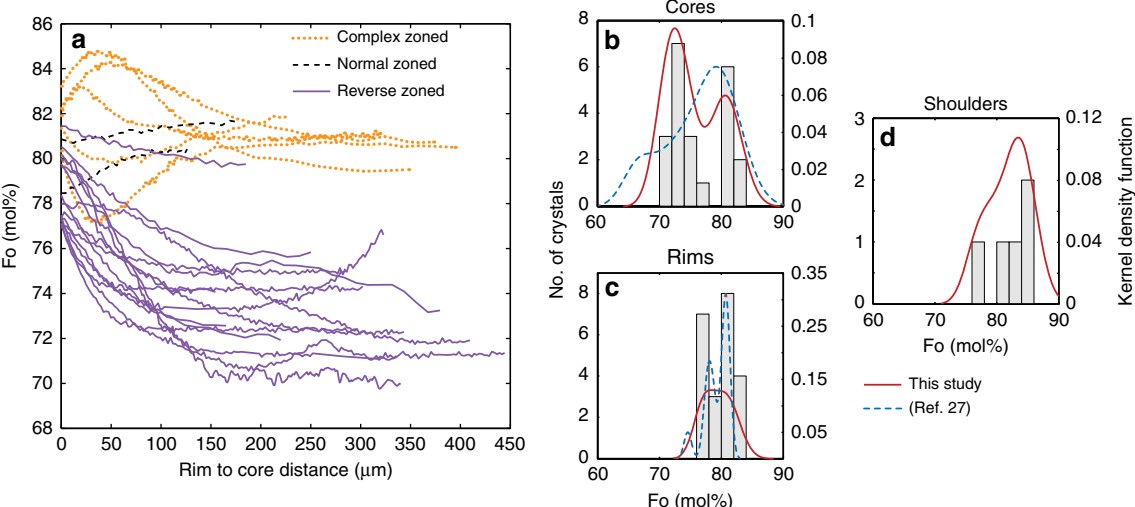

**Fig. 2** Summary of olivine compositions from 2008 Llaima scoria deposits. **a** Rim to core electron microprobe %Fo traverses, that show mainly reverse zoning and the presence of "shoulders" in some crystals. **b-d** histograms of compositions of cores (**b**), rims (**c**), shoulders (**d**). The black line is a kernel density function, the dashed line are the data from older Llaima eruptions[25,27]. See text for more discussion

zoned crystals as a recording a second mixing event. As such, we modelled these as two different timescales (e.g., br1-4 in Fig. 1). For many profiles we were also able to model the concentrations of Ca, Mn, and Ni, whose timescales generally coincide with the %Fo timescales. The diffusion model parameters and results are provided in Supplementary Data 3.

For normal and reverse zoned crystals we found timescales ranging from 37 to 1375 days (3.8 years; Figs. 3, 4), although 70% of timescales derived from %Fo zoning are <9 months. However, the timescale data do not show any dominant peak. This suggests that magma recharge and subsequent mixing was a progressive process rather than a punctuated, single, replenishment event (Fig. 4). Although we acknowledge that our dataset is small, the dominant population, with cores from %Fo$_{70-76}$, produce timescales that span the entire timescale range (see Supplementary Data 3). Moreover, the distribution and range of timescales in our dataset is similar to larger timescale datasets consisting of multiple populations, from the 1783–1784 eruption of Laki[36] and from multiple eruptions at Etna[23]. That the distributions and timescale ranges are consistent suggests that smaller datasets such as ours may still provide first-order information about the timing of magma movement within the plumbing system.

The complex crystals (e.g., br1-4 in Fig. 1, Supplementary Data 1, samples labelled "tt") have timescales for the interior zone that range from 128 to 461 days (see Fig. 1, br1-4), and rim timescales that are <90 days. These indicate that some olivine crystals likely record multiple mixing events where the interior zone records an earlier event and the rim timescale records a late stage magma mixing event before eruption. Most of these time determinations are longer than syn-eruptive magma ascent, degassing, and crystallization[37]. Moreover, since the crystals host the melt inclusions, these timescales represent the minimum melt inclusion residence times. We recognize that the absolute ages of the melt inclusions are unknown.

**Volatile contents and re-equilibration pressures**. Melt inclusion volatile contents can be used to obtain the pressures of magma residence assuming volatile saturation and a solubility model (for example refs. [38,39]). The melt inclusions in this study have $H_2O$ contents that range from 0.5 to 3.0 wt%, and $CO_2$ from 30 to 850 ppm (Fig. 5; data compiled in Supplementary Data 3) (27–28). Group A crystals exhibit the largest range of $H_2O$ (1.2–3.0 wt%) and $CO_2$ (26–796 ppm). These crystals host the highest $H_2O$ and $CO_2$ melt inclusions in the dataset. Calculated volatile saturation pressures range from 41 to 151 MPa, corresponding to 3.5 to 8.6 km beneath the summit. Group B melt inclusion $H_2O$ and $CO_2$ contents tend to be lower, 0.7–2.1 wt%, and 32–92 ppm respectively, which correspond to 1.8–5.0 km beneath the summit (13–73 MPa). Group C crystals have the most melt inclusions, some which may have been open to the matrix. These Group C melt inclusions exhibit the lowest $H_2O$ and $CO_2$ contents measured (0.5–2.6 wt% $H_2O$, and 12–127 ppm $CO_2$) and thus lowest volatile saturation pressures (8–68 MPa), and shallowest depths (1.1–4.7 km).

**Assessing post-entrapment modifications**. Post-entrapment modification of melt inclusions may alter their volatile contents

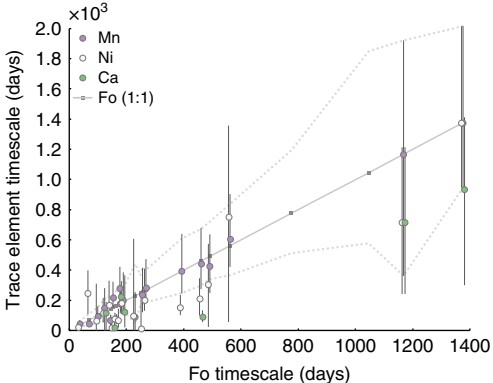

**Fig. 3** Timescales results for multiple elements (%Fo, Mn, Ni, and Ca). Note, the %Fo values for the Ni and Ca datapoints are shifted by −5 and +5 days, respectively, for visualization purposes. The timescales obtained for %Fo range from a couple of weeks to 4 years. The uncertainty for the % Fo timescales is given with the dotted lines. Timescales obtained from the minor elements (Mn, Ni, and Ca) for the same traverse overlap within error, and suggest that the chemical zoning is mainly controlled by diffusion and not growth[61]. Error bars shown obtained from DIPRA[35] and correspond to the analytical uncertainty and a temperature uncertainty of 30 °C

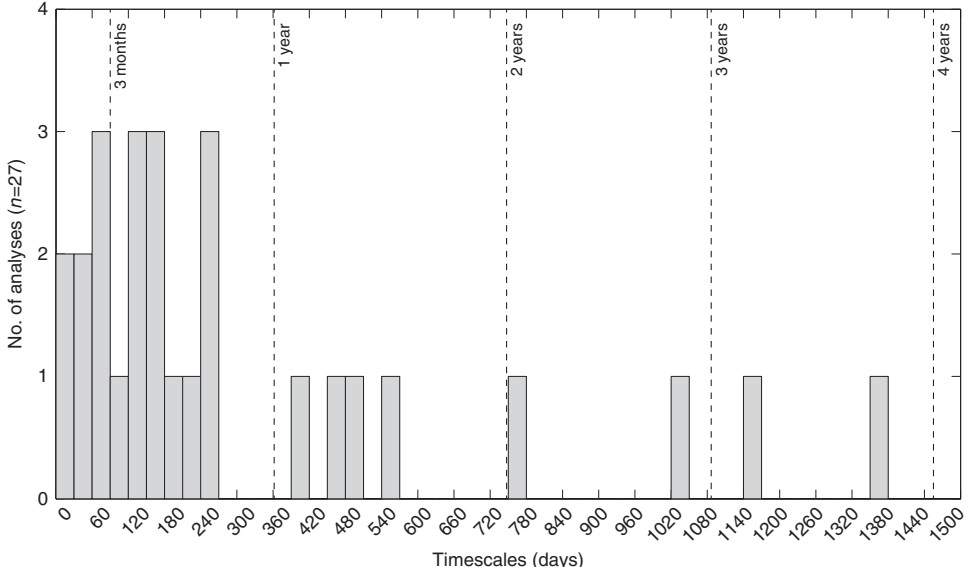

**Fig. 4** Histogram showing the distribution of timescales obtained from Fo. Note that about 70% of times are less than 9 months, and that there is no discernible mode, implying progressive magma additions rather than a single large replenishment

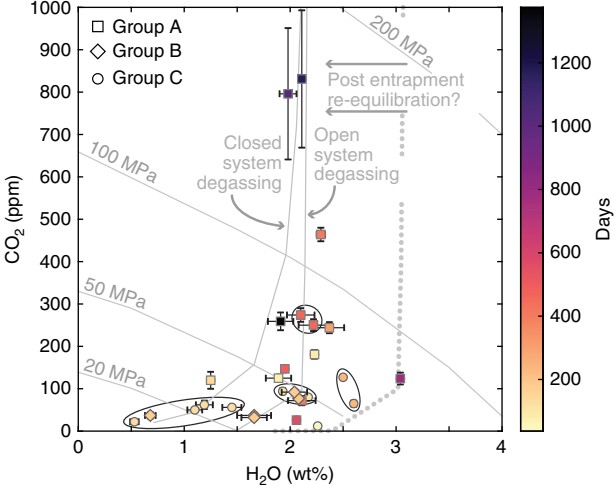

**Fig. 5** Volatile contents vs. timescales. $H_2O$ and $CO_2$ contents plotted with isobars and calculated diffusion timescales (shown in colour bar). Melt inclusions from the same crystals have been grouped by a circle. Melt inclusion pressures correlate with diffusion timescales, such that deeper melt inclusions are hosted in crystals with longer timescales. Samples with multiple melt inclusions are generally found within the shallow plumbing system (<100 MPa) and with shorter timescales. The isobars and degassing curves were calculated with the average whole rock composition[27,28] and the solubility model from ref. [38]. The location of the degassing curve was based on the entire melt inclusion suites from refs. [27,28]; we show an additional open system degassing curve based on 3 wt% $H_2O$ (dotted line). The shift to 2 wt% $H_2O$ suggests there may have been some post-entrapment re-equilibration at relatively shallow levels. The melt inclusion pressures may not align perfectly with the isobars because their compositions range from 48 to 56 wt% $SiO_2$, highlighting the limitations of this type of plot for suites of melt inclusions. Error bars show ±2 s.d. for the $H_2O$ and $CO_2$ concentrations

in various ways (for example refs. [8–16]). $CO_2$ contents may be modified via cooling-induced crystallization, $CO_2$ partitioning into a vapour bubble, and decrepitation[13–16]. It is possible to estimate the amount of $CO_2$ lost to a vapour bubble based on the difference between eruptive and entrapment temperatures[40]. However, that procedure may not be appropriate for the Llaima samples because entrapment temperatures were lower than eruption temperatures (as calculated with the protocol described in ref. [40])[28]. As such the $CO_2$ concentrations reported here are uncorrected and considered as minima values.

Water contents may be modified via $H^+$ diffusion through the host olivine crystal[6,10,11,41]. Evidence for $H_2O$ loss includes significant post entrapment crystallization[9,10] and the precipitation of fine magnetite[41]. The melt inclusions in this study experienced less than 7% post-entrapment crystallization (determined with Petrolog 3 (see ref.[28] for details), suggesting that the compositions $H_2O$ contents have not been significantly modified. We also removed melt inclusions with daughter crystals from the study (see ref.[28] for more details). Additionally, our olivine crystals were sourced from lapilli, which experience limited degrees of post-eruptive diffusion[10]. Therefore, even though we cannot entirely exclude the possibility of some modification, we assume our samples experienced minimal diffusive $H_2O$ loss resulting from post-eruptive processes. However, given that the host crystals show residence timescales on the order of 10s to 1000s of days, the $H_2O$ contents of our melt inclusions have likely been re-equilibrated during the prolonged magma residence prior to the eruption. This is supported by the relatively narrow range of $H_2O$ contents in the melt inclusions. Since we consider the reported $H_2O$ and $CO_2$ values as minima and the concentrations

likely reflect the last depth of re-equilibration, we also consider the calculated pressures as minima values.

**Combining re-equilibration pressures and timescales**. We find that the highest saturation pressures correspond to crystals that preserve the longest diffusion times (Fig. 5; data compiled in Supplementary Data 3). Additionally, there is a general apparent decrease in pressure with diffusion time. The trends of $H_2O$ and $CO_2$ contents on a volatile saturation plot display a classical pattern, with a shape similar to a hockey stick (Fig. 5). This pattern exhibits moderate changes in $H_2O$ and a vertical trend of decreasing pressure and $CO_2$, followed by a near horizontal trend at shallower saturation pressures, with large decreases of $H_2O$ contents at low $CO_2$ concentrations. Such patterns are typically used to distinguish between open (Rayleigh fractionation) and closed (equilibrium) system syn-eruptive degassing[2,42], which has been in turn used to interpret the explosivity of eruptions[2,4–6]. In the next section we discuss the various interpretations for such trends and how the data may be related to monitoring information and anticipation of eruptions.

## Discussion

With our combined dataset we try to interpret how long the melt inclusions were at their last depth of re-equilibration prior to eruption, and how this relates to magma movement. However, since we are inferring the timescales for magma mixing from the host crystals, and the depth where this mixing happened from the melt inclusion data, several interpretations are possible. We propose three different end member scenarios below.

In end member 1, all melt inclusions were trapped at the same depth (e.g., the deepest level we observe) and mixing occurred at this same depth over a variable period of time, producing the spread of timescales found. Magma movement towards the surface could allow for partial re-equilibration of the crystals and inclusions at shallower levels. This could explain the spread of times and pressures we observe. However, we think this case is unlikely for two reasons. First, our data show a general correlation of decreasing pressure with decreasing times, and secondly the range of observed $CO_2$ contents would be hard to reproduce in this scenario (e.g., Figure 5).

In end member 2, melt inclusions were originally trapped at a variety of depths within the conduit, but magma mixing occurred only at the deepest levels. When magma ascended, the deeply sourced crystals and their melt inclusions moved to a shallower level, and re-equilibrated. In this scenario, we also require continued magma intrusion at depth, but decoupling of the magma mixing depth and the re-equilibration depth of the melt inclusions. However, this case would tend to produce a relationship of increasing times with decreasing pressure. Moreover, most deeply sourced crystals and melt inclusions would be re-equilibrated, thus erasing evidence of the deeper plumbing system. Neither of these trends are consistent with our observations.

Finally, we propose end member 3 where melt inclusions formed at various depths but magma mixing progressed upwards towards the surface, so that magma mixing and melt inclusion volatile equilibration/entrapment for any individual crystal occur at about the same depth. This would be more akin to melt percolation and reaction processes in a crystal-rich mush (for example ref. [43]). The eruption quickly evacuated the volcanic conduit at all depths, with limited time for syn-eruptive modification of their %Fo compositions. This case would lead to a pressure vs. time trend similar to, but perhaps better defined than that exhibited by our data (Fig. 5). It seems likely that our data reflect a combination of end-members 2 and 3. That is, some melt inclusions and their host crystals partly equilibrated with the melt

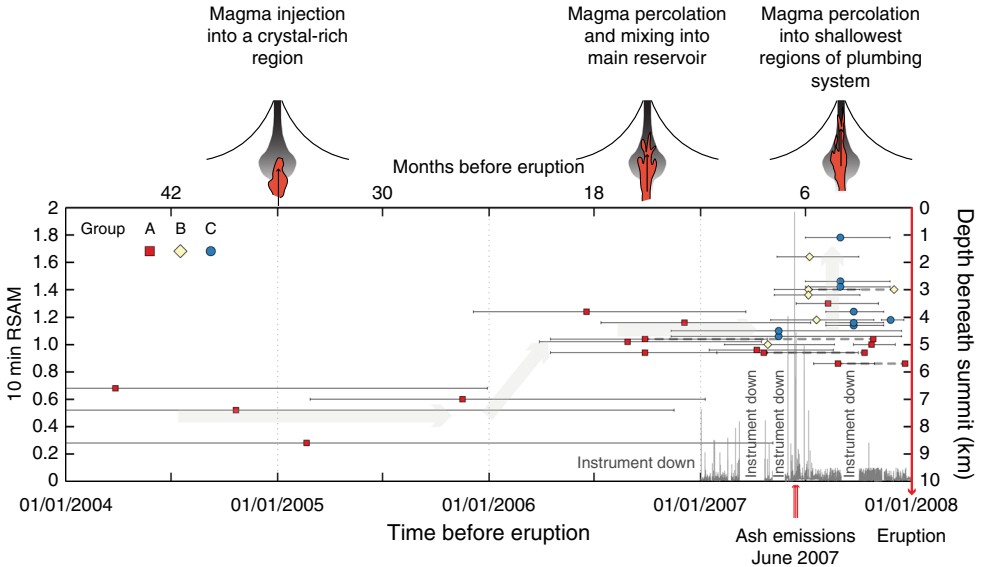

**Fig. 6** Combined geochemical, seismic, and observed datasets prior to the 2008 eruption. We observe that shallower pressures are recorded as the volcano gets closer to eruption. Also note that peak seismicity, ash-venting and some timescales from the crystals overlap, probably reflecting gas and magma movement towards the surface. Other crystals post-date the seismicity peak by a few months and may indicate that there is a reaction time for the crystal-rich system to record the processes of melt movement or/and that final melt movement towards the surface was aseismic. Also shown is a schematic cartoon showing the proposed timeline of magmatic processes is shown above. Gaps in the 10 min RSAM data indicate when the instruments were not collecting data. Grey arrows in the background show the general magma migration of magma path. Symbols connected by a dotted line show timescales for the complex crystals. Error bars show the uncertainty on the diffusion timescales based on the discrepancy from the DIPRA model

at shallower pressures as it migrated towards the surface and progressively reacted with the crystal-rich mush.

The re-equilibration depths provide information about the structure of the plumbing system and the timing of processes up to tens of days before the onset of the eruption. The eruption captured crystals that were distributed over a vertically extensive plumbing system that was evacuated rapidly over a short period of time as indicated by sampling of deeply sourced (based on volatile contents), unzoned crystals[27,28]. Although %Fo diffusion seems to record prolonged magma ascent and accumulation, it likely did not record the final ascent event (i.e., the eruption). Indeed, as shown in this and other studies, the resolution for %Fo diffusion is ≥1 day[20,21,24,26,36]. Thus, final, eruption-related ascent may be more faithfully recorded in other diffusion data-sets, for example diffusive re-equilibration of $H_2O$ in melt inclusions[10,12], or Li in olivine[44]. Future work could combine slow (%Fo) and fast diffusion chronometers to ensure temporal coverage of pre- and syn-eruption processes.

Our data may also provide insights on interpretation of open vs. closed system degassing scenarios for specific eruptions[37,45,46], as it has been traditionally assumed that the crystals and inclusions formed as the magma ascended during or immediately prior to eruption[46,47]. As such, the melt inclusions represent records of syn-eruptive processes providing information on the magma depths and degassing paths associated with that eruption[35,37,45–47]. If this were the case, very short residence times (days to hours) would be expected from the crystals and melt inclusions. Our data, however, reveal a different story—olivine hosts record magma mixing over months to years prior to the eruption with deeper crystals recording longer timescales than the shallower ones. Melt inclusions, then, must have been present for at least the same time as the zoning we have modelled in the crystals, and their trends in volatile contents cannot simply be recording eruption-related ascent and degassing. This is consistent with recent work that indicates that volatile contents within melt inclusions may be modified after formation,

sometimes significantly[9,10,14]. Therefore, we propose that the melt inclusions can be better considered as recording the background open-system degassing from the summit vent and a vertical arrangement of magma through the plumbing system (e.g., conduit or dikes), unless otherwise proven to be associated with entrapment during final ascent.

A similar interpretation for melt inclusion entrapment pressures was proposed for the 1974 Fuego eruption[48], and we suggest that other melt inclusion datasets may be recording analogous processes to those that we propose here. For example, melt inclusion data hosted in olivine crystals of the 2002 flank eruption of Etna are interpreted to record closed system degassing and rapid ascent[37]. However, olivine zoning patterns from the same eruption, though not the same crystals, indicate residence times up to 750 days[23]. Another example is the 1980 eruption of Mount St. Helens, in which the deposits contain melt inclusions in plagioclase and orthopyroxene with variations in $H_2O$ contents that have been interpreted as resulting from syn-eruption degassing[47]. Yet, orthopyroxene from the same eruption record diffusion timescales ranging from 15 days to 10 years (median 268 days)[19]. Thus, our findings and those from the literature suggest that melt inclusions do not necessarily, nor systematically record eruption-related ascent and degassing and we caution against direct interpretation of volatile contents trends from melt inclusions without coupled studies of the zoning patterns of the host crystals.

Previous studies have provided broad correlations between diffusion timescales and monitoring datasets[19,20,22,24], but lacked direct information about the depth progression of magmatic processes with time. The link between the pressure of melt inclusion last re-equilibration and their mixing timescales now provides some constraints on the magma ascent path and enable us to better interpret monitoring patterns and unrest at Llaima volcano (Fig. 6). We find that magma recharge and mixing began at least 4 years before the eruption at minimum depths of 8.6–6.6 km below the summit. The lack of measured deformation[49] prior to the 2008 year

eruption, combined with our broad timescale distribution, suggests that magma injection did not occur in large discrete batches. Rather, the magma replenishment probably occurred through melt percolation and reaction in the crystal-rich mush. About 1–2 years (200–600 days) before eruption magma continued to percolate upwards to depths of between 5.7 and 3.8 km, with continued mixing. Approximately 3–6 months before the eruption, magma migrated again, this time from ~4 km below the summit to within the volcanic edifice (Fig. 6). These profile-determined timescales occurred within weeks of elevated seismic energy release related to long period events at the end of May 2007 (Fig. 6). Such seismicity is classically interpreted to record gas and/or magma movement within the system[50], and the timing is consistent with the timescales and processes inferred from the reversely zoned olivine crystals. Moreover, the increased seismic activity occurred at the same time as three discrete ash emissions in June 2007, ~200 days before eruption. Although multi-station seismic data are not available prior to 2007, we know the activity between 2004 and 2007 consisted of seismic tremor, small volcano-tectonic events, and long period seismic events, which constitutes the general background behaviour at Llaima. We note that magma mixing timescales overlap, within uncertainty, with the timing of elevated RSAM values, which is likely recording magma and gas movement towards the surface. However, other crystals record a mixing event a few months later. These crystals could reflect delayed magmatic movement within the crystal-rich shallow plumbing system and/or other additional aseismic magmatic processes.

Overall, the magma ascent path obtained from combined melt inclusion and crystal zoning evidences progressive, prolonged magma mixing and ascent. The eruption tapped an extensive part of the vertically arranged, crystal-rich magma plumbing system carrying crystals and melt inclusions from various depths which record the various mixing processes that occurred prior to eruption. Rapid accumulation of a crystals from a wide range of depths has been previously suggested for eruptions at Irazu volcano based on Ni zonation in olivine[51]. The final stages of magma accumulation prior to the eruption was near coincident in time with increased seismic activity and ash emissions (Fig. 6).

Melt inclusion and diffusion chronometry investigations have opened up our understanding of the timing and location of magmatic processes prior to eruption. Our study highlights the critical importance of assessing the crystal zoning patterns and timescales, and their relation to different populations of melt inclusions, for a more complete understanding of magma ascent, degassing, and associated unrest. Despite the complexity of our dataset, we believe that systematic application of our approach to many case studies should lead to more realistic conceptual models of magmatic systems and open the door to deterministic forecasts of volcanic eruptions.

## Methods

**Sampling protocol.** Olivine crystals were sourced from the uppermost portion of the basaltic andesite tephra deposit emplaced during the initial phase of the 2008 eruption; samples were collected in January 2009 and January 2011. Building on previous work[27,28], we obtained major element zoning patterns, element concentration maps, and the crystallographic orientation of 22 olivine crystals (see Supplementary Data 1–4 for details). Specifically, we collected the host olivine compositional data with the aim to understand how the associated inclusions relate to the petrogenetic stories recorded by the crystals themselves.

Scoria samples were crushed and sieved using 25, 40, and 80 mesh sizes[27,28]. We handpicked crystals from the 25 and 40 mesh sieve fractions of the crushed scoria. Crystals with clear, glassy melt inclusions were selected for analysis. We note that single vapour bubbles were observed in several inclusions (Supplementary Data 1 and Supplementary Fig. 1[29]).

**Mineral textures and zoning.** Textural and morphological features of the olivine crystals and melt inclusions were characterized using a petrographic microscope and backscattered electron (BSE) images from the JEOL JXA-8530F electron microprobe (EPMA) at Nanyang Technological University (NTU), Singapore. X-ray intensity maps of Fe, Mg, Ca, Al and P were obtained on selected crystals using

the EMPA set with the following conditions: 15 kV, 100–300 nA, 60–100 ms dwell time, and 1–1.25 µm pixel size (Supplementary Fig. 1).

To guide traverse location selection, we inspected the zoning gradient on BSE and element maps using the profile tool in ImageJ[52]. This helped to decide the traverse length and analytical spacing within the profile. Profile lengths ranged from ~ 200 to 450 µm. Spacing between individual EMPA points was up to 5 µm in plateau regions where the compositional gradient was shallow, and 2.5–3 µm where the compositional gradient was steep. Where possible, profiles were collected perpendicular to a crystal face, or to the main Fe–Mg chemical gradients as seen in 2-D X-Ray maps.

**In situ mineral orientations.** Olivine crystallographic axes were measured using a JEOL JSM-7600F electron microscope (FE-SEM) equipped with an Oxford/HKL electron backscatter diffraction (EBSD) system, also at NTU. Operating conditions for the FE-SEM were: 30 kV, 30 nA, and a tilt angle of 70°. Crystals underwent automated beam scanning, where the beam moves over a selected area on the stationary sample. Crystallographic axes orientations for each crystal were derived from an average of at least three indexed points with <1° deviation and were consistent with the observed habit in two-dimensional cross section. We plotted the data onto a lower hemisphere stereographic projection and measured the angle between the axes orientation and the element profile locations using the Stereonet 9 software package[53].

**Mineral compositions.** Rim to core quantitative compositional profiles of 22 crystals were obtained for Si, Al, Ca, Mg, Mn, Fe, Cr, and Ni using the EPMA (for complete dataset see Supplementary Data 2 and ref. [30]). Instrument conditions were 15 kV, 20–40 nA, and 1 µm beam diameter. On-peak counting times were 20 s (Si), 30 s (Mg, Fe), 40 s (Ca), and 80 s (Al, Mn, Cr, Ni); high and low background count times were half the peak times. Analytical error for Si, Fe, and Mg is less than 1% relative, it is <5% for Ca and Mn, and about <10% for Ni. Replicate analyses of the olivine standard were collected and averaged, and compared to reported values to evaluate the accuracy. $SiO_2$, MgO, MnO, and NiO varied from reported values by <3% relative. FeO varied from reported values by 9%. CaO is not reported for our standard therefore we are unable to assess the variation in our analyses for this element. Some profiles were collected using the mean atomic number (MAN) background correction techniques, where the background is modelled based on the MAN of the analysed sample[54]. This technique reduces the analytical time by approximately half, while still producing precise measurements[55].

A more detailed inspection of sample bk4-1 showed the uneven zoning pattern in the BSE image and element maps show us that the original EPMA traverse was not ideally located. Therefore, we collected an additional profile with a calibrated BSE image. There is a linear relationship between greyscale values and composition in many minerals, including olivine[56], plagioclase[57], and orthopyroxene[19]. This approach has been used to quantify compositional zoning patterns at higher spatial resolutions than those produced by EPMA, with an uncertainty of <1 mol%[56,57]. We calibrated the high-resolution BSE images using at least seven analytical points from the original EPMA traverse. Greyscale values of the analytical points were retrieved from circles with 2–3 µm diameters selected in the ImageJ software package. The grayscale Fo calibration returned an $r^2$ value of 0.99 (Supplementary Fig. 2).

**Diffusion modelling protocol.** Chemical gradients (Fe–Mg, Mn, Ni, Ca) in olivine were modelled in one dimension with the DIPRA software program[58], which accounts for uncertainty in temperature and composition[59]. We accounted for diffusion anisotropy by determining the olivine crystallographic axes using EBSD and the angles between the electron microprobe traverse and the three axes[60]. Temperatures derived from olivine rims and nearby matrix glass average 1140 ± 26 °C[26]. We also modelled temperatures using MELTS[61] using the average whole rock composition: 51.5 wt% $SiO_2$, 1.02 wt% $TiO_2$, 18.25 wt% $Al_2O_3$, 2.76 wt% $Fe_2O_3$, 8.94 wt% FeO, 0.16 wt% MnO, 5.75 wt% MgO, 9.89 wt% CaO, 3.09 wt% $Na_2O$, 0.57 wt% $K_2O$, 0.19 wt% $P_2O_5$, and 2 wt% $H_2O$[28]. Based on the observed abundance of melt inclusions with apparent entrapment pressures between 50 and 100 MPa[27,28], we used 100 MPa for the model calculations. The liquidus temperatures modelled for olivine %Fo contents of 80 and 77 (the rim compositions) were ~1150 °C and 1120 °C, respectively. Therefore, chemical profiles were modelled ~1150 and 1120 °C depending on the olivine rim compositions. We used an uncertainty of ±30 °C, which is typical of many geothermometers[59]. This uncertainty also overlaps with the 1σ variation of our dataset. Given the low dependence of pressure on the diffusion coefficient at our pressure range, we set pressure to 50 MPa. We used the measured intrinsic oxygen fugacity obtained with XANES measurements on melt inclusions; this value is 0.2 log units below the Ni–NiO reaction buffer[28].

For initial conditions we used either a step function or homogenous profile, depending on the observed zoning (Supplementary Data 3). Seventeen crystals exhibited simple %Fo zoning which could be modelled with a single timescale. Five crystals exhibit complex zoning, with multiple alternations in %Fo content; these were best modelled by two timescales—one for the interior and another for the rim (e.g., see Fig. 1, br1-4). To assess if any of the %Fo profiles were significantly affected by growth zoning, we conducted two tests (following approaches outlined in ref. [40]). First, we measured the zoning in different axial directions. We found that the diffusion length is proportional to the experimentally calibrated diffusion anisotropy. This would not be expected from growth-related zoning because diffusion are more

anisotropic than growth (for example refs. [60,62,63]; see Supplementary Data 4 for details). We also modelled Ni, Ca, and Mn, concentrations to compare with timescales from %Fo modelling. Mn timescales correlate very well ($r^2 = 0.98$). The lower $r^2$ values for Ca and Ni vs. %Fo ($r^2 = 0.91$ and 0.82, respectively) suggest there is some mismatch; this could be related to minor growth zoning, and/or analytical uncertainty associated with measurements of minor elements in olivine. Despite the possibility for a minor role of growth, the generally good correlation between most the trace element modelling and %Fo timescales suggests that the observed %Fo zoning is controlled by diffusion rather than growth[62] (Fig. 3). This is consistent with recent work that suggests that growth related Fe–Mg zoning tends to be a minor component in many natural crystals[64].

**Melt inclusion volatile contents**. Melt inclusion volatile contents were collected with Fourier transform infrared spectroscopy (FTIR) or secondary ion mass spectrometry (SIMS), which determined the selected mounting material. Those crystals analysed via FTIR were mounted in epoxy and polished to expose two sides of each melt inclusion. Crystals analysed by SIMS were mounted in Crystalbond, single-side polished and then transferred to an indium mount for the SIMS analysis. For both approaches, we polished with diamond grit to 0.5–μm and used well established polishing techniques.

FTIR was used for melt inclusions in 3 crystals, and SIMS for melt inclusions in 19 olivine crystals. Here we provide a brief summary of the analytical techniques; for a more in depth discussion, please see refs. [27,28]. FTIR analyses were completed using the Bio-Rad Excalibur spectrometer or the Nicolet 6700 Analytical FTIR hosted at The National Museum of Natural History, Smithsonian Institution. We collected spectra from 1000 to 6000 $cm^{-1}$ using a liquid nitrogen-cooled MCT detector, KBr beam splitter, and tungsten halogen source. The instrument aperture ranged from $11 \times 22$ to $44 \times 44$ μm. Melt inclusion thickness was evaluated with either a piezometric micrometre, and/or using the wavelength of fringes from 2000 to 2700 $cm^{-1}$ [65]. Water concentrations were collected from background-corrected spectra and using the peak height from the total $H_2O$ peak at 3535 $cm^{-1}$ [66,67]. We measured $CO_2$ concentrations on the background-corrected spectra using the $CO_3^{2-}$ doublet between 1515 and 1435 $cm^{-1}$ [38].

SIMS analyses were collected using the Cameca *ims*-6f at Arizona State University and the Cameca *ims*-1280 at Woods Hole Oceanographic Institute. The samples were pre-sputtered for 4 min during which the beam was rastered over a square area of 20–30 μm. Analyses commenced after pre-sputtering using a $15 \times 15$ μm beam. Count times were 5 or 10 s for $H_2O$ and $CO_2$, with 10–30 cycles per analysis. $H_2O$ and $CO_2$ concentrations were converted using calibration curves developed using well establish standards[27,28].

We calculated the minimum pressures using the $H_2O$ and $CO_2$ contents and assuming volatile saturation and the solubility model of ref. [39]. Pressures were converted to depths based on a crustal density of 2500 kg m$^{-3}$, and also accounting for the volume of the conical volcanic edifice[27].

**Seismic data collection**. Seismic data (Supplementary Data 4) were collected by OVDAS-SERNAGEOMIN (Volcano Observatory of the Southern Andes—Chilean National Service of Geology and Mining), using single-component seismographs, positioned south-southeast of Llaima's summit crater at distances of 9.0 and 17.3 km, respectively. Seismic activity was assessed using real time seismic amplitude measurements (RSAM) based on data from the station 9.0 km from the summit (using approaches in ref. [68]).

**Data availability**. Images from Supplementary Fig. 1 that support the finding of this study are available in the EarthChem data repository with the identifier DOI[29]. The complete olivine compositional profile dataset that support the findings of this study are available in the EarthChem data repository with the identifier DOI[30].

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

## Acknowledgements

Discussions with K. Lynn, M. Myers, J. Ball, A. Graettinger, G. Valentine, T. Girona, B. Welsch, E. Cottrell, B. Taisne, helped to improve the manuscript. J. Herrin helped with EBSD and EPMA data collection. Three anonymous reviewers and especially P. Ruprecht provided many comments and critical feedback that improved the manuscript. This research was funded by a National Science Foundation East Asia and Pacific Summer Institute Fellowship (OISE-1309590) to D.C.S.R, a National Science Foundation (EAR-0828070) to E.S.C. (while at the University at Buffalo), and a Singapore National Research Foundation - Ministry of Education grant (MoE2014-T2-2-041) to F.C.

## Author contributions

D.C.S.R. and C.BdM. contributed the samples from Llaima as part of their PhD dissertations. D.C.S.R., J.A.C., and E.S.C. conducted fieldwork and sampling. L.F. provided access to and interpretation of the seismic data. D.C.S.R. and F.C. modelled the diffusion timescales and wrote the manuscript with contributions from all the co-authors.

## Additional information

**Competing interests:** The authors declare no competing interests.

