## [Peer Review File · Nature Communications]

Reviewers' comments:

Reviewer #1 (Remarks to the Author):

Review – Ruth et al. *submitted to Nature Communications*

The submitted manuscript by Ruth et al. uses Fe/Mg profiles from the rims towards the cores of olivines and diffusion modeling of the profiles to infer timescales of residence of olivine crystals in the magmatic plumbing systems of Llaima volcano (Chile). As these olivine crystals also host melt inclusions, they then use melt inclusion volatile data to infer storage depths of the olivine crystals. Combining this with seismic data, the authors present an integrated story about the temporal and spatial evolution of the plumbing system below the volcano leading up to the 2008 eruption.

The dataset presented is very interesting (and represents one aspect of the detailed work that the first author has done on Llaima volcano), however I have issues with the how the caveats/uncertainties associated with the data and the modeling are handled. In particular, there are important points to be discussed and incorporated into the conclusions concerning the volatile data for the melt inclusions, the assumptions that go into the diffusion chronometry, and the viability of combining inferred entrapment depths for melt inclusions with the diffusion chronometry in their host olivine. I think that this could be done sufficiently within the length of the current paper (with much of it located in a supplemental or analytical methods discussion), but it needs to be presented.

Ultimately, I think that this sort of detailed study is much needed in studies of melt inclusions and volcanic eruptions. It is quite a commendable and rich data set and I hope to see it published. A study successfully marrying information from temporal constraints provided by host olivines with potential pressure constraint from melt inclusions would be an excellent contribution (and would certainly guide the field), however as the manuscript stands, the uncertainties with doing so have not been fully explored. I go into these issues under the section entitled “Major Comments”.

The manuscript in general is well written with only a few typos or places that could be more clearly explained. I have highlighted these in the “Minor Comments” section.

I hope my comments are useful to the authors and can use them to make this a stronger study.

Sincerely,

Claire Bucholz, cbucholz@caltech.edu

MAJOR COMMENTS:

1) *The uncertainties associated with the modeling are not discussed in sufficient detail.*

-Temperature: Your estimates need to be discussed in more detail as diffusivities are so strongly controlled by this variable. Where do the $\pm 30^\circ\text{C}$ uncertainties on the temperature estimates come from? Are the plagioclase and liquid compositions used in the plagioclase-liquid given somewhere (from coexisting plagioclase phenocrysts in the scoria(?) and whole rock or glass analyses?) (Also, the reference given for the $1150^\circ\text{C}\pm 10^\circ$ (reference #12) does

not seem to be appropriate.) Why do you use the anhydrous version of this thermometer when you know that the magmas that these olivines crystallized from are hydrous?

-Have you tried multiple diffusion profiles on a single olivine grain? Perhaps along different orientations? Do they give the same timescales?

-When setting up your initial conditions, you choose two endmember scenarios (step-function versus homogeneous). For the step function profiles, this (if I understand correctly) is assumed to replicate the scenario of magma mixing, whereby a new rim of olivine grows around the core with a different Fo content. However, this growth must certainly take some time. How does simultaneous growth and diffusion affect your estimated timescales? It would likely shorten them, I would think. I understand that the model you use may not incorporate growth into the calculations, but this is something that you should at least discuss qualitatively.

-The melt inclusion profiles don't always plateau towards the cores (e.g., the group C olivine that you show in Figure 1, bul3, plate1, etc). In these cases, how do you know what to choose as your initial conditions?

2) *The coupling of timescales from olivine diffusion profiles and the melt inclusion pressure estimates may not be recording the same story.*

As I'm sure all the authors are aware, melt inclusion H₂O and CO₂ contents can be affected by post-entrapment diffusive re-equilibration of H⁺. With diffusive loss of H⁺ due to any degassing-induced concentration gradient in H₂O during ascent through the crust or eruption, water concentrations in the melt inclusions will decrease and CO₂ may be exsolved into a vapor bubble. If these processes aren't considered, then you will only be getting a minimum pressure estimate for depths of entrapment. The trends in CO₂-H₂O space produced by this process are identical to that of degassing. It also doesn't only happen during eruption, it can easily happen as the melt is migrating from depth in the crust to upper level reservoirs.

I haven't read the previous work done on the melt inclusion in these olivines, and perhaps the authors have dealt with this in a prior manuscript, but it should be discussed here and factored into their final story about the structure of the magmatic system below Llama.

For example, for the four melt inclusions that preserve higher volatile contents (in particular CO₂), how do their radii compare to the other melt inclusions in the study? Larger melt inclusions will re-equilibrate more slowly due to their larger volume:surface area ratio. Could this be an influencing factor?

Are their vapor bubbles present in these melt inclusions? They, of course can be formed through processes other than just diffusive loss of H⁺, but it would be good to note.

Do melt inclusions from the same olivines preserve similar volatile contents and thus calculated pressures? Or are they disparate?

Do all your profiles traverse regions populated by melt inclusions, suggesting that the calculated diffusive timescales are for regions of the olivine that grew during melt inclusion entrapment. In figure 2A the profile stops short of the core and only covers ~1/2 the imaged melt inclusion.

I hope this doesn't come across as a shameless troll to get my papers concerning this topic cited, but I do think it needs to be given some consideration when you are taking two potentially independent systems (volatile contents in melt inclusions as pressure estimates for entrapment and Fe/Mg profiles in olivine to estimate diffusive relaxation timescales).

3) There are only four olivine crystal recording long residence times. These particular samples deserve extra attention.

As inferred from your Table, the high pressure, long residence times melt inclusions are: bul3, plate1, br1-2, and B22. The data for the first three is presented in your supplementary table, but I could find the diffusion profile for B22. Bul3 and plate1 are distinct from all the other profiles presented. They don't have a clear plateau towards the core, demonstrate significant scatter, and don't really even preserve a classic diffusion profile. How do you know that they are recording diffusive processes? How do you take into account the uncertainties associated with the scatter of the data? Why are these olivine profiles so different and how does that impact the ultimate story you construct?

MINOR COMMENTS:

Line 25-26: are you referring to similar melt inclusion compositions here?

Line 100-102: I think this is a really cool observation between melt inclusion clustering and P zoning(!), but again how does subsequent slow growth affect your diffusion modeling?

Line 114: No figure 2a was provided with submitted manuscript.

Line 143-145: I found this sentence confusing. I'm not sure what "continuous summit degassing" means.

Line 203: "diffusion coefficient" should be "diffusivity"

Line 209-210: What were the XANES measurements of? $\text{Fe}^{3+}/\text{Fe}^{\text{T}}$ in melt inclusion glasses, I assume. Should also be ΔNNO .

Line 233: What densities do you use in your pressure calculations.

Table 1: Can you give H_2O and CO_2 contents here too so that the reader can assess on their own the pressure estimates? It will also help the reader in reference to Figure 2. Also, no need to put a comma after "output" in the Table title.

Figure 1: Why do your plane polarized light images look so scuzzy? Did you dissolve away the scoria clinging to the olivines before analyzing these? These images aren't super helpful as the olivine and melt inclusions are so obfuscated.

In the caption, please clarify the discussion about "open" versus "closed" melt inclusions. Do you mean there was diffusive re-equilibration of certain elements between the

melt inclusions and the external melt through the host olivine. You refer to “patchy zoning”, but for what elements? What’s the actual pattern consist of? (This obviously doesn’t all need to be in the figure caption, but discussed somewhere if you are going to bring it up).

Figure 2: You volatile solubility model reference should be to 29, not 28. Also, please define abbreviations of CSD and OSD as closed versus open system degassing.

Reviewer #2 (Remarks to the Author):

The manuscript "Melt inclusion residence times reveal the magma dynamics of open vent volcanoes" by Ruth et al. is combining melt inclusion (MI) volatile chemistry with host-olivine elemental zoning modeling to connect the temporal record with degassing at Llaima volcano, Chile. The approach is truly novel and potentially very interesting. However, I was not convinced that their conceptual model is properly taking into account what we know about melt inclusions.

Most importantly the authors claim that MI studies typically assume that MIs record the syn-eruptive degassing history of a volcano and therefore build an argument to provide an alternative model. This is unfortunately a strawman argument as it is well agreed upon in the community that MIs record the minimum pressure (depth) of last equilibration that is rarely equated with syn-eruptive ascent *sensu stricto*. Instead, the MI record just provides a general degassing pathway for the ascending magmas in the respective magmatic system. Underlying that idea is that different magma batches/parcels that arrive at upper crustal levels will experience similar degassing behavior, so they do record the degassing path associated with eruption, but only in the general sense. The fact that there are meaningful trends in CO₂-H₂O plots is commonly explained by magmas stalling on their way to the surface and thus having sufficient time to equilibrate to the new CO₂-H₂O equilibrium conditions. Thus, I think instead of trying to build a somewhat misguided argument against syn-eruptive degassing records, the data provides a very interesting case that shows that crystals of different residence age and storage region are assembled prior to eruption and this done likely over short timescales.

Secondly, despite the short format of this journal, I think the authors need to provide significantly more information for the reader to assess the manuscript. I am providing several key datasets to improve in the line-by-line comments, but I would like to summarize here the following aspects: 1) The samples, crystals, and MIs are insufficiently described. Where are samples coming from? Are crystals from one or more samples? How big are the MIs and the crystals (see below). 2) this is elemental diffusion modeling study on microprobe data, yet no microprobe data is presented. 3) Melt inclusion information may have been reported elsewhere, but in the current form it is impossible to follow the manuscript without going back to the older studies. 4) A short petrography discussion, at the very least in the supplement.

Thirdly, there are several issues regarding referencing and clarity of figure and captions. Especially when comparing table 1 with figures 2 and 3, I was very confused and was not able to resolve the issues (see below).

Minor comments:

II. 27: There is a gap in the arguments between sentence 2 and 3. The causality of how residence times and a range of such residence times are related to MIs is not developed.

II. 54: How is open vent system defined? What makes Llaima an open vent system over other frequently erupting volcanoes?

II. 58: The authors suggest that MI papers conventionally discuss MI data as a syn-eruptive record. They do not provide any citations and I disagree with this being true. MIs are commonly interpreted as a suite that provides a context about magmas of similar composition, but I rarely see that people link them specifically in time. You refer to this again II.114, but this seems like a strawman argument to me.

II. 65: Reference 11 is not correct here! This paper (Cashman 2004) predates the 2008 eruption!

II. 72: Can you be more specific? How many crystals did you look at? How many did you ultimately analyze by quantitative means and how many did you choose to model? Also, how many samples did your olivines come from? Where do these samples come from (both locality as well as level within the deposit)? Are these samples covering different episodes of the eruption? There are only

14 crystals shown (images in Supplement), but there are 22 crystals mentioned both in the table in II. 78.

II.105: It is said that 27 crystal traverses were modeled (that is also the number of olivine traverses in table 1), but I count 28 datapoints in figure 2 and 3 (not counting the open symbols). Also, I cannot reconcile the datapoints between figure 2 and 3: e.g. there are four data points that are deeper than all others (in figure 3). These points are showing the longest residence times: in figure 2, 2 point are (very) deep and have long residence times. There are two other points with long residence times (red), but they are at the same pressure as or lower pressure than three shorter residence time datapoints. It may help to add the CO₂/H₂O data to Table 1 so that one can better track which MIs belong to which olivine.

II.112: The authors argue that there are no discernible peaks, however, the complex crystals for which a second timescales for near-rim zoning was modeled clearly show a cluster of the shortest timescales for the near-rim zoning (not generally surprising). They make up more than 50% of the < 100 day timescales, and A21 with 37 days is the only non-complex crystal among the population of really short (< 50 days) timescales. This is not addressed at all.

II.116: This is a very important statement. Yes, the data shows that one cannot treat these as a single crystal population. But people do not do that anyways for MIs! They treat them (if at all) as single MI populations, which may seem like a semantic thing, but it is not. I would argue most MI studies use MI populations as a way to capture the general behavior and degassing path of magmas involved in an eruption. Few studies claim that the MI population is limited to a single crystal population that experience the various conditions in the (very) narrow definition that is applied here, ie the crystals were spatially and temporally together. Even the term crystal population is commonly used much more broadly. MI studies rarely claim that the MIs all represent the same parcel of magma (which is implied here for those studies). Moreover, if you make that claim, it is somewhat a pre-determined outcome in this study given that you look at olivines (and MIs) from more than one sample, at least that is what I got from the provided information!

II.125 How much post-entrapment modification occurred? As described above there is basically no information about the MIs! Sizes, compositions, in equilibrium with host?, location within the grains, presence of exsolution bubble... Many studies in the last years have shown that this information is important when estimating pressure via CO₂ (and H₂O).

II. 132: "This interpretation ...". It is not clear what "interpretation" is referred to here. But I assume the authors claim that one cannot relate the MIs in a CO₂-H₂O plot to open or closed-system behavior. In any case, the statement that "the interpretation" "implies" that the MIs formed during ascent or immediately prior to eruption is incorrect! The entrapment pressure that is recorded is the pressure of last equilibration. Of course the crystal + MI can be stored for a long time at these conditions. There is no question whether the ascent and duration of the assembly of all the MIs was short (otherwise they would re-equilibrate as is also explained by Lloyd et al. 2013, who they cite). So what it really means is that the eruption is composed/assembled of a diverse range of "magmas" that are brought together from zones as deep as 9 km as well as shallow as ~ 2 km, i.e. the eruption was not just tapping the shallow part of the magma system. And also what it means is that the upper part of the system has few (or no) crystals that get stored there for a long time, while the deeper storage system is not a direct "path" for magmas, but that they commonly are stored there and later remobilized during the eruption.

II. 138 again, this is a strawman argument, the studies rarely state syn-eruptive formation of melt inclusions. Instead they state depth of last equilibration (that by definition implies longer storage).

II.151: Here the authors do exactly the distorted interpretation they earlier criticized that this is one batch of magma experiencing a specific history, while the olivines likely reflect different storage regimes that were assembled late, just prior to eruption.

II. 155: Crystal dissolution. This is the first time the authors mention that the crystals get consumed. Seems, this is important to elaborate on a little more. I am missing at least a short petrography section (at the very least in the supplement).

II. 172: I guess here one should reference Table S1. However, here it would be also important to add more information about the crystals (size of the crystal, the inclusions, the distance of the inclusions to the rim, ...)

II.184: There is not table with actual microprobe data and no information about uncertainties and detection limits. Were reference materials analyzed together with the unknowns? I guess some of that information is in other papers, but there is no reference to any other publication.

II.195: How do zonings in Ca, Ni and potentially other minor elements compare to the major elements? What timescales do they suggest?

II.202ff: I am surprised there is no estimate of temperature using the olivines and their melt inclusions, as Ol-Melt is among the best known/tested thermometers! How about the temperature estimated from olivine-matrix melt or olivine/whole rock. All these estimates provide a range of temperatures that probably need to be considered and if the possible temperature range were to increase then that would probably be more robust.

II.209: I am not following how the oxygen fugacity was determined. It is said that it was obtained by XANES, but the reference that is used is a general review paper on volatiles in magmas that does not address Llaima magmatism nor does it talk about XANES as a technique to get after fO₂. Is reference 5 the correct one?

II.230 There is no detail on how these volatile concentrations were determined. Of course there are references to Ruth et al. and Bouvet et al., but at least a brief summary of the methodologies should be added. Is this SIMS, FTIR? Where exsolution bubbles taken into account?

Figure 2: The caption references the wrong citation, exchange 28 for 29. I would not refer to "samples" but "olivine traverses" or "olivines", whatever is correct. In any case, "samples" is not helpful as it seems that there are several (tephra) samples that were used in this study, from which X olivines were separated and Y traverses were modelled. There is no explanation of CSD and OSD, presumably closed-system and open system-development.

Figure 3: There is no explanation of what the open system and filled systems refer to. The legend in the upper left corner is not helping as it states just that. It does seem that the shift to the shallowest MI record coincides with the seismicity in June 2007. However, given the short seismic record (basically just 2007), it is difficult to assess how special the episode in June 2007 is as for a significant fraction of the MI datapoints there is no seismic record.

Philipp Ruprecht

Reviewer #3 (Remarks to the Author):

The authors present integrated olivine composition, chemical zonation, diffusion timescale calculations and volatile concentrations from melt inclusions from a total of 22 olivine crystals from a single eruption of Llaima volcano (Chile). The novelty of the paper is this integration; as a way forward in thinking about how to link evidence of volatile behaviour to the record of solid and melt phases from volcanic plumbing systems. Their key message is that assumptions regarding the timing and physical context of melt inclusions are undermined when integrated with diffusion timescales calculated using data from the same crystals. They also provide a model interpretation of magma migration through the upper 10 km of crust. They assert that their work will provide a more realistic view of plumbing systems which will support deterministic forecasts of volcanic behaviour.

Regrettably, the manuscript is not ready for publication on a number of scientific, data-quality, stylistic and standard-of-writing grounds. I have provided suggestions for edits in some detail to help reveal what I think has the potential to be a very good paper. The motivation and concept are excellent, and this is a timely and topical contribution. Much of the data reduction, however, needs a re-appraisal, and most of the content and line of argument needs a thorough re-think.

Through the edited document and in the points below I demonstrate how the interpretation leans too much on assumptions that cannot be simultaneously held, and far too often very important points are farmed-out to references that a Nature Communications reader would probably not have the motivation to track down and assess, let alone possess knowledge of already. It would take the same amount of words to state these arguments, so please do this. As these instances

accumulate through the paper the reader gains the impression that the timescales aspect of the work is actually all the new data that has been produced, yet the prior 'state-of-the-art' (i.e. all the knowledge before timescale calculations) is never presented as distinct. This leaves the impression that there was no central hypothesis or question that the timescales are used to test, at least it is unclear. The result is a considerably woolly and often circulatory logical chain that is very difficult to unpick. Unpicking it reveals some serious problems.

Please do not be discouraged - this is new and exciting science you are conducting and I have no doubt it will be published. Finding order in crystal cargo 'chaos' is delicate work and I think your approach should be applauded; volatile content in MIs is an independent (or at least semi-independent) line of evidence. This makes it all the more important, however, to accept the limitations of the dataset size and build carefully from the least refutable evidence and relationships to the most speculative ideas of how it all fits together. It is unclear in this MS as to how the evidence has been weighed, which is essential to make leaps you take convincing.

Major comments (minor comments in tracked-changes document*):

1. Title implies plural, yet the study is on one volcano. There is no justification offered why this study should implicitly or explicitly tell us things about other systems.
2. The crystal diffusion timescales are not timescales of the melt inclusions as stated in the title. The textures clearly demonstrate that disequilibrium was imposed after the melt inclusions were enclosed in most cases. How long after is not addressed. Why wasn't a single crystal really taken apart and all the inclusions analysed to make some sort of assessment on how the 'stratigraphy' of the crystal relates to any 'overprinting'? There might be potential to see a liquid line of descent (e.g. in Al or Ca?) from core to rim, for instance, and then the Fe-Mg contents could be altered by partitioning with the surrounding crystal as it changes Fo...
3. It is simultaneously assumed that 1) the moment a melt inclusion was captured is the same moment the diffusion stopwatch 'starts', and 2) that many of the crystals were resident before magma-mixing. Only one or the other assumption could be held, or maybe neither! What mechanism are the authors suggesting is responsible for (1). Are there data to support (2), or is this regarding resident magma as fact - if so why is this reasonable? If there really was a combination of the two, what evidence is there here to really discriminate between them? What assumptions need to be held for that interpretation to be internally consistent? Assigning groups on the basis of melt inclusion texture doesn't seem up to this task.
4. The (limited) discussion of the EBSD data is worrying. Were there a minimum of 7 'point and shoot' style measurements taken per crystal? If so, how were they distributed and why? Why wasn't mapping conducted? If it was, at what resolution were they gathered, and why was the indexing so very poor? While it could be reasonably assumed that these magmatic minerals grew free in a melt and thus should have single orientations, experience shows that if the indexing rate is poor the orientation data (even if they are self-consistent) are wrong.
5. There is no description at all of sample preparation. This in itself is troubling, since rounding effects imposed, for example, by alumina powder polishing grits do seem to have a non-linear effect on signal received at detectors, and these can affect the profile shapes derived from spot analyses as well as maps (backscatter or element concentration). A quick check of the Z height on the EMP spot-analyses might help here.
6. There were two different analytical methods used to gather quantitative in-situ chemical data, yet no discussion of how they were verified against each other, and whether or not a statistical difference in results was observed. MAN backgrounds should produce the same results in much less time (although this of any other reason is not offered as supporting the choice made), but surely there are crystals that both approaches were employed upon to check? You have two populations of rims, and two methods of compositional data acquisition – please put the reader's mind at ease by showing there is no correlation.
7. Rims: is there really no high-Fe final quench rim of a um or two? This is a surprise and would be worth stating. What is the equilibrium melt for each of the crystals on eruption? Does this match the composition of the surrounding glass (or is that resin on those images).
8. One of the three crystals shown displays the profile extending from a corner, which suggests

the influence from these different directions could have affected the measured profile shape. How has this been dealt with?

9. Is there any scientific basis for assigning crystal groups on the distribution and number of melt inclusions from experimental studies? This seems to be quite arbitrary and doesn't itself hold any petrologic meaning, at least none that is described or discussed. Surely zonation 'style' holds more potential for grouping crystals into populations, for these at least might be argued to hold some meaning as to magmatic environments through time? What controls the number of melt inclusions in olivine?

10. Statistical robustness is claimed, yet saying that (paraphrasing) 'the other eruptions have similar zoned crystals, this eruption has a variety of zoned crystals, therefore this is a representative dataset' does not hold tightly as an argument. There must be a demonstration of how this assessment was performed with some degree of specificity.

11. Magma-mixing is used to explain the crystal chemical and zonation diversity, yet there is no independent evidence, like whole rock compositions, glass compositions, melt-inclusion trace element compositions etc. either referred to explicitly or presented here to support this. Alternatives such as xenocrystic origin etc. are completely ignored. How would this change the story?

12. There is no statistical treatment of the data to support the main assertion that timescale duration correlates with depth. There is a loose association on the basis of 22 crystals that each seem to have their eccentricities, yet this dataset is simply not large enough to make for a convincing argument. The more you interrogate figure 2, the more you see exceptions until they outnumber the trends. I think there's probably something here, but at present the very strongly stated conclusions are not supported by equally convincing trends.

13. The poor structure of the paper makes for difficult reading. For instance; some ideas are re-introduced multiple times, some observations pop up surprisingly in the same sentence as a discussion point, and some phrasing really does smack of serious question-begging. It might be a structural issue only, yet when I think I follow what the authors are saying question-begging is very clear on occasion.

14. Some interpretations in other works are presented as irrefutable fact, and the authors use these many times to hang their arguments on. They simultaneously describe some previous work as 'improper' and this only serves to frame their work as highly model-driven.

15. There appears to be several instances of incorrect terminology, which reads as if the authors are not as familiar with these concepts and methods as I know them to be. I suspect these are simple mistakes, but I do feel strongly that with an author list such as this these mistakes simply should not be here.

16. The manuscript reads as if it is hurrying along and skipping over the central scientific arguments, which corresponds in most instances to those which are 'farmed-out' to references. I think the authors have a story here, but the writing is choking up the narrative. I would recommend a fresh start, because that is the most likely way this paper will have the largest impact.

17. I think the main assertion that melt inclusions are thought to record syn-eruptive processes but actually are formed earlier is an unfounded straw-man argument. Final cooling and crystallisation, element mobility etc. are certainly recorded during the eruption. Why is this a problem? This is not the same thing as assessing when and where those inclusions became enclosed, and I haven't seen a paper trying to make this claim that in this manuscript the authors pose as such a problem. Syn-eruptive processes probably convolve or destroy the primary signals that we wish to obtain, but I don't think anyone would say that the melt inclusions always form at shallow levels... I would add finally that there are no references cited when this problem is described toward the beginning of the MS, and this doesn't leave the reader who is unfamiliar with the concepts with a great deal of confidence in the initial motivation for the study.

*Editorial Note: In their review of the first version of this manuscript, reviewer 3 added their minor comments to the manuscript file. These comments, excluding minor textual revisions, have been copied into this Peer Review File.

Line 27: Redundancy here

Line 29: This opening statement could be ironed out. I've given it a go!

Line 30: Conjunction used before the ideas conjoined! Please check for further instances.

Line 34: Is the data volatile?!

Line 43: Do you mean the timing of such emissions?

Line 45: I don't think this logically follows the above as written. I know what you mean, but this statement needs a linking sentence of some sort to be supported.

Line 56: I think you're better off separating these two thoughts into two sentences.

Line 58: I'd prefer to see 'properly' not used. It implies others are not doing their jobs correctly, and I think your points are better couched in a less bullish manner. Interpretations are what they are, and are in most cases based upon the prevailing paradigms. Those of the past are not 'improper'; they're just potentially outdated by your present work! Try "more accurately".

Line 63: A Nature Comms audience needs this help I think.

Line 70: I don't think you need to re-introduce all of this. Leave it for the conclusions!

Line 75: Please separate into two sentences. As this stands it is borderline illogical – how can they be studied previously if you've sampled them afterwards?

Line 75: I'd prefer to see a simple sentence in here describing on what basis this premise is supported. Referencing out what is a central pillar in the paper's development is not going to help really engage the reader.

Line 77: What does this mean?

Line 80: I'd like to see a little critical appraisal of why this should be accepted as fact. Or, even better, couch it in terms of a question you're addressing in this paper. Without this you're describing something that is less innovative, and more incremental, than this paper deserves.

Line 81: Specify please!

Line 82: No, what you mean is element concentration maps.

Line 84: Focussing on variety doesn't make sense – maybe try 'capture' and thus gauge the variability?

Line 86: This must be a new paragraph

Line 87: Kernel density estimate (see John MacLennans' work) should be applied to give the reader confidence.

Line 87: Are these really rims? I suspect you mean the zone immediately in-board of the final quench rim, and this really needs to be stipulated either way.

Line 89: Saying e.g. implies that reverse zoning could be something else as well.

Line 93: I struggle to accept this as written. Firstly it really does seem like accepting the conclusion in order to support the premise, and secondly, there is no quantitative treatment of the data to support the claim. It's fine to just say "zoned crystals of these kinds have been reported in previous eruptions, although a statistical treatment to assess how representative those are, or how they may have changed through..."

Line 95: Again, please quantify this.

Line 100: Needs a REF I feel.

Line 101: It'd be nice to see some melt or whole rock data to support this hybridisation scenario. If you rationalise this with degrees of mixing, the onus is to demonstrate this is reflected at a scale larger than a crystal.

Line 103: Why?

Line 103: Morphology twice?

Line 107: If you've got 2D data then this is probably best rationalised as a sectioning effect. A tube in one orientation is round... Saying sphere implies you have 3D data...

Line 111: Why do these observations go together?

Line 113: This does not make sense as written. An inclusion is only an inclusion when it is closed?!

Line 119: Why is this a backslash here and a hyphen elsewhere?

Line 121: A little more formal language is probably needed. Returned?

Line 121: This is quite confusing, please re-phrase, or step out.

Line 122: Plus and minus? Please indicate the error envelope.

Line 123: Is this not better stated in the discussion?

Line 129: Well, in time no, but in equilibrium space maybe?

Line 149: Mmm, try to let the reader feel this rather than saying it is so... I think it is interesting, and I don't mind it being signposted as such per se, but experience shows this does rankle some readers!

Line 151: Again, this is up to the reader to decide. Just leave out the word significant and the sentence is better for it.

Line 157: Separate into two sentences. Your data and interpretation neither needs supporting by, or validates, this other example. It's good for comparison, but here it is used as a crutch when it's walking along fine!

Line 162: Still on the basis of 22 crystals.... You might want to ease back a little on this, or at least put the observation before the interpretation.

Line 167: So we're talking about the same pulse? The same physical material?

Line 170: I really think this needs to be higher, in the more result-oriented paragraphs. I like how the logic goes, but you need to talk about the crystal histories first, and then talk about the magma evolution. Mixing them together (no pun intended) like this doesn't make for as compelling reading as the study deserves.

Line 172: This is still a leap of logic that needs explaining.

Line 174: I'd really discourage the use of the word 'overall'. It's very weak, and implies an inconsistent and only loosely-held-together logic chain that you are unable or unwilling to guide the reader through... It suggests the reader should believe you at face value...

Line 178: By now the reader could be quite annoyed...

Line 166: All very rushed at the end without solid, logical, justification!

Line 177: I don't see why this needs to be stated; 70 degrees is an inherent part of the instrumentation.

Line 178: Firstly, 7 is the minimum; it can't be both 7 and 9. Secondly, why was the index rate so very poor? How were they mounted and polished? In my experience experimenting with polishing, you can get the wrong results with such a poor index rate. When the same crystals are given a good colloidal silica polish (as opposed to a 1 or ¼ um diamond for instance) the index rate is excellent (nearing 99%), and the results are considerably different from the several points indexed at the coarser polish. You either must demonstrate why confidence is assigned to the data by giving qualitative comparison to the crystal morphology, or re-polish and re-analyse. It could be that you performed a 'point and shoot' method of data gathering, as opposed to programming a map. I would recommend programming a map; this gives you a much more meaningful dataset and you can assess the index rate with meaningful statistics. If they are maps, I highly suspect the polish isn't good enough.

Line 180: This could be rephrased please.

Line 181: Does this imply the maps using WDS are less quantitative? Or, were the maps actually EDS?

Line 183: I think you should talk in terms of spot size. Beam diameter is not as meaningful, and is not equivalent to spot size.

Line 186: Not sure this suggestion is much better, but please re-word somehow.

Line 187: I don't understand how you could have made a decision on step sizes before you had the data...?

Line 190: This isn't always the case in the figures you show...

Line 193: I think you should specify which ones, and what difference it made. Were they all from one group? Also briefly describe the advantages and limitations of the MAN method. This can be done in two sentences, and would give the reader a much better feeling of confidence!

Line 196: This isn't quite correct. It's not the % difference, but the shape. You might want to justify why you say 'minimum' and include a brief discussion about the influence of growth on shape and calculated timescales.

Line 198: Did you model other elements? If not, don't say e.g.

Line 198: How good were these matches?

Line 202: I don't think 'account' is the word you're searching for. Correct is probably better.

Line 203: ??? Diffusion speed is highly sensitive to ambient temperature. Temperature does not change the diffusivity of a material.

Line 206: Why?

Line 209: Why triple the temperature error compared to the plagioclase thermometry results?

Line 212: This is confusing. Please specify clearly in what sense are you using 'initial' and 'boundary'. This is the second instance of this strange phrase and it is not clear from the grammar how you are distinguishing between the two, or if you are combining them, or what 'both' refers to.

Line 214: Reference needed here.

Line 214: Reference here and please qualify 'usually' with counter example perhaps/

Line 220: Please re-phrase

Line 223: Infinite?

Line 226: ?? I think you've lost most readers now. This is an important description and the paper would be better received if this was re-written.

Line 229: This really needs to be stepped out.

Line 232: This is really shutting the reader out, please include these however briefly.

Line 235: This crucial section reads like a series of excuses as to why the authors don't want to discuss anything here with any real rigour. Just summarise each paper in a paragraph each; you have space!

Line 244: This doesn't really tell us anything?

Figure 1 Legend: The third olivine profile is taken from a corner? What does the profile opposite (on a fairly nice straight crystal edge) suggest?

Figure 1 Legend: Is that not a contradiction then?

Figure 2 Legend: What is CSD, what is OSD?

Figure 2 Legend: Some statistical assessment is needed to help convince the reader. As currently presented this is not particularly strong; try randomly selecting portions of the data and see how this holds. Maybe just point out that the 100-ish day 'population' are all shallow?

Figure 3 Legend: I'd prefer to see this as a line on the figure; a gap is never as good as showing exactly where data were not collected! How can one really be sure that a gap means no data, or no signal? Without reading the figure caption how this figure is presented is not how it first comes across. The dates are hard to read. Just use a field to indicate which year you're in as time goes on...

Figure 3 Legend: Consider revising.

Figure 3 Legend: But they don't change depth? Why are they complex then?

Response to Reviewer 1 – Dr. Claire Bucholz

NB: for all line numbers refer to the final version of the track changes document.

1) The uncertainties associated with the modeling are not discussed in sufficient detail.

(pg 7: line 154-171)

We have now included a lot more detail directly in the main text and also in the methods.

-Temperature: Your estimates need to be discussed in more detail as diffusivities are so strongly controlled by this variable. Where do the $\pm 30^\circ\text{C}$ uncertainties on the temperature estimates come from?

(pg 7: line 155-163)

The $\pm 30^\circ\text{C}$ covers the range of observed values in the olivine-matrix glass geothermometer, and also the plagioclase-melt thermometer. Moreover, this range is typically of many geothermometers as quoted by the authors (e.g, Putirka).

Are the plagioclase and liquid compositions used in the plagioclase liquid given somewhere (from coexisting plagioclase phenocrysts in the scoria(?) and whole rock or glass analyses?)

(pg 7: line 156-163)

The plagioclase-melt temperatures are based on plagioclase phenocryst and matrix glass data; these are reported in detail in Ruth et al. (2016). More over these temperatures are consistent with those obtained by olivine-melt thermometry, with data from Bouvet de Maisonneuve et al., 2012 . Because the compositions are discussed in greater detail in the stated papers, we do not include them here.

(Also, the reference given for the $1150^\circ\text{C}\pm 10^\circ$ (reference #12) does not seem to be appropriate.)

(pg 7: line 158)

Yes reference 12 was a typo. The reference list has been updated.

Why do you use the anhydrous version of this thermometer when you know that the magmas that these olivines crystallized from are hydrous?

(pg 7: line 158)

An increase in H_2O content to 0.5 wt% lowers the magmatic T to average of 1123°C , at 1 wt% the average T is 1118°C . These temperatures are within the uncertainty discussed in Putirka, $\pm 25\text{-}30^\circ\text{C}$. We have stated these temperatures are based on anhydrous magmas. If we use the hydrous temperature for the crystals that have the highest water contents (e.g. the deeper ones) the time scales would become longer, but the conclusions would be the same.

- Have you tried multiple diffusion profile on a single olivine grain? Perhaps along different orientations? Do they give the same timescales?

(pg 7: line 167-171)

We did collect multiple traverses on two different crystals. We have now added in appendix some data and calculations that show that the diffusion distance is proportional to diffusion anisotropy, and thus they do give the same time scales. Also, we did model minor elements diffusion as an additional test. Timescales from the majority of the trace elements overlap with the Fo timescales (see Fig 4).

- When setting up your initial conditions, you choose two endmember scenarios (step function versus homogeneous). For the step function profiles, this (if I understand correctly) is assumed to replicate the scenario of magma mixing, whereby a new rim of olivine grows around the core with a different Fo content. However, this growth must certainly take some time. How does simultaneous growth and diffusion affect your estimated timescales? It would likely shorten

them, I would think. I understand that the model you use may not incorporate growth into the calculations, but this is something that you should at least discuss qualitatively.

(pg 7: line 171-173)

In our study and many previous ones (e.g. Costa & Dungan, 2005; Marti et al., 2013; Albert et al., 2016) we find that growth is much faster than diffusion, as could be expected in open system processes with large changes of composition or intensive variables. But we have actually tested this possibility by comparing the time scales obtained between Fe-Mg, Mn, Ni, and Ca and found that they are the same within error and thus shows that the profiles shapes are dominated by diffusion and not growth. Thus a model of growth + diffusion should not affect our results in a significant manner (e.g., still within the uncertainty of the determined time scales). A similar argument was made above about the zoning of the crystals being proportional to the crystallographic orientation and thus diffusion anisotropy. And the reviewer is correct that if the zoning would be growth dominated our time scales would be maximum values as discussed in quite a few papers (e.g. Costa et al., 2008).

-The melt inclusion profiles don't always plateau towards the cores (e.g., the group C olivine that you show in Figure 1, bul3, plate1, etc.). In these cases, how do you know what to choose as your initial conditions?

(pg 7: line 164-165)

We chose our initial conditions based on the observed zoning in the electron microprobe traverse. For two profile traverses (bul3 and plate1) we assumed the last measured composition in the traverse was the core composition, and used a homogenous initial profile. Although we did not get the true plateau core, the composition we used is similar to other core compositions in crystals which we did reach the true plateaus. Moreover, if we modeled the profile with a higher composition (>Fo82-83), the timescale would be longer, and thus our inference for very early intrusions would still be valid.

2) The coupling of timescales from olivine diffusion profiles and the melt inclusion pressure estimates may not be recording the same story.

As I'm sure all the authors are aware, melt inclusion H₂O and CO₂ contents can be affected by post-entrapment diffusive re-equilibration of H⁺. With diffusive loss of H⁺ due to any degassing-induced concentration gradient in H₂O during ascent through the crust or eruption, water concentrations in the melt inclusions will decrease and CO₂ may be exsolved into a vapor bubble. If these processes aren't considered, then you will only be getting a minimum pressure estimate for depths of entrapment. The trends in CO₂-H₂O space produced by this process are identical to that of degassing. It also doesn't only happen during eruption, it can easily happen as the melt is migrating from depth in the crust to upper level reservoirs.

(pg 8: line 187-193)

We agree with reviewer about these issues, which for these inclusions were discussed in Ruth et al (2016) and Bouvet de Maisonneuve et al. (2012). We have now added some details by noting that we studied quickly quenched tephra samples that show limited post-entrapment crystallization and thus H loss. We also note that we recognize in the text that in any case we are only obtaining minimum entrapment depths.

I haven't read the previous work done on the melt inclusion in these olivines, and perhaps the authors have dealt with this in a prior manuscript, but it should be discussed here and factored into their final story about the structure of the magmatic system below Llaima.

For example, for the four melt inclusions that preserve higher volatile contents (in particular CO₂), how do their radii compare to the other melt inclusions in the study? Larger melt inclusions will re-equilibrate more slowly due to their larger volume:surface area ratio.

Could this be an influencing factor? Are their vapor bubbles present in these melt inclusions?

They, of course can be formed through processes other than just diffusive loss of H⁺, but it would be good to note.

(pg 8: line 192)

As noted above we have now summarized the work completed in Ruth et al. (2016), which discusses, in more detail, the post entrapment modification issues. We acknowledge that melt inclusion size may have an effect on post entrapment modification. We have checked and melt inclusions in crystals with the longest timescales have sizes that range from 20 to 238 μm . The “oldest” melt inclusion is 1375 days old and is 131 μm , whereas the next oldest is 1167 days old and the melt inclusion is only 20 μm . Based on the four melt inclusions, we do not see any correlation with size (and volatile content) and crystal diffusion timescale as could be expected if the reequilibration processes proposed by the reviewer would be a major player. Some of the inclusions have vapor bubbles and we have now included information about the size and presence of bubbles in supplemental table S1. However, as noted in Ruth et al (2016) it was not possible to correct for the CO₂ partitioning to the bubble and thus we explicitly say that our volatile contents are minima.

- Do melt inclusions from the same olivines preserve similar volatile contents and thus calculated pressures? Or are they disparate?

Some melt inclusions have similar volatile contents. Br5-1 has a wider range, which may reflect that these melt inclusions are actually open embayments, if you will, and are therefore able to ‘communicate’ with the exterior melt. The volatile data are given in Table 2 in the main text and again in supplemental table S1. We note that we have also highlighted in the figures the variety of volatiles of different MI trapped in the same crystal.

- Do all your profiles traverse regions populated by melt inclusions, suggesting that the calculated diffusive timescales are for regions of the olivine that grew during melt inclusion entrapment? In figure 2A the profile stops short of the core and only covers ~1/2 the imaged melt inclusion.

No, not all traverse regions with melt inclusions. But we note that we chose the traverse locations according to the diffusion gradients and not to the presence of MI. The MI formation and that of the zoning happened at different times.

I hope this doesn’t come across as a shameless troll to get my papers concerning this topic cited, but I do think it needs to be given some consideration when you are taking two potentially independent systems (volatile contents in melt inclusions as pressure estimates for entrapment and Fe/Mg profiles in olivine to estimate diffusive relaxation timescales).

(pg 8: line 188)

Ruth et al. (2016) discusses post-entrapment modifications of the melt inclusions. Bucholz et al. (2013) is cited within that paper. Since we summarize much of that work, we cite Bucholz et al. (2013) as well in this paper.

3) There are only four olivine crystal recording long residence times. These particular samples deserve extra attention.

As inferred from your Table, the high pressure, long residence times melt inclusions are: bul3, plate1, br1-2, and B22. The data for the first three is presented in your supplementary table, but I could find the diffusion profile for B22. Bul3 and plate1 are distinct from all the other profiles presented. They don’t have a clear plateau towards the core, demonstrate significant scatter, and don’t really even preserve a classic diffusion profile.

How do you know that they are recording diffusive processes? Why are these olivine profiles so different and how does that impact the ultimate story you construct?

(pg 13: 333-334)

We have insured that all the compositional traverses are now available in the supplemental material table S2, in addition to Figure 2.

It is unclear to us what the reviewer means by a ‘classic diffusion profile’. We suspect she is referring to a S-shaped profile. But the olivines show a decreasing towards the rim Fo content that can also be due to diffusion and it is actually seen in other systems involving mixing (a population of olivines shows normal zoning and the other reverse zoning; e.g. Kahl et al., 2015). The normal zoning and the high Fo of the plateau reflects re-equilibration (or growth followed by re-equilibration) of the deepest mafic magma, so we think it makes sense that deepest crystals are also the most mafic and with the longest times as this is one of the important aspects of our story. As noted above in an answer to a similar question pertaining to these crystals we believe the zoning reflects diffusion processes based on multiple elements giving the same timescale within uncertainty. Additionally, normally zoned crystals have been observed in previous work in Llaima so that they are not unusual (Bouvet de Maisonneuve et al., 2016), so are not unusual at Llaima. Finally our interpretation would not change drastically if we remove bul3 and plate1 because it would still be supported by two other crystals.

We are unclear what the reviewer means about the scatter in the data. If referring to compositional noise in the dataset, this is accounted for in the DIPRA model

How do you take into account the uncertainties associated with the scatter of the data?
Incorporated into the +/- values calculated in the DIPRA model. These include the uncertainty in the temperature, but also the observed compositional range see Girona and Costa, 2013.

IN LINE COMMENTS – Reviewer 1

Reviewer 1 – Dr. Claire Bucholz

Line 25-26: are you referring to similar melt inclusion compositions here?

(pg 2: line 33-44)

Similar mineral chemistry. We have completely revised the Introduction paragraph/Abstract and the sentence has been removed.

Line 100-102: I think this is a really cool observation between melt inclusion clustering and P zoning(!), but again how does subsequent slow growth affect your diffusion modeling?

(pg 5: line 112-116 for P discussion with respect to melt inclusions, and pg 7: lines 168-173 for growth and diffusion)

We assessed if growth is a dominant process recorded by looking at the timescales from the minor elements (Mn, Ni, and Ca) and checking the diffusion anisotropy tests (e.g. Costa & Dunagn, 2005) as noted in an answer of the comment above.

Line 114: No figure 2a was provided with submitted manuscript.

(pg 8: line 174-184)

This section has been revised significantly and references new figure 4 rather than old figure 2.

Line 143-145: I found this sentence confusing. I'm not sure what "continuous summit degassing" means.

(pg 10: lines 228-230)

Open vent systems exhibit prolonged/persistent degassing from a summit vent, or vents that may last decades or even centuries. We have modified the sentence for clarity

Line 203: "diffusion coefficient" should be "diffusivity"

(pg 7: lines 154-163)

We have moved this discussion to the main text and revised for clarity. A brief summary is in the methods section (pg 15: lines 336-350).

Line 209-210: What were the XANES measurements of? Fe³⁺/Fe^T in melt inclusion glasses, I assume. Should also be ΔNNO .

(pg 15: lines 348-350)

Yes, they were Fe³⁺/Fe^T in melt inclusion glasses which vary from 0.163 to 0.194, which equates to ΔNNO -0.2. We have reworded the sentence for clarity and include the proper citation.

Line 233: What densities do you use in your pressure calculations.

(pg 8: lines 195-196)

We used 2200 kg m⁻³. We also move this to the main text.

Table 1: Can you give H₂O and CO₂ contents here too so that the reader can assess on their own the pressure estimates? It will also help the reader in reference to Figure 2. Also, no need to put a comma after "output" in the Table title.

(New Table 2, supplemental table S1)

We have added the H₂O and CO₂ contents to a new table which provides the melt inclusion size, approximate location. We have revised the table caption.

Figure 1: Why do your plane polarized light images look so scuzzy? Did you dissolve away the scoria clinging to the olivines before analyzing these? These images aren't super helpful as the olivine and melt inclusions are so obfuscated. In the caption, please clarify the discussion about

“open” versus “closed” melt inclusions. Do you mean there was diffusive re-equilibration of certain elements between the melt inclusions and the external melt through the host olivine. You refer to “patchy zoning”, but for what elements? What’s the actual pattern consist of? (This obviously doesn’t all need to be in the figure caption, but discussed somewhere if you are going to bring it up).

The olivine crystals were originally mounted in indium and we then removed them for the transmitted light images. They are now mounted in crystalbond and may not be completely clean. We have removed them from the main figure as we agreed that they were not so clear. We have revised the caption to reflect the reviewer’s suggestions. We do believe that given the Fe-Mg zoning around some inclusions they were in communication with the outside melt and thus “open” and this is what we meant by patchy zoning. However, we realize now that this distinction and wording might be confusing to the reader and we have removed it.

Figure 2: You volatile solubility model reference should be to 29, not 28. Also, please define abbreviations of CSD and OSD as closed versus open system degassing.

(new figure 5)

CSD and OSD are closed and open system degassing respectively. We have revised the figure to explicitly state this. We have also fixed the citation in the caption.

Response to Reviewer #2 - Dr. Phillip Ruprecht

The manuscript “Melt inclusion residence times reveal the magma dynamics of open vent volcanoes” by Ruth et al. is combining melt inclusion (MI) volatile chemistry with host-olivine elemental zoning modeling to connect the temporal record with degassing at Llaima volcano, Chile. The approach is truly novel and potentially very interesting. However, I was not convinced that their conceptual model is properly taking into account what we know about melt inclusions.

Most importantly the authors claim that MI studies typically assume that MIs record the syn-eruptive degassing history of a volcano and therefore build an argument to provide an alternative model. This is unfortunately a strawman argument as it is well agreed upon in the community that MIs record the minimum pressure (depth) of last equilibration that is rarely equated with syn-eruptive ascent *sensu stricto*. Instead, the MI record just provides a general degassing pathway for the ascending magmas in the respective magmatic system. Underlying that idea is that different magma batches/parcels that arrive at upper crustal levels will experience similar degassing behavior, so they do record the degassing path associated with eruption, but only in the general sense. The fact that there are meaningful trends in CO₂-H₂O plots is commonly explained by magmas stalling on their way to the surface and thus having sufficient time to equilibrate to the new CO₂-H₂O equilibrium conditions. Thus,

I think instead of trying to build a somewhat misguided argument against syn-eruptive degassing records, the data provides a very interesting case that shows that crystals of different residence age and storage region are assembled prior to eruption and this done likely over short timescales.

(pg 3: lines 54-59 – introduction section; pg 10: lines 231-244 – discussion section)

In our original submission we did not include enough references to make our point clear about how different authors interpret MI data. We now present in the introduction and the discussion the “syn-eruptive” entrapment and interpretation proposed by some authors and also the alternative interpretation alluded by the reviewer. We use this as a starting point to highlight the need to clarify the meaning/interpretation of MI data by studying the crystals that host the inclusions.

Secondly, despite the short format of this journal, I think the authors need to provide significantly more information for the reader to assess the manuscript. I am providing several key datasets to improve in the line-by-line comments, but I would like to summarize here the following aspects: 1) The samples, crystals, and MIs are insufficiently described. Where are samples coming from? Are crystals from one or more samples? How big are the MIs and the crystals (see below). 2) this is elemental diffusion modeling study on microprobe data, yet no microprobe data is presented. 3) Melt inclusion information may have been reported elsewhere, but in the current form it is impossible to follow the manuscript without going back to the older studies. 4) A short petrography discussion, at the very least in the supplement.

We have now added the following as keyed to the reviewer request: 1) pg 4: lines 84-90 – sample collection; pg 12: lines 282-285 – sample preparation. 2) We have now a new figure with electron microprobe traverse plus all the data in the supplemental table S2. 3) The melt inclusions data were reported and extensively discussed in previous work (Ruth et al., 2016; Bouvet de Maisonneuve et al., 2012). We have now added melt inclusion size, and H₂O and CO₂ contents in supplemental table S1. 4) We have added a short petrography section (pg 4: lines 174-282) to describe in better detail the samples and added a new figure that shows many profiles with electron microprobe data (new figure 2).

Thirdly, there are several issues regarding referencing and clarity of figure and captions. Especially when comparing table 1 with figures 2 and 3, I was very confused and was not able to resolve the issues (see below).

We have revised the figures and added text to the captions to better clarify and explain the data.

IN LINE COMMENTS - Reviewer 2

27-30: Gap in arguments between these two sentences. The causality of how residence times and a range of residence times are related to MI is not developed

(pg 2: line 33-44)

We have completely revised the Introduction paragraph/Abstract and the sentence has been removed.

54: How is open vent system defined? What makes Llaima an open vent system over other frequently erupting volcanoes?

(pg 10: lines 237-240)

Open vents are volcanoes that have a permanent volcanic plume even if they are not erupting. Many frequently active volcanoes are also open vents, but not all (e.g. Kelut is not). We notice however that several reviewers were confused by our use of open vent volcanoes so we decided to remove the classification of Llaima or other into open vent.

58: The authors suggest that MI papers conventionally discuss MI data as a syn-eruptive record. They do not provide any citations and I disagree with this being true. MIs are commonly interpreted as a suite that provides a context about magmas of similar composition, but I rarely see that people link them specifically in time. You refer to this again (114), but this seems like a strawman argument to me.

(pg 3: lines 54-59 – introduction section)

We agree that some authors view melt inclusions as a suite of data, but others do not. There are several widely cited papers that assume melt inclusions form just before and during eruptions, thus recording eruptive processes (e.g. Shinohara et al., 2003; Metrich et al., 2004; Blundy and Cashman, 2005; Spillaert et al., 2006; Metrich et al., 2010). Now we have modified the text to clarify the two main interpretations that exist about MI data and added the necessary references accordingly.

65: Wrong citation

(pg 3-4: lines 69-80)

This should be Ruth and Calder, 2014. However, we have revised this section extensively and removed the original sentence.

72: Can you be more specific? How many crystals did you look at? How many did you ultimately analyze by quantitative means and how many did you choose to model? Also, how many samples did your olivines come from? Where do these samples come from (both locality as well as level within the deposit)? Are these samples covering different episodes of the eruption? There are only 14 crystals shown (images in Supplement), but there are 22 crystals mentioned both in the table in ll. 78.

(pg 4: lines 84-90)

Our overall melt inclusion dataset from previous work (Ruth et al., 2016 and Bouvet de Maisonneuve et al., 2012) had over 200 crystals, from which we selected 22 crystals for modelling. There were two explosive phases of the 2008-2009 eruption: 1 Jan 2008 and April 2009. We collected samples in Jan 2009 and Jan 2011. Since we collected the largest scoria possible, we assume these originated from the initial 1 Jan 2008 phase as it was the most intense and produced a plume with the most extensive dispersal.

We have added more information about sample sourcing at the beginning of the results section. There was a mistake made when producing the image file. All 22 crystals are now present in figures S1-S3.

105: It is said that 27 crystal traverses were modeled (that is also the number of olivine traverses in table 1), but I count 28 data points in figure 2 and 3 (not counting the open symbols). Also, I

cannot reconcile the data points between figure 2 and 3: e.g. there are four data points that are deeper than all others (in figure 3). These points are showing the longest residence times: in figure 2, 2 point are (very) deep and have long residence times. There are two other points with long residence times (red), but they are at the same pressure as or lower pressure than three shorter residence time datapoints. It may help to add the CO₂/H₂O data to Table 1 so that one can better track which MIs belong to which olivine.

(pg 7: lines 164-166, new figure 5 and 6, table 2)

We did model 27 traverses. However, several of these crystals had more than 1 melt inclusion. So for figure 2 (new figure 5 and 6), we included every melt inclusion with H₂O and CO₂. Yes, we see the confusion concerning the isobars. We constructed the isobars and degassing models using the average bulk chemistry of the whole rocks (after Ruth et al., 2016). This reveals the limitations of the degassing models, since they are constructed with one composition only, whereas melt inclusions trap melts of a wide range of compositions. We rechecked the pressure calculation with the Papale model and the values we report are correct. We have added the volatile content for these specific melt inclusions to the supplemental material. We also include the H₂O and CO₂ content to table 2.

112: The authors argue that there are no discernible peaks, however, the complex crystals for which a second timescales for near-rim zoning was modeled clearly show a cluster of the shortest timescales for the near-rim zoning (not generally surprising). They make up more than 50% of the < 100 day timescales, and A21 with 37 days is the only non-complex crystal among the population of really short (< 50 days) timescales. This is not addressed at all.

(pg 8: lines 176-184)

We have added a statement explaining the short timescales for the complex crystals. We have also added a histogram with 30 day bins and note there is not distinct peak. Rather, the data are distributed over a long range of time.

116: This is a very important statement. Yes, the data shows that one cannot treat these as a single crystal population. But people do not do that anyways for MIs! They treat them (if at all) as single MI populations, which may seem like a semantic thing, but it is not. I would argue most MI studies use MI populations as a way to capture the general behavior and degassing path of magmas involved in an eruption.

(pg 10: lines 231-244)

We have discussed this argument above. We agree that many authors view melt inclusions as a suite of data. However, there are several widely cited papers that assume melt inclusions form just before and during eruptions, thus recording eruptive processes (e.g. Shinohara et al., 2003; Metrich et al., 2004; Blundy and Cashman, 2005; Spillaert et al., 2006; Metrich et al., 2010). Yes the crystals and melt inclusions were sourced from more than one sample, but they did come from the same eruption.

Few studies claim that the MI population is limited to a single crystal population that experience the various conditions in the (very) narrow definition that is applied here, ie the crystals were spatially and temporally together. Even the term crystal population is commonly used much more broadly.

The reviewer is correct, to our knowledge, no one has published the spatial and temporal relationship between crystals.

MI studies rarely claim that the MIs all represent the same parcel of magma (which is implied here for those studies). Moreover, if you make that claim, it is somewhat a pre-determined outcome in this study given that you look at olivines (and MIs) from more than one sample, at least that is what I got from the provided information!

We do not think that melt inclusions represent one parcel of magma and nowhere is this stated explicitly in the text. We argue that because melt inclusions form at a minimum of 10s to 1000s of days before an eruption, their chemistries cannot be a record of syn-eruptive processes. Moreover, if others are treating melt inclusions as a suite, then it seems to us an implicit assumption that has never been tested.

125 How much post-entrapment modification occurred? As described above there is basically no information about the MIs! Sizes, compositions, in equilibrium with host?, location within the grains, presence of exsolution bubble... Many studies in the last years have shown that this information is important when estimating pressure via CO₂ (and H₂O).

(pg 17: lines 396-410, Supplemental table S1)

We addressed this question in answer to R#1. In short, we studied quickly quenched samples, and melt inclusions with more than 7-10% post entrapment modification were removed from the dataset (see Ruth et al., 2016; Bouvet de Maisonneuve et al., 2012). We discuss in the methods section more details about melt inclusion selection and post-entrapment modification. A more detailed discussion is beyond the scope of this paper, given that these melt inclusions are extensively discussed in Ruth et al. (2016) and Bouvet de Maisonneuve et al. (2012). We have provided the appropriate data about the melt inclusions in the supplemental material and modified the text accordingly.

132: “This interpretation ...”. It is not clear what “interpretation” is referred to here. But I assume the authors claim that one cannot relate the MIs in a CO₂-H₂O plot to open or closed-system behavior. In any case, the statement that “the interpretation” “implies” that the MIs formed during ascent or immediately prior to eruption is incorrect! The entrapment pressure that is recorded is the pressure of last equilibration. Of course the crystal + MI can be stored for a long time at these conditions.

(pg 10: lines 228-230)

Our data show quantitatively that MIs form/re-equilibrated much earlier than eruptions, and therefore cannot be recording syn-eruptive processes. We agree that degassing curves could be used to describe generalized, time-averaged behavior; we state this explicitly for Llaima. Moreover, we have now recognized more explicitly that the pressures we calculate can reflect the original formation or re-equilibration, but our point is that if they were re-equilibrated it was not syn-eruptively because of the longer times of the Fe-Mg.

There is no question whether the ascent and duration of the assembly of all the MIs was short (otherwise they would re-equilibrate as is also explained by Lloyd et al. 2013, who they cite). So what it really means is that the eruption is composed/assembled of a diverse range of “magmas” that are brought together from zones as deep as 9 km as well as shallow as ~ 2 km, i.e. the eruption was not just tapping the shallow part of the magma system. And also what it means is that the upper part of the system has few (or no) crystals that get stored there for a long time, while the deeper storage system is not a direct “path” for magmas, but that they commonly are stored there and later remobilized during the eruption.

(pg 11: lines 249-252)

We agree and have revised the text accordingly.

138 again, this is a strawman argument, the studies rarely state syn-eruptive formation of melt inclusions. Instead they state depth of last equilibration (that by definition implies longer storage).

(pg 10: lines 231-244)

We agree that melt inclusions record the last depth of equilibration; our data quantitatively proves this. However, several studies explicitly state that melt inclusions record syn-eruptive processes. We show that melt inclusions cannot record syn-eruptive processes because the times we obtain from Fe-Mg zoning. If there was re-equilibration it was pre-eruptive and not syn or post-eruptive.

151: Here the authors do exactly the distorted interpretation they earlier criticized that this is one batch of magma experiencing a specific history, while the olivines likely reflect different storage regimes that were assembled late, just prior to eruption.

(pg 11: lines 249-252)

We do not believe that it is one magma batch ascending and experiencing one history. It is an assumption that magma recharge is batched based, rather than constant percolation. Rather our hypothesis is that melt is percolating and entraining crystals during progressive ascent. We have revised the section for clarity.

155: Crystal dissolution. This is the first time the authors mention that the crystals get consumed. Seems, this is important to elaborate on a little more. I am missing at least a short petrography section (at the very least in the supplement).

(pg 6: lines 142-145)

We have added more about the textural interpretations in the main text.

172: I guess here one should reference Table S1. However, here it would be also important to add more information about the crystals (size of the crystal, the inclusions, the distance of the inclusions to the rim, ...)

(supplemental table S1)

We added a table that includes the approximate size and location of the melt inclusions within the crystals.

184: There is not table with actual microprobe data and no information about uncertainties and detection limits. Were reference materials analyzed together with the unknowns? I guess some of that information is in other papers, but there is no reference to any other publication.

(supplemental table S2)

We have now included the full probe datasets for the diffusion traverses.

195: How do zonings in Ca, Ni and potentially other minor elements compare to the major elements? What timescales do they suggest?

(pg 7: lines 171-173; new figure 3)

Where possible, we modeled the Ni, Mn, and Ca diffusion. For the majority of samples, the timescales overlap within uncertainty. Note for those that do not overlap, we speculate it may be related to the higher noise in the trace element profile compositional data.

202ff: I am surprised there is no estimate of temperature using the olivines and their melt inclusions, as Ol-Melt is among the best known/tested thermometers! How about the temperature estimated from olivine-matrix melt or olivine/whole rock. All these estimates provide a range of temperatures that probably need to be considered and if the possible temperature range were to increase then that would probably be more robust.

(pg 7: lines 154-163)

Yes, we agree that the olivine-melt thermometer is very well constrained. However, entrapment temperatures are not the same as eruption temperatures, or magma mixing temperatures which is more likely to be related to the observed zoning. Bouvet de Maisonneuve et al., provide olivine+matrix glass temperatures and they range from 1114 to 1198 °C, averaging 1140 °C. These overlap with the plagioclase-melt thermometry discussed in the text. We added language to the text to discuss the overlap of the plg-melt thermometer with respect to the olivine-matrix glass thermometry.

209: I am not following how the oxygen fugacity was determined. It is said that it was obtained by XANES, but the reference that is used is a general review paper on volatiles in magmas that does not address Llama magmatism nor does it talk about XANES as a technique to get after fO₂. Is

reference 5 the correct one?
(pg 15: lines 348-350)

It was the wrong citation. It should be for Ruth et al. (2016) which discusses our XANES data at length; the citation has been fixed.

230: There is no detail on how these volatile concentrations were determined. Of course there are references to Ruth et al. and Bouvet et al., but at least a brief summary of the methodologies should be added. Is this SIMS, FTIR? Where exsolution bubbles taken into account?

(pg 16: lines 372-410)

We have provided a more detailed summary of our volatile content determinations and our handling of the vapor bubble issue.

Figure 2: The caption references the wrong citation, exchange 28 for 29. I would not refer to “samples” but “olivine traverses” or “olivines”, whatever is correct. In any case, “samples” is not helpful as it seems that there are several (tephra) samples that were used in this study, from which X olivines were separated and Y traverses were modelled. There is no explanation of CSD and OSD, presumably closed-system and open system-development.

(new figure 5)

CSD and OSD are closed and open system degassing respectively. We have revised the figure to explicitly state this. We have reworded and fixed the citation in the caption.

Figure 3: There is no explanation of what the open system and filled systems refer to. The legend in the upper left corner is not helping as it states just that. It does seem that the shift to the shallowest MI record coincides with the seismicity in June 2007. However, given the short seismic record (basically just 2007), it is difficult to assess how special the episode in June 2007 is as for a significant fraction of the MI datapoints there is no seismic record.

(new figure 6)

The open and closed refer to open and closed melt inclusions. We have removed this from the figure. Our colleagues at OVDAS inform us that from 2004-2007 the seismic activity was similar to that recorded from Jan-May 2007.

Response to Reviewer 3

1. Title implies plural, yet the study is on one volcano. There is no justification offered why this study should implicitly or explicitly tell us things about other systems.

Llaima is an example of open vent volcanoes and thus our results and interpretation may have implications for open vent volcanoes. Our novel approach of combining melt inclusion and time scale data can be applied to many other volcanoes. However, we have removed reference to open vent volcanoes because we argue that our approach applies to all melt inclusion studies.

2. The crystal diffusion timescales are not timescales of the melt inclusions as stated in the title. The textures clearly demonstrate that disequilibrium was imposed after the melt inclusions were enclosed in most cases. How long after is not addressed. Why wasn't a single crystal really taken apart and all the inclusions analysed to make some sort of assessment on how the 'stratigraphy' of the crystal relates to any 'overprinting'? There might be potential to see a liquid line of descent (e.g. in Al or Ca?) from core to rim, for instance, and then the Fe-Mg contents could be altered by partitioning with the surrounding crystal as it changes Fo...

(pg 3: lines 64-68)

The crystals are hosting the inclusions so they are, to a large degree, the same time scales. Their MI could have resided longer than the zoning in the crystal is recording and thus we have made this more explicit in the introduction by noting that these are minimum times. We have also now included "crystal" in the title so that the reader can explicitly see that the times were obtained from the crystals and not directly from the MIs. Yes, the crystal zoning occurred after melt inclusion formation. The diffusion timescales give a minimum "age" of the melt inclusions, based on superposition principles. With that in mind, we can then assess which melt inclusions are minimally temporally related.

We agree that an extensive stratigraphic study of the melt inclusions should be initiated. However, this is beyond the scope of this paper. We welcome the reviewer to do such a study of MI chemistry in detail. We do have multiple MI analyzed for their volatile contents as already reported in the manuscript.

3. It is simultaneously assumed that 1) the moment a melt inclusion was captured is the same moment the diffusion stopwatch 'starts', and 2) that many of the crystals were resident before magma-mixing. Only one or the other assumption could be held, or maybe neither! What mechanism are the authors suggesting is responsible for (1).

We do not assume that diffusion starts *immediately* after melt inclusion formation. We are using the zoning from the margins to the interior, which post-dates melt inclusion formation.

Are there data to support (2), or is this regarding resident magma as fact - if so why is this reasonable?

The crystal zoning shows that crystals need to be present before magma mixing. Similar textures and timescales have been reported for other eruption at Llaima (Bouvet de Maisonneuve et al., 2016) and for other volcanoes (Kahl et al. 2011, 2013, 2015).

If there really was a combination of the two, what evidence is there here to really discriminate between them? What assumptions need to be held for that interpretation to be internally consistent? Assigning groups on the basis of melt inclusion texture doesn't seem up to this task.

Some embayments/open inclusions are surrounded by zoned olivine; in this case, the zoning could be simultaneous to formation. This was also clarified in the answer to the comment above.

4. The (limited) discussion of the EBSD data is worrying. Were there a minimum of 7 ‘point and shoot’ style measurements taken per crystal? If so, how were they distributed and why? Why wasn’t mapping conducted? If it was, at what resolution were they gathered, and why was the indexing so very poor? While it could be reasonably assumed that these magmatic minerals grew free in a melt and thus should have single orientations, experience shows that if the indexing rate is poor the orientation data (even if they are self-consistent) are wrong.

(pg 13: lines 294-300)

We are puzzled by this comment; it is unclear to us why we need to determine the crystal axes of the same crystal multiple times. Since these are quickly quenched volcanic phenocrysts a single good determination of the crystallographic orientation is enough, so there is no need for mapping. In any case, we have at least 3 EBSD indexes per crystal, which is higher than any other study of volcanic rock using olivine that we know of. Moreover, in our experience the most difficult part of getting good crystallographic orientation and indexing is related to sample preparation and surface polishing. For the cases where axes were oriented parallel to perpendicular to the thin section plane we found that our visual estimation of the axes using the optical microscope and that provided by EBSD matched. We have expanded our methods section for clarity.

5. There is no description at all of sample preparation. This in itself is troubling, since rounding effects imposed, for example, by alumina powder polishing grits do seem to have a non-linear effect on signal received at detectors, and these can affect the profile shapes derived from spot analyses as well as maps (backscatter or element concentration). A quick check of the Z height on the EMP spot-analyses might help here.

We are at a little bit of a loss on this comment. It is well known that the quality of the polishing (e.g same topography) can affect the quantitative results but we are not aware that it has been documented in the literature or anywhere that the type of polishing material leads to effects that would be measurable at the precision we are working with. We used diamond polish and pads, and well established procedures to polish our samples. We do not have auto-focus on our probe and have to set a Z height for each analytical point. If our set height is drastically different from the actual height, it would be reflected in the totals. Our totals remain within the acceptable range (98-102 wt.%) for EPMA. We do not think our preparation techniques have affected our results.

6. There were two different analytical methods used to gather quantitative in-situ chemical data, yet no discussion of how they were verified against each other, and whether or not a statistical difference in results was observed. MAN backgrounds should produce the same results in much less time (although this of any other reason is not offered as supporting the choice made), but surely there are crystals that both approaches were employed upon to check? You have two populations of rims, and two methods of compositional data acquisition – please put the reader’s mind at ease by showing there is no correlation.

(pg 14: lines 319-322)

(supplemental table S2)

We have added a table with the EPMA data in the supplemental material, this includes whether or not the samples were analyzed using traditional EPMA techniques or the MAN background correction. We would like to reassure the reviewer that we had two population of rims with the traditional EPMA technique, which indicates the rims compositions are not related to analytical protocol.

We did not run duplicate analyses with traditional and MAN background approaches because the MAN technique has been shown continually to produce high-quality results (see Donovan and Tingle, 1996; Donovan et al., 2016). We did run a natural olivine standard as unknowns in both cases. Differences between the MAN and traditional approach ranges

from 3% to 10%. We added language to the text to explain better the justification for using the MAN approach.

7. Rims: is there really no high-Fe final quench rim of a um or two? This is a surprise and would be worth stating.

(pg 4: lines 93-95)

We do see fine scale Fe-quench rims. We have added text in the results section to indicate this.

What is the equilibrium melt for each of the crystals on eruption? Does this match the composition of the surrounding glass (or is that resin on those images).

We calculated the equilibrium melts using the MELTS algorithm and the composition of the olivine rim (excluding the quench) and we found different temperatures depending on the Fo rim compositions. Given that there are different rim compositions they can't all be in equilibrium with the matrix glass.

8. One of the three crystals shown displays the profile extending from a corner, which suggests the influence from these different directions could have affected the measured profile shape. How has this been dealt with?

The reviewer did not specify which crystal he/she is referring to. However, we think the reviewer is referring to crystal br5-1. Yes, we agree that the corner can affect the diffusion timescale due to diffusion from additional dimensions (Shea et al., 2015). To test this, we collected an additional profile produced with a calibrated BSE image (e.g. Martin et al., 2008; Cashman and Blundy, 2013; Fabbro, 2014). This profile was perpendicular to the crystal face and the modeled timescale was only 7 days shorter than the original traverse; this is within the uncertainty due to the temperature calculation. Therefore, we use the original modeled timescale in our interpretations.

9. Is there any scientific basis for assigning crystal groups on the distribution and number of melt inclusions from experimental studies? This seems to be quite arbitrary and doesn't itself hold any petrologic meaning, at least none that is described or discussed. Surely zonation 'style' holds more potential for grouping crystals into populations, for these at least might be argued to hold some meaning as to magmatic environments through time? What controls the number of melt inclusions in olivine?

The abundance and morphology of melt inclusions do provide information about processes (Faure et al., 2005) although it is still unclear what controls the number of MI that each crystal may trap, and int could be a kinetically controlled effect that also is affected by randomness of crystal nucleation. The relationship between the P maps and melt inclusion location provides information about growth (Welsch et al., 2013). We did look at zoning, but no discernable patterns were observed. Finally, there is a relationship between melt inclusion textures and the timescales, suggesting that our scheme has merit.

10. Statistical robustness is claimed, yet saying that (paraphrasing) 'the other eruptions have similar zoned crystals, this eruption has a variety of zoned crystals, therefore this is a representative dataset' does not hold tightly as an argument. There must be a demonstration of how this assessment was performed with some degree of specificity.

(pg 4: lines 102-106)

We added data from Bouvet de Maisonneuve et al. (2012) which shows the compositional range and distribution are the same for different eruptions at Llaima. This is shown in fig 2.

11. Magma-mixing is used to explain the crystal chemical and zonation diversity, yet there is no independent evidence, like whole rock compositions, glass compositions, melt-inclusion trace element compositions etc. either referred to explicitly or presented here to

support this. Alternatives such as xenocrystic origin etc. are completely ignored. How would this change the story?

(pg 8: line 176-180)

These crystals are technically xenocrystic because the cores are not in equilibrium with the melt. There are two core populations with 10 mol% compositional difference, suggesting two magmas as opposed to one magma and xenocrysts. Melt inclusions from this eruption also show a wide range of compositions both lower and higher than the bulk rock composition (Bouvet de Maisonneuve et al., 2012; Ruth et al., 2016).

12. There is no statistical treatment of the data to support the main assertion that timescale duration correlates with depth. There is a loose association on the basis of 22 crystals that each seem to have their eccentricities, yet this dataset is simply not large enough to make for a convincing argument. The more you interrogate figure 2, the more you see exceptions until they outnumber the trends. I think there's probably something here, but at present the very strongly stated conclusions are not supported by equally convincing trends.

Shea et al., (2015) suggests that a minimum of 20 crystals be collected for diffusion analyses. Therefore, in terms of diffusion, we consider the size of our dataset is perhaps limited but still sufficient to the first degree. New figure 5 shows a general trend of shallowing with time. We are aware that our dataset does not show a perfect correlation between depth and timescales. It is in our experience that data from natural systems will show scatter when comparing two variables. This is why we state that it is a broad trend. Moreover, the scatter could be related to 3D cut effects with diffusion, post entrapment modification, among other processes. Notwithstanding, out of our dataset, the four crystals with the longest timescales are derived from deeper depths, and those with shorter timescales come from shallower depths. From this we derive the stated relationship which we believe is robust.

13. The poor structure of the paper makes for difficult reading. For instance; some ideas are re-introduced multiple times, some observations pop up surprisingly in the same sentence as a discussion point, and some phrasing really does smack of serious question-begging. It might be a structural issue only, yet when I think I follow what the authors are saying question-begging is very clear on occasion.

We have significantly revised the text to improve readability. Moreover, since we were not provided specific examples, it is difficult to address the criticism in a specific way. We note that the other reviewers did not find issue with the structure of the paper.

14. Some interpretations in other works are presented as irrefutable fact, and the authors use these many times to hang their arguments on. They simultaneously describe some previous work as 'improper' and this only serves to frame their work as highly model-driven.

We are not sure what the reviewer means by "interpretations in other works are presented as irrefutable fact". The reviewer could be referring to the comparison of views where some view melt inclusions as a generalized suite, whereas other view them as recording eruption specific processes. We have included specific references for both views and discuss them in the introduction (pg 3: lines 54-59) and the discussion section (pg 10: lines 231-244). If the reviewer is not referring to this interpretation, then we ask that the reviewer specify to which they are referring.

15. There appears to be several instances of incorrect terminology, which reads as if the authors are not as familiar with these concepts and methods as I know them to be. I suspect these are simple mistakes, but I do feel strongly that with an author list such as this these mistakes simply should not be here.

We have carefully checked through the manuscript and find no instances where the incorrect terminology has been used. However, if we have missed any incorrect usages, we please ask the reviewer to specify instances to us.

16. The manuscript reads as if it is hurrying along and skipping over the central scientific arguments, which corresponds in most instances to those which are ‘farmed-out’ to references. I think the authors have a story here, but the writing is choking up the narrative. I would recommend a fresh start, because that is the most likely way this paper will have the largest impact.

We have significantly revised the paper to add more detail to avoid “farming out” the already published data. This includes more information about the melt inclusions and how volatile contents were obtained (supplemental table s1, pg 18-19: lines 372-410). We also fail to recognize what the reviewer means about our writing but we have rewritten good parts of the text.

17. I think the main assertion that melt inclusions are thought to record syn-eruptive processes but actually are formed earlier is an unfounded straw-man argument. Final cooling and crystallization, element mobility etc. are certainly recorded during the eruption. Why is this a problem?

This is not the same thing as assessing when and where those inclusions became enclosed, and I haven't seen a paper trying to make this claim that in this manuscript the authors pose as such a problem. Syn-eruptive processes probably convolve or destroy the primary signals that we wish to obtain, but I don't think anyone would say that the melt inclusions always form at shallow levels...

I would add finally that there are no references cited when this problem is described toward the beginning of the MS, and this doesn't leave the reader who is unfamiliar with the concepts with a great deal of confidence in the initial motivation for the study.

(pg 3: lines 54-59, pg 10: lines 231-244)

We disagree with the reviewer concerning this argument. In short, the problem is how do we know the genetic relations between various melt inclusions, and how does this in turn affects our interpretations of degassing patterns. As explained for reviewer 2, we note there are two schools of thought and discuss this in more detail in the text. Although some papers interpret melt inclusions as a suite that record the overall behavior of a volcanic system (e.g. Wade et al., 2006 among others), there are others where melt inclusion compositions are explicitly linked to syn-eruption processes (e.g. Spilliaert et al., 2006; Blundy and Cashman, 2005, among others). Our dataset shows that melt inclusions formed or reequilibrated, at a minimum, over many weeks to years before the eruption occurred. Therefore, these melt inclusions cannot have recorded syn-eruptive processes. We discuss this more in the introduction and the discussion sections, respectively.

IN LINE COMMENTS – Reviewer 3

27: Redundancy here

We have significantly revised the abstract. This has been removed.

29: This opening statement could be ironed out. I've given it a go.

We have significantly revised the abstract. This has been removed.

30: Conjunction used before the ideas conjoined. Please check for further instances.

We have significantly revised the abstract. This has been removed.

34: Is the data volatile?!

We have significantly revised the abstract. This has been removed.

43: Do you mean the timing of such emissions?

Yes, we are referring to the timing of ash emissions. We have significantly revised the abstract. This has been removed.

45: I don't think this logically follows the above as written. I know what you mean, but this statement needs a linking sentence of some sort to be supported.

We have significantly revised the abstract. This has been removed.

56: I think you're better off separating these two thoughts into two sentences.

We have significantly revised the intro section of the paper.

58: I'd prefer to see 'properly' not used. It implies others are not doing their jobs correctly, and I think your points are better couched in a less bullish manner. Interpretations are what they are, and are in most cases based upon the prevailing paradigms. Those of the past are not 'improper'; they're just potentially outdated by your present work! Try "more accurately".

(pg 3: lines 67-68)

We have significantly revised the text and 'properly' has been removed.

63: A Nature Comms audience needs this help I think.

(pg 3: lines 69-70)

We have revised the text and removed reference to the southern volcanic zone and only reference southern Chile.

63-70: I don't think you need to re-introduce all of this. Leave it for the conclusions!

(pg 3-4: lines 69-80)

We have revised the text, but still kept the teaser for the reader.

75: Please separate into two sentences. As this stands it is borderline illogical – how can they be studied previously if you've sampled them afterwards?

(pg 3-4: line 70-72)

This work uses samples from previous studies, where the melt inclusions were the main focus of investigation. The current study focusses on the diffusion chronometry of the host crystals to better interpret the melt inclusions. We have revised the section for clarity.

75: I'd prefer to see a simple sentence inhere describing on what basis this premise is supported. Referencing out what is a central pillar in the paper's development is not going to help really engage the reader

(pg 4: lines 72-76)

Previous work (Bouvet de Maisonnueve et al., 2012 ; Ruth et al., 2016) infer the plumbing system structure based on the abundance of melt inclusions recording specific entrapment depths. We have added content for clarity

77: What does this mean?

(pg 4: lines 75-76)

We infer a well-developed region of magma storage based on the large abundance of melt inclusions that record entrapment pressures at 2-4km beneath the summit.

80: I'd like to see a little critical appraisal of why this should be accepted as fact. Or, even better, couch it in terms of a question you're addressing in this paper. Without this you're describing something that is less innovative, and more incremental, than this paper deserves.

(pg 4: 76-81)

We do not agree that the inclusion of previous diffusion timescale data makes our paper more incremental. The point of our contribution is to show that melt inclusion "ages" can be quite long and therefore using them to infer syn-eruptive processes is in appropriate. To our knowledge, there is no study that assesses the relative "ages" of melt inclusions. The similarity in timescale length between eruptive events gives additional credence to our timescale dataset. No change

81: Specify please?

(pg 5: line 97)

We revised to indicate the number of crystals analysed in this study

82: No, what you mean is element concentration maps.

(pg 4: line 88)

Yes, We have revised the wording to improve clarity

84: Focussing on variety doesn't make sense – maybe try 'capture' and thus gauge the variability?

(pg 4: line 90)

The sentence was changed per the reviewer's suggestion

86: This must be a new paragraph.

(pg 4-5: lines 92-107)

We have revised to include the paragraph break

87: Kernel density estimate (see John Maclennans' work) should be applied to give the reader confidence.

(pg 5: line 100-102, see figure 2)

We produced a kernel density function for the cores, rims, and shoulders regions of the crystals (see Fig 2). Cores are bimodally distributed. The rims show a bimodal distribution as well (Fig. 2c), though with the compositional proximity, it looks like a broad normal distribution.

87: Are these really rims? I suspect you mean the zone immediately in-board of the final quench rim, and this really needs to be stipulated either way.

(pg 4: lines 93-96)

Yes we are referring to the zone directly in-board of the final quench rim. In some crystals we do observe the Fe-rich quench rim of 1-2 um and now note this in the text.

89: Saying e.g. implies that reverse zoning could be something else as well.

(pg 5: line 97)

We have revised this for clarity.

93: I struggle to accept this as written. Firstly it really does seem like accepting the conclusion in order to support the premise, and secondly, there is no quantitative treatment of the data to support the claim. It's fine to just say "zoned crystals of these kinds have been reported in previous eruptions, although a statistical treatment to assess how representative those are, or how they may have changed through..."

NB: This comment was cut off in our pdf file; therefore we do not have access to the complete comment.

(pg 5 : line 105-107)

Bouvet de Maisonneuve et al. (2012 – JPet) provides histograms showing the similarity in olivine compositions between eruptions. We have included these data in Fig. 2 for comparison, showing that the compositions from our dataset are similar and representative.

95: Again, please quantify this.

We have revised the section extensively, and the sentence was removed.

100: Needs a REF I feel.

(pg 6: lines 138-140)

These crystals are chemically zoned, which is classic evidence for magma mixing (e.g. Kahl et al., 2015; Marti et al., 2013; Albert et al., 2015). It is reasonable to infer that the proportion of normal vs. reverse zoned crystals could provide information about the degree of mixing. We are not aware of a reference that tests this. If the reviewer has suggestions, we welcome their input.

101: It'd be nice to see some melt or whole rock data to support this hybridization scenario. If you rationalize this with degrees of mixing, the onus is to demonstrate this is reflected at a scale larger than a crystal.

(pg 6: lines 138-140)

The magma mixing and hybridization are well described in the previous work by Bouvet de Maisonneuve et al. (2012-JVGR, JPet) and Ruth et al. (2016). An in depth discussion of this is beyond the scope of this paper. No change.

103: Why?

(pg 5: lines 118-146)

We have revised the section considerably.

We looked for a relationship between zoning and timescale, but did not observe one. When we looked at other textural features such as melt inclusion morphology and abundance, and host crystal morphology, we noticed the difference in times.

103: Morphology twice?

(pg 5: lines 118-120)

One is host crystal morphology and the other is melt inclusion morphology. We have revised the text for clarity.

107: If you've got 2D data then this is probably best rationalized as a sectioning effect. A tube in one orientation is round...Saying sphere implies you have 3D data...

(pg 5-6: lines 118-128)We can observe the 3D shape of the melt inclusions in transmitted light and many show tube morphologies. However, we have removed the original sentence due to extensive revision to the section.

111: Why do these observations go together?

(pg 5: line 113)

We are trying to point out specifically br5-1 in the old Fig. 1. We have revised this for clarity.

113: This does not make sense as written. An inclusion is only an inclusion when it is closed?!

We agree. We have significantly revised this section and the sentence has been removed.

119: Why is this a backslash here and a hyphen elsewhere?

We have revised the section considerably and the sentence was removed. However, we have ensured that formatting is consistent throughout the document.

121: A little more formal language is probably needed. Returned?

(pg 7: line 166)

We have revised the section and the sentence was removed.

121: This is quite confusing, please re-phrase, or step out.

(pg 7: line 166)

We have revised the sentence for clarity.

122: Plus and minus? Please indicate the error envelop.

(pg 8: lines 174-175)

The uncertainty envelop is provided in Table 2, also in figures 4 and 6. We call out the uncertainty in text.

123: Is this not better stated in the discussion?

We have removed the sentence during revision to the section.

129: Well, in time no, but in equilibrium space maybe?

(pg 8: lines 174-184)

We agree that crystals may be treated as single population in equilibrium space. However, many volcanologists are interested in the timing of magmatic processes before eruptions therefore equilibrium space is not as important. There may be many “equilibrium spaces” that record processes at more or less the same time. However, we did revise the text and the sentence was removed.

149: Mmm, try to let the reader feel this rather than saying it is so...I think it is interesting, and I don't mind it being signposted as such per se, but experience shows this does rankle some readers!

(pg 10: lines 223-224)

We have revised this section and removed this wording.

151: Again, this is up to the reader to decide. Just leave out the word significant and the sentence is better for it.

We have revised the section and the sentence was removed.

155-157: Separate into two sentences. Your data and interpretations neither needs supporting by, or validates, this other example. It's good for comparison, but here it is used as a crutch when it's walking along fine!

(pg 10: line 230-231)

We have separated the two clauses. Indeed we lead the following sentence with the second clause.

162: Still on the basis of 22 crystals... You might want to ease back a little on this, or at least put the observation before the interpretation.

(pg 10-11: lines 245-273)

The discussion section has been revised and the sentence of removed.

170: I really think this needs to be higher, in the more results-oriented paragraphs. I like how the logic goes, but you need to talk about the crystal histories first, and then talk about the magma evolution. Mixing them together (no pun intended) like this does make for as compelling reading as the study deserves.

(pg 6: lines 143-145)

We have moved this to the results section.

172: This is still a leap of logic that needs explaining.

(pg 11: lines 257-259)

The elevated RSAM were from abundant long period events, which are related to fluid movements. Several crystals record diffusion starting a few weeks after the increased RSAM, and we interpret this as being related to or triggered by the seismic events. We have added text to clarify this in the manuscript.

174: I'd really discourage the use of the word 'overall'. It's very weak, and implies an inconsistent and only loosely-held-together logic chain that you are unable or unwilling to guide the reader through... It suggests the reader should believe you at face value...

We have removed the sentence during the revision process.

178: By now the reader could be quite annoyed...

(pg 11: lines 267-270)

We have revised the wording of this sentence.

166 (their pdf lines): All very rushed at the end without solid, logical, justification!

(pg 12: line 274-278)

We added some additional justification to the text. However, given the short length of the paper, we feel a more extensive conclusion section would be overkill.

METHODS

(line numbering on pdf is odd)

177: I don't see why this needs to be stated; 70 degrees is an inherent part of the instrumentation.

(pg 13: lines 296-297)

We have included this for completeness in terms of methodology. No change.

178: Firstly, 7 is the minimum; it can't be both 7 and 9. Secondly, why what the index rate so poor? How were they mounted and polished? In my experience experimenting with polishing, you can get the wrong results with such a poor index rate. When the same crystals are given a good colloidal silica polish (as opposed to 1 or ¼ um diamond for instance) the index rate is excellent (nearing 99%), and the results are considerably different from the several points indexed at the coarser polish. You either must demonstrate why such confidence is assigned to the data by giving qualitative comparison to the crystal morphology, or re-polish and re-analyze. It could be that you performed a 'point and shoot' method of data gathering as opposed to programming a map. I would recommend programming a map; this gives you a much more meaningful dataset and you can assess the index rate with meaningful statistics. If they are maps, I highly suspect the polish isn't good enough.

Most EBSD studies use maps for *deformed* crystals. These crystals were hand picked out of crushed tephra, therefore mapping is not as relevant. We collected EBSD data by selecting

an area on each crystal, with the beam scanning for indexing. For those with few indexed points, the returned orientations were consistent with observed habits in cross section, especially where interfacial angles could be measured.
We have expanding on our methods discussion for clarity.

180: This could be rephrased please.

(pg 13: line 309)

We have kept quantitative to denote it from qualitative assessments of the zoning, for example visually or with image processing software.

181: Does this imply the maps using WDS are less quantitative? Or, where the maps actually EDS?

(pg 13: lines 290-292)

The maps were semi-quantitative collected with WDS and background corrected. The compositional traverses were collected with WDS and are quantitative. We have revised the sentence for clarity.

183: I think you should talk in terms of spot size. Beam diameter is not as meaningful, and is not equivalent to spot size.

(pg 14: lines 310-311)

We provide all traditionally reported information for the beam conditions, including beam diameter. If we were analysing plagioclase or other unstable material, then the spot size may be more relevant. However, olivine is stable under the beam and we think the beam diameter as measured in the faraday cup is the same as in the olivine. No change.

185-186: Not sure this suggestion is much better, but please re-word somehow.

(pg 14: lines 312-313)

We have revised the sentence for clarity.

187: I don't understand how you could have made a decision on step sizes before you had the data...?

(pg 13-14: lines 302-308)

We visually inspected and used the profile tool in ImageJ to assess the zoning for point spacing. We aimed to have the tight spacing over the compositional gradient and wider point spacing on plateaus. We have revised the section for clarity.

190: This isn't always the case in the figures you show.

(pg 14: lines 307-308)

We have revised the sentence to indicate that some profiles may be collected at less than perpendicular angles.

193: I think you should specify which ones, and what difference it made. Were they all from one group? Also briefly describe the advantages and limitations of the MAN method. This can be done in two sentences, and would the give the reader a much better feeling of confidence!

(pg 14: lines 319-322)

We provide this information in the supplemental material table S2. We also include a few sentences justifying the use of this technique.

196: This isn't quite correct. It's not the % difference, but the shape. You might want to justify why you say 'minimum' and include a brief discussion about the influence of growth on shape and calculated timescales.

(pg 14: lines 336-350)

We agree it is the diffusion shape and not the compositional difference between the zones. We have revised the sentence for clarity. We also include a section about growth may affect the diffusion timescale in the main text.

198: Did you model other elements? If no, don't say e.g.

(pg 15: lines 338-339)

We did model other elements, but we revised the sentence for clarity.

198: How good were these matches?

(see new Fig 3, inset, and Table 2)

The discrepancy value returned by the DIPRA software reflects the % difference between the model output and the original traverse. We report the diffusion timescale that reflects the lowest % difference; for Fo this ranges from 0 to 6 %. For trace elements, the discrepancy ranged up to 20%. The discrepancy is also reflected in the uncertainty values. See Table 2 for details.

201-202: I don't think 'account' is the word you're searching for. Correct is probably better.

(pg 15: lines 343-345)

We have changed the wording of the sentence.

203: ??? Diffusion speed is highly sensitive to ambient temperature. Temperature does not change the diffusivity of a material.

(pg 7: line 153-155)

We agree. We have move it to the main text and revised the section for better clarity.

206: Why?

We chose the bulk chemistry because Llaima magmas are remarkably consistent between eruptions. Moreover, the bulk rock is only slightly more mafic than the matrix glass. Modelling with the bulk composition would produce Fo80 crystals whereas the evolved matrix composition would not.

209: Why triple the temperature error compared to the plagioclase thermometry results?

(pg 7: line 161-162, 347-348)

The ± 30 °C is the reported uncertainty for most geothermometers (Putirka, 2008). We have added text to state this explicitly.

212: This is confusing. Please specify clearly in what sense you are using 'initial' and 'boundary'. This is the second instance of this strange phrase and it is not clear from the grammar how you are distinguishing between the two, or if you are combining them, or what 'both' refers to.

(pg 15-16: lines 351-364)

The initial condition is the starting composition and profile for the model.

We have revised the section for clarity.

214: Reference needed here.

(pg 15: line 351-352)

Reverse zoning is classic evidence for magma mixing. We have provided an appropriate citation for this sentence.

214: Reference here and please qualify 'usually' with counter example perhaps.

We have removed this sentence during revision.

220: Please re-phrase.

(pg 16: lines 356-357)

The model needs to know what to do with at the boundaries, thus we either fix the boundaries (closed), or they can be variable. Core boundaries are always fixed, whereas the rim boundaries can change, especially for the complex zoned crystals. No change

223: Infinite?

(pg 16: line 358)

Yes, we have rephrased the sentence.

226: ?? I think you've lost most readers now. This is an important description and the paper would be better received if this was re-written.

(pg 16: line 361)

We have revised the sentence for clarity.

229: This really needs to be stepped out.

(pg 16: lines 363-364)

We have revised the sentence for clarity.

232: This is really shutting the reader out, please include these however briefly.

(pg 16-17: lines 372-410)

We have added a substantial section on the melt inclusions including their preparation and analysis.

235: This crucial section reads like a series of excuses as to why the authors don't want to discuss anything here with any real rigour. Just summarize each paper in a paragraph each; you have space!

(pg 16-17: lines 372-410)

We have added a substantial section on the melt inclusions including their preparation and analysis.

244: This doesn't really tell us anything?

(pg 18: lines 419-420)

The seismic events recorded in the RSAM resulted from abundant long period events and tremor. These are classically linked to fluid movement. This information is provided for context. No change.

Figure 1: The third olivine profile is taken from a corner? What does the profile opposite (on a fairly nice straight crystal edge) suggest?

Is that not a contradiction then?

We collected traverses that we as close to perpendicular to the crystal margin as possible. In some cases, this was more difficult due to the presence of melt inclusions. In those circumstances, it was necessary to choose a less than ideal traverse location.

Figure 2: What is CSD, what is OSD?

Some statistical assessment is needed to convince the reader. As currently presented this is not particularly strong; try randomly selected portions of the data and see how this holds. Maybe just point out that the 100-ish day 'population' are all shallow?

(New figure 5)

CSD and OSD are closed and open system degassing respectively. We have revised the figure to explicitly state this.

Figure 3: I'd prefer to see this as a line on the figure; a gap is never as good as showing exactly where data were not collected!

How can one really be sure that a gap means no data, or no signal? Without reading the figure caption how this figure is presented is not how it first comes across. The dates are hard to read. Just use a field to indicate which year you're in as time goes on...

Consider revising.

But they don't change depth? Why are they complex?

(New figure 6)

Given that this figure shows a time series, a gap shows longer periods of time where data were not collected. Our colleagues at OVDAS assure us that the data gaps reflect downed instruments. There may have been seismic events, but they were not recorded by these instruments. We have included text in the figure to indicate the instruments were down during the data gaps.

We have revised the figure to improve readability. The complex crystals are those with multiple zones. They may not necessarily change depth and that is something we cannot assess at this time because we did not measure melt inclusions within the shoulder-rim zones.

Reviewers' comments:

Reviewer #2 (Remarks to the Author):

The manuscript "Crystal and melt inclusion relative ages expose the depth and time evolution of magma migration" by Ruth et al. is a resubmission of a manuscript that I previously reviewed. This new version is much improved and most of the comments raised by the reviewers are adequately addressed. However, a few major aspects remain questionable.

1. The use of melt inclusions (MIs) and the derived pressure estimates are taken as face value, while there is likely some post entrapment modification that can change the inferred calculated pressure and weaken somewhat the conclusions of the manuscript. While CO₂ is the primary variable affected by pressure, in hydrous arc magmas isobars are also a function of H₂O (see manuscript fig. 5). Thus, any H₂O loss can significantly alter pressure estimates especially at low CO₂; e.g. a change from 3 to 1 wt.% H₂O can lead to a difference of >50 MPa (or >1.5 km). CO₂ diffusion in olivine is relatively slow -potential reequilibration cannot be completely neglected-, while H diffusion in olivine is rapid (e.g., Kohlstedt and Mackwell 1998, Demouchy and Mackwell 2006, Gaetani et al. 2012, Lloyd et al. 2013). Such rapid H diffusion allows for re-equilibration at magmatic temperatures within hours to days depending also on melt inclusion radius, crystal size, and location of the melt inclusion relative to the crystal edge – for an extensive discussion of H re-equilibration in melts and mafic minerals see fig. 14 of Lloyd et al. 2016. Therefore, unless melt inclusions are transported rapidly to the surface they will record at least partially a depth of last equilibration (mostly H loss as melt inclusions ascend to lower P). It remains unclear how common recording of the true depth of formation really is. These effects may (I think quite likely) lead to a decoupling of pressure information recorded in melt inclusions and timescale information derived from olivine elemental zoning.

Let me illustrate how quite probable scenarios are going to affect the inferred model of Ruth et al. The issue lies in the fact that magma mixing derived from olivine zonation provides timescale information but does not provide spatial information within the plumbing system. Thus we know the time between mixing and eruption to the surface (this is obviously well-known to the authors). We do NOT know where the mixing takes place and whether the crystal moved around. That is where the authors turn to the MI record. In the manuscript the authors frequently tie events of magma mixing and interaction to the MI-derived pressures, sometimes that is specifically done and other times it is implied. However, one can imagine a least one scenario that decouples timescale and pressure constraints significantly. E.g. if an olivine with a melt inclusion that has formed in equilibrium at 100 MPa at hydrous conditions (e.g. 3 wt% H₂O and 300 ppm CO₂, see fig.5) mixes with a recharge magma at that depth an olivine rim may form and the diffusion clock starts. The melt inclusion volatile content would be unaffected as it is primarily controlled by pressure. This magma may partially ascend to more shallow levels at a later time after mixing (1 hour, 1 day, 100 days) not necessarily affecting the olivine elemental zoning and diffusion (which will just continue). However, now the melt inclusion will start to equilibrate its water content resulting in a significantly lower recorded pressure. In this case, the mixing timescale is related to an event that is decoupled from the recorded pressure. Another alternative scenario involves an olivine that hosts a melt inclusion where the olivine gets picked up by a new magma and rapidly transported to shallow levels. The olivine zonation-derived mixing timescale may start at this shallow level as the crystal grows delayed and significantly removed from the pickup region of the olivine grain (again note that transient magma ascent is commonly rapid, e.g., Lloyd et al. 2014, and whether olivine crystallization can keep up kinetically is unclear). The melt inclusions again will equilibrate to that new depth. In this case, timescale and pressure estimate are better linked but still do not represent properly the location of the origin of the melt inclusions or where magma interaction has commenced. The latter example may be more in line with what the authors present but it still highlights the difficulties that emerge when combining melt inclusion pressure estimates with olivine zonation timescale information. In summary, building a model for the pre-eruption architecture of the Llama plumbing system may always strongly biased towards lower pressure estimates. Suggesting that there is a general upward progression shortly prior to the eruption is

one potential outcome, but a large portion of the magma may still come from deeper levels. As I said previously what is supported is that olivines are picked up from a range of depths briefly before eruption. The first statement in the discussion "... connect ... the depth of crystal and melt inclusion formation/last depth of equilibration with their residence times ..." is promising more than what really can be done here!

Whether there is water loss in these melt inclusions is hard to assess; there is one MI that could be treated as an outlier (at 3 wt% H₂O) but then it still requires some explanation of why there is a more water-rich magma and where it resides. So, should we believe that the general degassing curve drawn in fig.5 at ~ 2wt% H₂O is meaningful? I hope to have made a point that water content can easily represent a lower pressure re-equilibration level and given the time of months to years that are assessed with the olivine zonation data this is very likely the case (again, Lloyd et al. 2016 summarize that these olivines take days to re-equilibrate).

So the authors should not just focus solely on post-entrapment modification after eruption (e.g. II. 384ff) as they do when I reviewed their response to reviewer comments, but consider re-equilibration at high temperature in the volcanic subsurface. It may not complete change the story, but it needs to be considered. Reviewer 1 also raises this issue (her comment 2) and the authors' response is focused primarily on post-eruption effects, which they say can be neglected. The latter is potentially true, but a response to the reviewers by the authors questions that assumption somewhat, too; see below.

2. Fig 5 and table 2 are still inconsistent. If you look at the P estimate in table 2 and compare it with the isobars and where points lie in fig.5; they do not match! What is wrong here? Which of the two things is correct, the figure isobars or the table? I did not see anything in the supplement that would help to understand the inconsistency here. E.g. the 3wt% H₂O MI is calculated to record a pressure of 117 MPa, however on fig. 5 it plots between the 50 and 100 MPa isobar. Other datapoints are also not matching.

3. The authors responded to my argument that most people consider MIs as a general suite rather than a strictly syn-eruptive record. They pointed to several papers claiming that those say that MIs are formed syn-eruptive. I re-read those quickly and while the discussion in those papers are much more related to syn-eruptive processes, those studies (with one exception) do not say (unless I missed it) that the MIs form on ascent. (Yet, reviewer 3 may be in such a camp, though I had trouble deciphering his/her comment and under which premise s/he operates.) So, of the cited papers the only study that truly speaks of syn-eruptive processes is Blundy and Cashman 2005, whether that is a good citation is unclear given that they deal with plagioclase and much more evolved magmas, so the system is significantly different. My personal sense is that this is still mostly a strawman argument, but I appreciate the effort that the authors have done to address it.

4. There are some odd choices of references. E.g. while Sparks et al. 1977 is clearly a landmark paper it does not really focus on crystal scale information that corroborates a diverse origin. There are plenty of other studies that address much more directly this concept and one extreme example is one of my own studies that suggest mineral assembly from near-Moho levels as well as a subvolcanic magma chamber (Ruprecht and Plank, 2013). I apologize for promoting my own work, but given this being a diffusion work and that paper focusing on diffusion, magma assembly and magma ascent, I would like to draw the attention to it here. I am sure the authors can think of numerous other alternative studies that are better choices than Sparks et al. (1977).

5. The data presentation is much improved and much more information about the samples and analytical work is available. One final aspect that has not been addressed is the fact that the olivines are from multiple samples (2 if I am correct). Is there a systematic difference between those samples? They may come from a similar time slice of the eruption, but they may still represent different regimes the volcano was in. At this point one cannot address that issue.

6. Responses to reviewers:

- i. The choice of just 30 °C variability in T that was raised by reviewer 1 and partially by me are brushed away. Simply looking at olivine-matrix temperatures likely underestimates the temperature variability under which diffusion occurred. Of course, this only extends the error bars in your time estimates, so it is not the most crucial point, but there is a level of certainty implied currently that may not be warranted. And the comment about anhydrous use of the plag thermometer could be addressed in the paper as it is addressed in the comment to reviewer 1. Someone who is not an expert would wonder why one chooses anhydrous conditions. Lastly, in my line-by-line comments (202ff) the authors explain that a much larger temperature range in the samples has been inferred in previous studies (1114 – 1198 °C, potentially plus the 30 degree uncertainties), so I would think a conservative T estimate beyond the 30 degree analytical uncertain should be considered.
- ii. As noted above, Reviewer 1 also is concerned about sub-volcanic modification of the melt inclusions (comment 2), she asks whether melt inclusions in a single crystal consistently give the same volatile content/pressure estimate. The response by the authors regard Br5-1 shows that they are not considering re-equilibration of MIs in the subsurface and in their view that differences need to be related to melt embayments, which is a possibility but not required to explain a variable set of melt inclusions in the same crystal given how fast H diffuses in olivine.
- iii. Reviewer 1 brought up concerns about some of these grains and their profiles. Looking at the four long timescale grains, with the exception of “plate 1”, all three other grains are quite complex and both the location of melt inclusion and choice of diffusion profile are difficult to interpret. Relative to other grains these have not simple geometries. That said, I would probably tone down the importance somewhat in what these “older” olivines tell us. This is just a suggestion.
- iv. In my line-by-line comments (72) the authors provide an extensive answer that raised my concern about the samples. They state that they collected the “largest scoria possible”. Well how large? Are we still talking lapilli? As you are aware the larger scoria can be affected by post-eruptive equilibration (Lloyd et al. 2013) and that does not necessarily need to be correlated with post-entrapment crystallization.
- v. You highlight in your response to Reviewer 3 (point 11) that mineral chemistry shows that there is mixing because of different cores. I just became curious whether mixing expresses itself in the melt inclusions record too. Are high Fo cores with melt inclusions different in terms of P, H₂O, CO₂ than low Fo cores?

7. Line by line comments (in the Tracked changes version!!!):

L.513ff: This final statement if you make it, probably should have some citations of studies that say that too.

L.571ff: “the crystal-rich shallow plumbing system” has not been mentioned in the paper at all and probably requires some citation to assume this model.

L.683ff: I think here the paper to be cited is Ginibre et al. 2002, who did this gray scale calibration on BSE images to determine compositional zoning quantitatively long before these others studies that are cited. Even though this was done on plagioclase not on olivine it still is the paper to cite.

Philipp Ruprecht

REFS:

Blundy, J., and Cashman, K., 2005, Rapid decompression-driven crystallization recorded by melt inclusions from Mount St. Helens volcano: *Geology*, v. 33, p. 793–796, doi: 10.1130/G21668.1.

Demouchy S, Mackwell SJ (2006) Mechanisms of hydrogen incorporation and diffusion in iron-bearing olivine. *Phys Chem Miner* 33: 347–355.

Gaetani, G.A., O’Leary, J.A., Shimizu, N., Bucholz, C.E., and Newville, M., 2012, Rapid reequilibration of H₂O and oxygen fugacity in olivine-hosted melt inclusions: *Geology*, v. 40, p. 915–918, doi: 10.1130/G32992.1.

Ginibre, C., Kronz, A., and Woerner, G., 2002, High-resolution quantitative imaging of plagioclase

composition using accumulated backscattered electron images: new constraints on oscillatory zoning: *CMP*, v. 142, p. 436–448.

Kohlstedt DL, Mackwell SJ (1998) Diffusion of hydrogen and intrinsic point defects in olivine. *Z Phys Chem* 207: 147–162.

Lloyd, A., Plank, T., Ruprecht, P., Hauri, E., and Rose, W., 2013, Volatile Loss from Melt Inclusions in Pyroclasts of Differing Sizes: Contributions to Mineralogy and Petrology, v. 165, p. 129–153, doi: 10.1007/s00410-012-0800-2.

Lloyd, A.S., Ruprecht, P., Hauri, E.H., Rose, W., Gonnermann, H.M., and Plank, T., 2014, NanoSIMS results from olivine-hosted melt embayments: Magma ascent rate during explosive basaltic eruptions: *Journal of Volcanology and Geothermal Research*, v. 283, p. 1–18.

Lloyd, A., Ferriss, E., Ruprecht, P., Hauri, E., Jicha, B., and Plank, T., 2016, An assessment of clinopyroxene as a recorder of magmatic water and ascent: *Journal of Petrology*, v. 57, p. 1865–1886, doi: 10.1093/petrology/egw058.

Ruprecht, P., and Plank, T., 2013, Feeding andesitic eruptions with a high-speed connection from the mantle: *Nature*, v. 500, p. 68–72, doi: 10.1038/nature12342.

Sparks, S.R.J., Sigurdsson, H., and Wilson, L., 1977, Magma mixing: a mechanism for triggering acid explosive eruptions: *Nature*, v. 267, p. 315–318, doi: 10.1038/267315a0.

Reviewer #3 (Remarks to the Author):

The MS is much improved. The authors have successfully brought out much more clearly was always a very nice study.

The level of detailed observation and good, logical explanation of the interpretation is of a much higher standard than that previously. The introduction and discussion much better reflects the quality of this work, and its importance.

The major claims are much better supported.

There is still room for improvement in terms of stylistic issues, and I have tried to help with the attached .doc** (sorry for the formatting; it was saved into word from a pdf). This is not exhaustive, but hopefully it provides some assistance where it is arguably needed most (in the opening few paragraphs to captivate the reader).

Likewise the figures need improving for readability; this shouldn't be arduous. For example, text needs to be made much larger in some instances. I think these are obvious and don't need pointing out specifically.

I look forward to reading the final version in print.

**Editorial Note: In their review of the third version of this manuscript, reviewer 3 added their comments to the manuscript file. These comments, excluding minor textual revisions, have been copied into this Peer Review File.

Line 86: "Is this not methods? It's only one paragraph worth, maybe put this at the end of the introduction, and lead into it with something like "to address these questions we..."

Line 303, '3 points': "Which were averaged using a mis-orientation index of 2 deg or similar?"

Line 339, 'package': "Needs reference (Rasband)"

Reviewer #4 (Remarks to the Author):

Summary - full review attached as Word document***

In this contribution, Ruth et al. present new diffusion timescales from melt inclusion-bearing olivine crystals from the 2008 Llama eruption in Chile. By integrating these timescales with melt inclusion analyses from the same crystals they present a model of magma plumbing system evolution during the years and months prior to the eruption. They report a shoaling of melt inclusion pressures at a time that corresponds with other signals of incipient unrest. They present a case for improving our understanding of pre-eruptive processes by effectively combining different geochemical datasets.

When receiving this manuscript for review, I was informed that I was reviewing re-submitted paper. I therefore carried out the whole review without looking at the enclosed response to reviews document to avoid being biased by the other reviewers' opinions. Once I finished writing my review, I looked at the response to reviews document, and appended a short commentary at the end of this document.

The authors' multi-faceted (combining diffusion chronometry with melt inclusion barometry) approach is laudable, and has the potential to greatly improve the community's understanding of magmatic processes. In this sense, the manuscript is suitable for publication in Nature Communications. However, I have a number of concerns that affect the manuscript's suitability for publication, at least in its current form. I have supplied some general comments as well as line-by-line comments in an attached Word file.

***Full review:

Summary

In this contribution, Ruth et al. present new diffusion timescales from melt inclusion-bearing olivine crystals from the 2008 Llama eruption in Chile. By integrating these timescales with melt inclusion analyses from the same crystals they present a model of magma plumbing system evolution during the years and months prior to the eruption. They report a shoaling of melt inclusion pressures at a time that corresponds with other signals of incipient unrest. They present a case for improving our understanding of pre-eruptive processes by effectively combining different geochemical datasets.

When receiving this manuscript for review, I was informed that I was reviewing re-submitted paper. I therefore carried out the whole review without looking at the enclosed response to reviews document to avoid being biased by the other reviewers' opinions. Once I finished my review, I looked at the authors' response to reviews document, and appended a few resulting thoughts at the end of this document

The authors' multi-faceted (combining diffusion chronometry with melt inclusion barometry) approach is laudable, and has the potential to greatly improve the community's understanding of magmatic processes. In this sense, the manuscript is suitable for publication in Nature Communications. However, I have a number of concerns that affect the manuscript's suitability for publication, at least in its current form. Line-by-line comments follow some more general comments.

General comments

1. The manuscript is not especially well written, especially for a re-submission. Many sentences are too long and confusion. The authors often use complicated words when simple ones would do. Many interpretations are not well explained, leaving the reader confused. In short, this paper is much more difficult to understand than the authors' various previous contributions.
2. The authors very rightly question the assumptions that many workers have previously made about the meaning of open- and closed-system degassing trends. However, they are very willing to make a number of assumptions about the meaning of melt inclusion entrapment pressures. Many recent studies have illustrated that both the H₂O and CO₂ contents of olivine-hosted melt inclusions are more difficult to interpret than previously thought: Portnyagin et al. (2008; EPSL), Bucholz et al. (2013; EPSL), Hartley et al. (2015; EPSL), Mironov et al. (2015; EPSL), Moore et al. (2015; AmMin), Wallace et al. (2015; AmMin), and Maclennan (2017; G3). While the authors address some of the points raised by these papers, they do not engage as much as is probably necessary given what the community now knows. If only melt inclusions were as trustworthy as we initially hoped they would be! I know the complexities of melt inclusions are frustrating and confound our ability to make progress, but I think the community has to move beyond the paradigm that apparent inclusion pressures automatically equate to depths of inclusion formation (or even equilibration).
3. Uncertainties in estimated temperatures, which of course propagate uncertainties in estimated timescales, are inadequately discussed. What does the quoted ± 30 °C actually correspond to? 1σ , 2σ , an absolute uncertainty?
4. The information provided by Ca, Mn and Ni profiles is only discussed in a very superficial way. I wholly agree that timescales calculated from these elements can contribute useful information. However, the timescales do not agree perfectly with the timescales from Mg-Fe diffusion. Although this is not at all a problem for the arguments presented, the discrepancies between timescales calculated from different element should be

discussed in more detail – a truly multi-element approach is still quite an unusual contribution to the literature after all. Essentially, Fig. 4 should be discussed more (see line-by-line comments).

Line-by-line comments

- L46: The abstract is written in unnecessarily complicated language that makes it difficult to understand. I have suggested some edits below.
- L47: Replace “inferences on” with “information about” and “founded on” with “based on”.
- L48: Replace “reputed crystals that host the inclusions” with “inclusion-bearing crystals”.
- L50: Replace “the melt inclusion information” with “information from melt inclusions”.
- L52: Replace “timing” with “timescales”.
- L51-53: Tense agreement. Either “used” and “found” or “use” and “find” should be used.
- L55: Replace “shallowed” with “became shallower”.
- L56: Replace “could be linked to the observed increased seismicity” with “can be linked to increases in seismicity”.
- L199: The reason for mentioning both inclusion formation and equilibration should be explained more here. The use of “and/or” leaves the reader begging for more information.
- L125-126: Did these previous studies look at the same eruption as the authors? It's not clear whether they did or not from the current wording.
- L129-130: Replacing “based on the abundance of melt inclusions from specific depths” with “based on apparent melt inclusion pressures” would be more transparent because inferring storage depths makes a number assumptions about the reliability of melt inclusion records that may or may not have been met.
- L143-145: I think I understand this sentence, but should probably be rewritten to improve clarity. If I understand correctly, you collected compositional data from the host olivine crystals with the aim of understanding how the inclusions relate to the petrogenetic stories recorded by the crystals themselves? I suggest splitting this discussion over more sentences.
- L146: Replace “more or less concentric zoning” with “concentric zoning of variable intensity”
- L216: Is the Fe-rich quench rim found on all crystals?
- L218: You could add “: reverse, normal and complex zoning” to the sentence that introduces the different styles of zoning to make the following sentences easier to follow.
- L221: No need for quotation marks.
- L222-224: Compositions are surely “of” a forsterite content rather than “at” one.
- L231: Why is it important to mention that regions of high P do not correlate with regions of high Fo? This should be explained. Rapid dendritic crystallisation could feasibility

happen in evolved or primitive melts anyhow. Moreover, any initial Fo-P coherence would be swiftly obliterated by diffusion, as the authors are no doubt aware.

- L255: Crystals have already been divided into three groups based on their Fo zoning patterns. You need to make it clear that this textural classification is different and wholly independent of the previously discussed compositional classification. Anyhow, should this classification come before the description of olivine compositions? Textural information seems more fundamental than compositional zoning as it is basic information that can be obtained simply by optical microscopy.
- L266: Replace “that lead to the high-P zoned and inclusion formation” with “that lead to the growth of high-P zones and subsequent melt inclusion formation”. The current sentence does not make grammatical sense.
- L267-268: This section is confusing. Are the now euhedral olivines in the primitive magma or in the mush that has been intruded? Also, the inclusion of a citation makes the source of this information uncertain. Is it new, or is it from the cited paper.
- L276: Replace “the presence of the crystal-rich (mush) region” with “mixing taking place in a crystal-rich mush”.
- L277: Replace “interior/core portions of the” with “cores of their”.
- L278: These melt inclusions do not necessarily the earliest processes recorded in the eruption products. If the crystals initially grew with dendritic habits, melt inclusion entrapment would have potentially (see later) recorded the conditions prevalent during the development of euhedral crystal habits.
- L301: How were the Ca, Mn and Ni profiles modelled? Presumably with DIPRA as well?
- L303: The first sentence should be plural because you are discussing a general case: “Diffusion coefficients, and consequently modelled timescales, are most influenced by temperature”.
- L307: Why anhydrous when you use 2 wt.% H₂O in your MELTS calculations and measure appreciable H₂O in many melt inclusions?
- L311: What does this ± 30 °C uncertainty correspond to? Is it 1σ , 2σ , an absolute uncertainty? This is very important because it affects how the resulting timescales can be interpreted.
- L364: You need to explain observing this diffusional anisotropy argues against a role for growth in generating the observed zonation. I suggest appending the following after “anisotropy”: “which would not be expected if zoning resulted from growth because rates of diffusion are more anisotropic than rates of growth”. This should probably be substantiated with a reference to crystal growth rates. Do B. Welsch or F. Faure have something to say about this?
- L366: Though only Ca (and possibly Ni) may be illustrative in this sense because Mn is usually modelled as diffusing at the same rate as Mg-Fe.
- L370: The number of profiles investigated may not be statistically robust. Thus, although you cannot identify a dominant peak, I’m not sure you can be sure that there is no peak either. Some people consider 20 to be a statistically relevant number of datapoints, but

this only holds if a single population is present: if there are multiple populations, more data are needed.

- L371: The end of this paragraph is drifting into discussion rather than purely presenting results.
- L371: Is the phrase “magma recharge and subsequent mixing” an explanation for the development of reverse zoning?
- L372: Which timescale is this second timescale? Is it derived from the rims, or from the cores? In any case, the dual timescales from the complexly zoned crystals need to be described more carefully. Is one timescale from the complexly zoned crystals grouped with the previously mentioned timescales (i.e., those showing recharge), for example.
- L376: Replace “are a proxy for the” with “represent”.
- L383: CO₂ loss may also occur by irreversible melt inclusion rupture (decrepitation). This has been recognised by the fluid inclusion community for a long time (Wanamaker et al. (1990; JGR) and Campione et al. (2015; JGR)). See Maclennan (2017; G3) for a recent discussion about how this may impact on our ability to understand melt inclusion CO₂ contents.
- L384: Comparatively dry (i.e., low $a_{\text{H}_2\text{O}}$) melt inclusions can also gain H⁺ from wet magmas following mixing. See Hartley et al. (2015; EPSL) and Mironov et al. (2015; EPSL) for discussions about diffusive loss and gain. The ease with which H⁺ can re-equilibrate through olivines ghost crystals (on the order of days at magmatic temperatures) means that interpreting melt inclusion H₂O contents is very difficult when there are no independent indicators of magma H₂O content. Koleszar et al. (2009) also the open system behaviour of H⁺ in natural olivines.
- L410: Or the depth of decrepitation, which is a disequilibrium process driven by inclusion overpressure, and thus unrelated to magma plumbing architecture.
- L414: You present H₂O values before CO₂ values in all the following sentences, so it would improve the flow to swap these values around here.
- L418: 151 MPa may be close to the limit of what olivine crystals can hold (Wanamaker et al., 1990; JGR).
- L427: Hartley et al. (2016; EPSL) used Mg-Fe diffusion profiles to estimate pre- and syn-eruptive CO₂ fluxes from the Laki eruption – something not dissimilar to what the authors present here.
- L456: No need for quotation marks.
- L450-461: Replace “which in turn provides key insights into the explosivity of the eruptions” with “which has been in turn used to interpret the explosivity of eruptions”.
- L462: How many people in the petrology/volcanology community would still argue that this is the case? How about something like: “it has been traditionally assumed: the cited studies largely pre-date the comparatively recent developments in diffusion chronology that have revolutionised our understanding of the rates of magmatic processes.

- L467: This assumes that the H₂O-CO₂ systematics of the melt inclusions are truly indicative of the storage pressures given that combinations of H⁺ gain, H⁺ loss and CO₂ loss could both increase or decrease apparent entrapment pressures.
- L469: I wholly agree here – the inclusions are probably old. Their current volatile contents, however, may not be old.
- L517: This is a very important point, but it is also important to consider whether melt inclusions are robust archives of magmatic volatile contents.
- L519: Insert “of magmatic processes” before “with time”.
- L524-525: Please be more specific about what you mean by “en mass(e) magma replenishment” as it is unclear what you mean. It would also be useful to have some further justification of why you believe that melt percolation would be more likely.
- L530: What are RSAM data? This acronym should be explained in full at this first occurrence.
- L572-575: This sentence is complex and unwieldy. Please break it up and simplify the language to make it clearer.
- L700: As before, the nature of these uncertainties need to be discussed in much more depth.
- L809: See my comment on the main text: H⁺ gain can also affect the H₂O content of melt inclusions on rapid timescales (Hartley et al., 2015; Mironov et al., 2015; both in EPSL).
- L816: CO₂ contents may also be affected by decrepitation.
- Fig1: Modelling of Group B crystals: Can you be sure that you have modelled to the edge of this crystal if it has experienced resorption, which is clearly evidenced by its rounded form.
- Fig1: Modelling of group C crystals: Can these crystals really be modelled with this step-like initial profile given the wavy nature of the data? In other words, there does not seem to be any evidence for a compositional plateau in the middle of the profile.
- Fig2: It would be very useful to present panels b, c and d with common x-axes, so that the different compositions can be readily compared with each other.
- Fig3: Main plot: It is very difficult to see which error bars correspond to which element. Perhaps you could offset the different elements very slightly for each crystal. It would also help if different elements were plotted with different symbols.
- Fig3: Inset plot: Is “discrepancy” some kind of relative summed misfit? The inset plot would benefit greatly from more explanation; I'm not sure what these discrepancy values mean. Are they good values or bad values?
- Fig4: Is it meaningful to present the timescale data without their errors? Also, is it realistic to discuss the presence of absence of a mode with so few data points? If uncertainties in the timescales are considered, a mode may appear. See Fig. 4 in Hartley et al. (2016; EPSL).
- Fig5: See my comments on the main text about the complexities of interpreting melt inclusion data, particularly with respect to H⁺ exchange and CO₂ loss.

Thoughts arising from reading the response to reviews

Rev #1, Pt 1: Many uncertainties are remain inadequately explained. In particular, the meaning of the oft-mentioned ± 30 °C uncertainty. While I accept that the anhydrous plagioclase-liquid thermometer may be sufficiently accurate for the purposes, why draw attention to it? Why not just use a hydrous model? Also, while I buy the argument that diffusion signals dominate over growth signals, the in-text discussion is still inadequate

Rev #1, Pt 2: This is the same as my Pt 2! Although the authors have incorporated most of Rev #1's comments, I fear they still overstate how faithfully melt inclusion may record magma storage pressures.

Rev #2: While the authors have clearly toned down their remarks about open- and closed-system degassing trends, I think they still overstate the syn-eruptive degassing argument that Rev #2 criticises them for. It is possible to simply present their interpretation without having to build up a position to knock down first.

Rev #3: I share Rev #3's concerns about whether 22 crystals can be considered to be representative. 20 crystals may be sufficient to robustly define a single monodisperse population, but, as the authors convincingly demonstrate, the Llama olivines tell a much more complex story!

Note: All line numbers given are those found in the track changes version of the text.

Comments from Reviewer 2 – Philipp Ruprecht

1. The use of melt inclusions (MIs) and the derived pressure estimates are taken as face value, while there is likely some post entrapment modification that can change the inferred calculated pressure and weaken somewhat the conclusions of the manuscript. While CO₂ is the primary variable affected by pressure, in hydrous arc magmas isobars are also a function of H₂O (see manuscript fig. 5). Thus, any H₂O loss can significantly alter pressure estimates especially at low CO₂; e.g. a change from 3 to 1 wt% H₂O can lead to a difference of >50 MPa (or >1.5 km). CO₂ diffusion in olivine is relatively slow -potential reequilibration cannot be completely neglected-, while H diffusion in olivine is rapid (e.g., Kohlstedt and Mackwell 1998, Demouchy and Mackwell 2006, Gaetani et al. 2012, Lloyd et al. 2013). Such rapid H diffusion allows for re-equilibration at magmatic temperatures within hours to days depending also on melt inclusion radius, crystal size, and location of the melt inclusion relative to the crystal edge – for an extensive discussion of H re-equilibration in melts and mafic minerals see fig. 14 of Lloyd et al. 2016. Therefore, unless melt inclusions are transported rapidly to the surface they will record at least partially a depth of last equilibration (mostly H loss as melt inclusions ascend to lower P). It remains unclear how common recording of the true depth of formation really is. These effects may (I think quite likely) lead to a decoupling of pressure information recorded in melt inclusions and timescale information derived from olivine elemental zoning.

Let me illustrate how quite probable scenarios are going to affect the inferred model of Ruth et al. The issue lies in the fact that magma mixing derived from olivine zonation provides timescale information but does not provide spatial information within the plumbing system. Thus we know the time between mixing and eruption to the surface (this is obviously well-known to the authors). We do NOT know where the mixing takes place and whether the crystal moved around.

That is where the authors turn to the MI record. In the manuscript the authors frequently tie events of magma mixing and interaction to the MI-derived pressures, sometimes that is specifically done and other times it is implied.

However, one can imagine a least one scenario that decouples timescale and pressure constraints significantly. E.g. if an olivine with a melt inclusion that has formed in equilibrium at 100 MPa at hydrous conditions (e.g. 3 wt% H₂O and 300 ppm CO₂, see fig.5) mixes with a recharge magma at that depth an olivine rim may form and the diffusion clock starts. The melt inclusion volatile content would be unaffected as it is primarily controlled by pressure. This magma may partially ascend to more shallow levels at a later time after mixing (1 hour, 1 day, 100 days) not necessarily affecting the olivine elemental zoning and diffusion (which will just continue). However, now the melt inclusion will start to equilibrate its water content resulting in a significantly lower recorded pressure. In this case, the mixing timescale is related to an event that is decoupled from the recorded pressure.

We agree that the scenario as described would lead to a time-pressure decoupling. However, volcanic systems are often vertically extensive, and it is reasonable to think that shallower levels would have cooler, more evolved magmas. If the example crystal in the scenario ascended to the shallower levels, would it not also experience growth and develop another diffusion profile? We think it would. In such cases, the additional zoning would reflect the second mixing event. Indeed, the complex zoned crystals with troughs and shoulders show evidence for late stage mixing and diffusion before the eruption. Others have noted similar trough and shoulder textures suggesting that late stage mixing is recorded more frequently than reported (e.g. Moore and Erlank, 1978; Kohn et al., 1989; Bouvet de Maisonneuve et al., 2012; Kahl et al., 2013). In this case, late stage ascent is recorded in the crystals on the order of a few to 10s of days before an eruption.

Another alternative scenario involves an olivine that hosts a melt inclusion where the olivine gets picked up by a new magma and rapidly transported to shallow levels. The olivine zonation-derived mixing timescale may start at this shallow level as the crystal grows delayed and significantly removed from the pickup region of the olivine grain (again note that transient magma ascent is commonly rapid, e.g., Lloyd et al. 2014, and whether olivine crystallization can keep up kinetically is unclear). The melt inclusions again will equilibrate to that new depth. In this case, timescale and pressure estimate are better linked but still do not represent properly the location of the origin of the melt inclusions or where magma interaction has commenced. The latter example may be more in line with what the authors present but it still highlights the difficulties that emerge when combining melt inclusion pressure estimates with olivine zonation timescale information.

In summary, building a model for the pre-eruption architecture of the Llaima plumbing system may always strongly biased towards **lower pressure estimates**. Suggesting that there is a general upward progression shortly prior to the eruption is one potential outcome, but a large portion of the magma **may still come from deeper levels**. As I said previously what is supported is that olivines are picked up from a range of depths briefly before eruption. The first statement in the discussion "... connect ... the depth of crystal and melt inclusion formation/last depth of equilibration with their residence times ..." is promising more than what really can be done here!

Whether there is water loss in these melt inclusions is hard to assess; there is one MI that could be treated as an outlier (at 3 wt% H₂O) but then it still requires some explanation of why there is a more water-rich magma and where it resides. So, should we believe that the general degassing curve drawn in fig.5 at ~ 2wt% H₂O is meaningful? I hope to have made a point that water content can easily represent a lower pressure re-equilibration level and given the time of months to years that are assessed with the olivine zonation data this is very likely the case (again, Lloyd et al. 2016 summarize that these olivines take days to re-equilibrate).

Our timescale data shows that degassing curves such as those shown in Figure 5 (and very commonly elsewhere in the literature) are not likely representative of degassing at one period in time. Our inferences combined with the re-equilibration work on Hauri, Lloyd, and others seems to indicate that degassing curves are idealized models whose inclusion in the literature should perhaps be revisited.

So the authors should not just focus solely on post-entrapment modification after eruption (e.g. ll. 384ff) as they do when I reviewed their response to reviewer comments, but consider re-equilibration at high temperature in the volcanic subsurface. It may not complete change the story, but it needs to be considered. Reviewer 1 also raises this issue (her comment 2) and the authors' response is focused primarily on post-eruption effects, which they say can be neglected. The latter is potentially true, but a response to the reviewers by the authors questions that assumption somewhat, too; see below.

We have been explicit throughout the text and in previous response to reviewers that the pressures obtained from the melt inclusion volatile contents are minimum values. However, to improve clarity we have removed references to melt inclusion formation depth

Finally, we have added text in the form of three different scenarios that discuss the limits of our current model (see lines 682-717).

2. Fig 5 and table 2 are still inconsistent. If you look at the P estimate in table 2 and compare it with the isobars and where points lie in fig.5; they do not match! What is wrong here? Which of the two things is correct, the figure isobars or the table? I did not see anything in the supplement that would help to understand the inconsistency here. E.g. the 3wt% H₂O MI is calculated to record a pressure of 117 MPa, however on fig. 5 it plots between the 50 and 100 MPa isobar. Other datapoints are also not matching.

We have changed the figure text in the caption to explicitly state that the isobars are produced with the average whole rock composition and 2 wt% H₂O. We note that this is a standard way of producing isobars. That melt inclusions actually trap a wide range of compositions highlights that this way of producing isobars may be limiting.

3. The authors responded to my argument that most people consider MIs as a general suite rather than a strictly syn-eruptive record. They pointed to several papers claiming that those say that MIs are formed syn-eruptive. I re-read those quickly and while the discussion in those papers are much more related to syn-eruptive processes, those studies (with one exception) do not say (unless I missed it) that the MIs form on ascent. (Yet, reviewer 3 may be in such a camp, though I had trouble deciphering his/her comment and under which premise s/he operates.) So, of the cited papers the only study that truly speaks of syn-eruptive processes is Blundy and Cashman 2005, whether that is a good citation is unclear given that they deal with plagioclase and much more evolved magmas, so the system is significantly different. My personal sense is that this is still mostly a strawman argument, but I appreciate the effort that the authors have done to address it.

We still posit that melt inclusions may be interpreted as we describe. However, we have changed the language within the text to focus on other topics, namely that crystals from a variety of depths are gathered over a short period of time prior to eruption (Lines: 704-710) and that good melt inclusion interpretation requires a good handle on the host crystal timescales.

4. There are some odd choices of references. E.g. while Sparks et al. 1977 is clearly a landmark paper it does not really focus on crystal scale information that corroborates a diverse origin. There are plenty of other studies that address much more directly this concept and one extreme example is one of my own studies that suggest mineral assembly from near-Moho levels as well as a subvolcanic magma chamber (Ruprecht and Plank, 2013). I apologize for promoting my own work, but given this being a diffusion work and that paper focusing on diffusion, magma assembly and magma ascent, I would like to draw the attention to it here. I am sure the authors can think of numerous other alternative studies that are better choices than Sparks et al. (1977).

LINES:

We have included more up to date references including Davidson et al 2007 and Cashman and Blundy, 2013.

5. The data presentation is much improved and much more information about the samples and analytical work is available. One final aspect that has not been addressed is the fact that the olivines are from multiple samples (2 if I am correct). Is there a systematic difference between those samples? They may come from a similar time slice of the eruption, but they may still represent different regimes the volcano was in. At this point one cannot address that issue.

The samples were collected by two different research groups. Despite this, there is no systematic compositional difference between the two suites of samples, see fig. 12 in Ruth et al. (2016) for a complete compositional comparison of the datasets.

Reviewer #3 (Remarks to the Author) - Anonymous

The MS is much improved. The authors have successfully brought out much more clearly was always a very nice study. The level of detailed observation and good, logical explanation of the interpretation is of a much higher standard than that previously. The introduction and discussion much better reflects the quality of this work, and its importance. The major claims are much better supported. There is still room for improvement in terms of stylistic issues, and I have tried to help with the attached .doc (sorry for the formatting; it was saved into word from a pdf). This is not exhaustive, but hopefully it provides some assistance where it is arguably needed most (in the opening few paragraphs to captivate the reader). Likewise the figures need improving for readability; this shouldn't be arduous. For example, text needs to be made much larger in some instances. I think these are obvious and don't need pointing out specifically. I look forward to reading the final version in print.

In Line Comments:

Main Text

Line 86: Is this not methods? It's only one paragraph worth, maybe put this at the end of the introduction, and lead into it with something like "to address these questions we..."

We have modified the text significantly and moved the methods to the front. The original section has been revised

Line 303: Which were averaged using a mis-orientation index of 2 deg or similar?

(Lines 214-219: Yes, these analyses were averaged for index points with less than 1° deviation.

This has been added to the text.)

Line 339: Needs reference (Rasband)

(LINE 221: We have added the most up to date reference for ImageJ. This is reference 29 in our reference list.)

Reviewer #4 (Remarks to the Author):

Summary - full review attached as Word document

In this contribution, Ruth et al. present new diffusion timescales from melt inclusion-bearing olivine crystals from the 2008 Llama eruption in Chile. By integrating these timescales with melt inclusion analyses from the same crystals they present a model of magma plumbing system evolution during the years and months prior to the eruption. They report a shoaling of melt inclusion pressures at a time that corresponds with other signals of incipient unrest. They present a case for improving our understanding of pre-eruptive processes by effectively combining different geochemical datasets.

When receiving this manuscript for review, I was informed that I was reviewing re-submitted paper. I therefore carried out the whole review without looking at the enclosed response to reviews document to avoid being biased by the other reviewers' opinions. Once I finished writing my review, I looked at the response to reviews document, and appended a short commentary at the end of this document.

The authors' multi-faceted (combining diffusion chronometry with melt inclusion barometry) approach is laudable, and has the potential to greatly improve the community's understanding of magmatic processes. In this sense, the manuscript is suitable for publication in Nature Communications. However, I have a number of concerns that affect the manuscript's suitability for publication, at least in its current form. I have supplied some general comments as well as line-by-line comments in an attached Word file.

General comments

1. The manuscript is not especially well written, especially for a re-submission. Many sentences are too long and confusion. The authors often use complicated words when simple ones would do. Many interpretations are not well explained, leaving the reader confused. In short, this paper is much more difficult to understand than the authors' various previous contributions.

We have carefully reviewed and simplified the text.

2. The authors very rightly question the assumptions that many workers have previously made about the meaning of open- and closed-system degassing trends. However, they are very willing to make a number of assumptions about the meaning of melt inclusion entrapment pressures. Many recent studies have illustrated that both the H₂O and CO₂ contents of olivine-hosted melt inclusions are more difficult to interpret than previously thought: Portnyagin et al. (2008; EPSL), Bucholz et al. (2013; EPSL), Hartley et al. (2015; EPSL), Mironov et al. (2015; EPSL), Moore et al. (2015; AmMin), Wallace et al. (2015; AmMin), and Maclennan (2017; G3). While the authors address some of the points raised by these papers, they do not engage as much as is probably necessary given what the community now knows. If only melt inclusions were as trustworthy as we initially hoped they would be! I know the complexities of melt inclusions are frustrating and confound our ability to make progress, but I think the community has to move beyond the paradigm that apparent inclusion pressures automatically equate to depths of inclusion formation (or even equilibration).

We have added additional text highlighting the limitations of melt inclusions in the introduction section (lines 78-82) as well as the methods (lines 309-324). In the discussion section (lines 628-717), we added more text provided three different scenarios that include melt inclusion modifications. We speculate on how the data would look given each scenario and give our reasons for using one over the other scenarios.

3. Uncertainties in estimated temperatures, which of course propagate uncertainties in estimated timescales, are inadequately discussed. What does the quoted ± 30 °C actually correspond to? 1σ , 2σ , an absolute uncertainty?

Lines 257-265: The ± 30 °C is based on the absolute uncertainty in the thermometer models (after Putirka, 2008). Moreover it matches the 1σ range of values we observe in our natural data. We have modified the text to explicitly state. Temperatures for the plagioclase-melt thermometer and the olivine-melt (e.g. matrix glass) thermometer returned similar values. Despite this, we have removed reference of the plagioclase-melt thermometer to improve the clarity of the section.

4. The information provided by Ca, Mn and Ni profiles is only discussed in a very superficial way. I wholly agree that timescales calculated from these elements can contribute useful information. However, the timescales do not agree perfectly with the timescales from Mg-Fe diffusion. Although this is not at all a problem for the arguments presented, the discrepancies between timescales calculated from different element should be discussed in more detail – a truly multi-element approach is still quite an unusual contribution to the literature after all. Essentially, Fig. 4 should be discussed more (see line-by-line comments).

Lines 274-280: We think the reviewer is referencing Fig. 3. We have revised the figure and added text briefly discussing the timescale differences between the different elements.

Line-by-line comments

ABSTRACT

We have re-written the abstract to 1) reduce the word count per the Nature guidelines, 2) improve clarity and 2) simplify the language of the section.

L46: The abstract is written in unnecessarily complicated language that makes it difficult to understand. I have suggested some edits below.

L47: Replace “inferences on” with “information about” and “founded on” with “based on”.

(Removed from text)

L48: Replace “reputed crystals that host the inclusions” with “inclusion-bearing crystals”.

(Lines 34-35: Revised in text)

L50: Replace “the melt inclusion information” with “information from melt inclusions”.

(Removed from text)

L52: Replace “timing” with “timescales”.

(Removed from text)

L51-53: Tense agreement. Either “used” and “found” or “use” and “find” should be used.

(Tense agreement resolved)

L55: Replace “shallowed” with “became shallower”.

(Line 39: changed per review suggestion)

L56: Replace “could be linked to the observed increased seismicity” with “can be linked to increases in seismicity”.

(Line 40: Reworded text)

MAIN BODY TEXT

L199: The reason for mentioning both inclusion formation and equilibration should be explained more here. The use of “and/or” leaves the reader begging for more information.

(Line 68: Removed reference to formation. Just discuss the last depth of re-equilibration)

L125-126: Did these previous studies look at the same eruption as the authors? It's not clear whether they did or not from the current wording.

(Line 70: They did look at the same eruption. Re-written for clarity.)

METHODS AND SAMPLES

L129-130: Replacing “based on the abundance of melt inclusions from specific depths” with “based on apparent melt inclusion pressures” would be more transparent because inferring storage depths makes a number assumptions about the reliability of melt inclusion records that may or may not have been met.

(Line 166: Reworded per reviewer's suggestion)

L143-145: I think I understand this sentence, but should probably be rewritten to improve clarity. If I understand correctly, you collected compositional data from the host olivine crystals with the aim of

understanding how the inclusions relate to the petrogenetic stories recorded by the crystals themselves?
I suggest splitting this discussion over more sentences.

(Line 178-180: Reworded per reviewer's suggestion)

L146: Replace "more or less concentric zoning" with "concentric zoning of variable intensity"

(Line 146: Reworded)

RESULTS

L216: Is the Fe-rich quench rim found on all crystals?

(Line 351: In 13 of 22 crystals. Included in text)

L218: You could add ": reverse, normal and complex zoning" to the sentence that introduces the different styles of zoning to make the following sentences easier to follow.

(Line 353: Reworded)

L221: No need for quotation marks.

(Line 370: Removed)

L222-224: Compositions are surely "of" a forsterite content rather than "at" one.

(Lines 371: Reworded)

L231: Why is it important to mention that regions of high P do not correlate with regions of high Fo? This should be explained. Rapid dendritic crystallisation could feasibility happen in evolved or primitive melts anyhow. Moreover, any initial Fo-P coherence would be swiftly obliterated by diffusion, as the authors are no doubt aware.

(Line 378-387: We wanted to establish that there was no relationship between rapid growth and specific olivine chemistry. In this case, high Fo and rapid growth due to mixing. Our observations support what the reviewer says that rapid growth could occur in any composition magma. We have added the text indicating that if there was a correlation it would be decoupled with the rapid diffusion of Fo, per the reviewer's suggestion.)

L255: Crystals have already been divided into three groups based on their Fo zoning patterns. You need to make it clear that this textural classification is different and wholly independent of the previously discussed compositional classification. Anyhow, should this classification come before the description of olivine compositions? Textural information seems more fundamental than compositional zoning as it is basic information that can be obtained simply by optical microscopy.

(Lines 339-349: We placed the textural discussion first and then discuss the zoning observations.)

L266: Replace "that lead to the high-P zoned and inclusion formation" with "that lead to the growth of high-P zones and subsequent melt inclusion formation". The current sentence does not make grammatical sense.

(Lines 386: Reworded)

L267-268: This section is confusing. Are the now euhedral olivines in the primitive magma or in the mush that has been intruded? Also, the inclusion of a citation makes the source of this information uncertain. Is it new, or is it from the cited paper.

(Line 387-389: Given that the majority of the crystals are reverse zoned, we are referring to the euhedral olivines in the resident mush. We are provided a process scenario to explain our data. We have revised the text and removed the citation, which was found to be unnecessary.)

L276: Replace "the presence of the crystal-rich (mush) region" with "mixing taking place in a crystal-rich mush".

(Line 471-472: Reworded)

L277: Replace "interior/core portions of the" with "cores of their".

Removed this sentence to improve clarity.

L278: These melt inclusions do not necessarily the earliest processes recorded in the eruption products. If the crystals initially grew with dendritic habits, melt inclusion entrapment would have potentially (see later) recorded the conditions prevalent during the development of euhedral crystal habits.

We agree and have removed this sentence to improve clarity.

L301: How were the Ca, Mn and Ni profiles modelled? Presumably with DIPRA as well?

(Line 253: All elements were modelled with DIPRA. This information is in the methods section.)

L303: The first sentence should be plural because you are discussing a general case: “Diffusion coefficients, and consequently modelled timescales, are most influenced by temperature”.

(The methods section (Lines 253-268) was revised and this sentence was removed.)

L307: Why anhydrous when you use 2 wt.% H₂O in your MELTS calculations and measure appreciable H₂O in many melt inclusions?

(Line 257: Since we added the olivine-matrix melt, we have removed the discussion of the plagioclase thermometer. We believe this will reduce the confusion in this section.)

L311: What does this ± 30 °C uncertainty correspond to? Is it 1σ , 2σ , an absolute uncertainty? This is very important because it affects how the resulting timescales can be interpreted.

(Line 264-265: First the ± 30 °C is the error on the geothermometers as discussed in Putirka, 2008. Second, it is the 1 sigma standard deviation of our dataset. Reworded.)

L364: You need to explain observing this diffusional anisotropy argues against a role for growth in generating the observed zonation. I suggest appending the following after “anisotropy”: “which would not be expected if zoning resulted from growth because rates of diffusion are more anisotropic than rates of growth”. This should probably be substantiated with a reference to crystal growth rates. Do B. Welsch or F. Faure have something to say about this?

(Line 276-277: Reworded per the reviewer's suggestion. Note we did not find any explicit content in the Faure or Welsch papers about anisotropy. However, Faure and Schiano (2003 and 2005) discuss growth morphology differences with undercooling. With higher undercooling, there is more rapid growth along the c-axis to produce the dendritic features observed in the Welsch papers.)

L366: Though only Ca (and possibly Ni) may be illustrative in this sense because Mn is usually modelled as diffusing at the same rate as Mg-Fe.

(Line 278: We agree. However, previous work (e.g. Kahl et al. 2013) provide Mn as well. We provide it for comparison.)

L370: The number of profiles investigated may not be statistically robust. Thus, although you cannot identify a dominant peak, I’m not sure you can be sure that there is no peak either. Some people consider 20 to be a statistically relevant number of datapoints, but this only holds if a single population is present: if there are multiple populations, more data are needed.

(Line 487-488: We explicitly acknowledge that ours is a small dataset. However, the general distributions observed in our dataset overlap with those presented in Hartley et al. 2016 (their fig 4a), Kahl et al. 2015 (their fig. 15), and our unpublished data on OPX diffusion from Mayon volcano. In all three examples, natural data from similar types of samples are presented. We have added text to indicate the similarity with previous work. Recent work by Couperthwaite and Morgan on Iceland data presented at IAVCEI 2017 shows that 30 crystals is adequate to identify a trend in the diffusion dataset.)

L371: The end of this paragraph is drifting into discussion rather than purely presenting results.

(Line 590: We have moved the last sentences of this paragraph to the discussion section.)

L371: Is the phrase “magma recharge and subsequent mixing” an explanation for the development of reverse zoning?

(LINE 485-486: Yes. Magma recharge and mixing is commonly invoked as a mechanism that results in reverse zoning.)

L372: Which timescale is this second timescale? Is it derived from the rims, or from the cores? In any case, the dual timescales from the complexly zoned crystals need to be described more carefully. Is one timescale from the complexly zoned crystals grouped with the previously mentioned timescales (i.e., those showing recharge), for example.

(Line 489-490: The overall timescales are derived from diffusion of the interior and rim zones. The second timescales are from the rim zone only. Each value is given in table 2. We have revised the section for clarity.)

L376: Replace “are a proxy for the” with “represent”.

(Line 589: Reword per the reviewer’s suggestion.)

L383: CO₂ loss may also occur by irreversible melt inclusion rupture (decrepitation). This has been recognised by the fluid inclusion community for a long time (Wanamaker et al. (1990; JGR) and Campione et al. (2015; JGR)). See Maclennan (2017; G3) for a recent discussion about how this may impact on our ability to understand melt inclusion CO₂ contents.

(Line 81, 319: We were unaware of decrepitation as a process that modifies CO₂. Thanks for the reference. However, the work in Maclennan 2017 really focuses on dry magmas from MOR, ocean islands, and continental settings. Llama magmas are on average 2 wt% water. Additionally, the lower pressure limit for decrepitation is 220 MPa, and it is less likely for shallow melt inclusions to rupture. Our samples all have melt inclusion pressures less than 150 MPa. This could be related to loss of H₂O, which has been discussed at length. Our timescale suggest we cannot rule it out entirely. Our CO₂ data show a reasonable decrease with pressure. This is consistent with the persistent quiescent degassing. Reworded text and added reference in the intro section.)

L384: Comparatively dry (i.e., low *a*H₂O) melt inclusions can also gain H⁺ from wet magmas following mixing. See Hartley et al. (2015; EPSL) and Mironov et al. (2015; EPSL) for discussions about diffusive loss and gain. The ease with which H⁺ can re-equilibrate through olivines ghost crystals (on the order of days at magmatic temperatures) means that interpreting melt inclusion H₂O contents is very difficult when there are no independent indicators of magma H₂O content. Koleszar et al. (2009) also the open system behaviour of H⁺ in natural olivines.

(Line 80: From Hartley et al. 2015 a key indicator of potentially re-hydrated melt inclusions is anomalously high H₂O/Ce values. For Iceland, these are 150-280. Based on data from Naumov et al, 2017 arc magmas have H₂O/Ce values on the order of 900. H₂O/Ce values based on data from Ruth et al. are no more than 190 for melt inclusions from the same eruption. Based on this, we do not think re-hydration is a process at play in this instance. We do added the reference to acknowledge that this process can modify melt inclusion compositions.)

L410: Or the depth of decrepitation, which is a disequilibrium process driven by inclusion overpressure, and thus unrelated to magma plumbing architecture.

(Line 597: If decrepitation occurs, the remaining CO₂ still represents some saturation pressure within the plumbing system. The melt inclusion then records this last environment before eruption. No change.)

L414: You present H₂O values before CO₂ values in all the following sentences, so it would improve the flow to swap these values around here.

(Line 600: Reworded per the reviewer's suggestion.)

L418: 151 MPa may be close to the limit of what olivine crystals can hold (Wanamaker et al., 1990; JGR).

Noted.

L427: Hartley et al. (2016; EPSL) used Mg-Fe diffusion profiles to estimate pre- and syneruptive CO₂ fluxes from the Laki eruption – something not dissimilar to what the authors present here.

(Line 488: We acknowledge their work and added the reference.)

L456: No need for quotation marks.

(Line 680: Removed per the reviewer's suggestion.)

L450-461: Replace “which in turn provides key insights into the explosivity of the eruptions” with “which has been in turn used to interpret the explosivity of eruptions”.

(Line 684-685: Reworded per the reviewer's suggestion.)

L462: How many people in the petrology/volcanology community would still argue that this is the case? How about something like: "it has been traditionally assumed: the cited studies largely pre-date the comparatively recent developments in diffusion chronology that have revolutionised our understanding of the rates of magmatic processes.

(Line 732-736: Reworded per the reviewer's suggestion. Also added text about the reliability of melt inclusions in the introduction and provide 3 scenarios that discuss how timescales and modified melt inclusions might be interpreted.)

L467: This assumes that the H₂O-CO₂ systematics of the melt inclusions are truly indicative of the storage pressures given that combinations of H⁺ gain, H⁺ loss and CO₂ loss could both increase or decrease apparent entrapment pressures.

(Line 739: We do not see evidence for H⁺ gain. Therefore, with H⁺ and CO₂ loss, the most likely modification is a decrease in apparent pressure, shifting the samples to shallower levels in the crust. This does not change our timescale-pressure relationship. We do not see short timescale from high depths.)

L469: I wholly agree here – the inclusions are probably old. Their current volatile contents, however, may not be old.

We agree. But the last volatile contents, whatever age, do provide some evidence of the last location of the melt inclusions.

L517: This is a very important point, but it is also important to consider whether melt inclusions are robust archives of magmatic volatile contents.

We agree. Now assuming that a suite of melt inclusions are thought to be reliable after careful inspection, we think it important to consider the zoning behavior as well.

L519: Insert “of magmatic processes” before “with time”.

(Line 774: Reworded per the reviewer’s suggestion.)

L524-525: Please be more specific about what you mean by “en mass(e) magma replenishment” as it is unclear what you mean. It would also be useful to have some further justification of why you believe that melt percolation would be more likely.

(Line 779: By en mass, we mean large, discrete batch injections. If batch injection were occurring we might see deformation to accommodate such a volume. This was not observed (Delgado et al., 2017). We have added this to the text.)

L530: What are RSAM data? This acronym should be explained in full at this first occurrence.

(Line 792: Reworded for clarity. Since we moved the methods section forward, the RSAM definition is provided there.)

L572-575: This sentence is complex and unwieldy. Please break it up and simplify the language to make it clearer.

(Line: 810-811: We have split the sentence into two to separate the ideas. We have also simplified the language.)

L700: As before, the nature of these uncertainties need to be discussed in much more depth.

(Line 819-820: Added text about how we set the uncertainty based on the full uncertainty in the thermometers and the natural variation in our temperature dataset.)

L809: See my comment on the main text: H⁺ gain can also affect the H₂O content of melt inclusions on rapid timescales (Hartley et al., 2015; Mironov et al., 2015; both in EPSL).

See answer from L384. We do not think that re-hydration is an important process for these samples, but have added the reference to acknowledge that the process can modify melt inclusions in some circumstances.

L816: CO₂ contents may also be affected by decrepitation.

(Line 319: We moved the methods forward and added text and the reference.)

FIGURES

Fig1: Modelling of Group B crystals: Can you be sure that you have modelled to the edge of this crystal if it has experienced resorption, which is clearly evidenced by its rounded form.

This is true, and does add uncertainty. However, we can see some remnant straight edges in bk4-1. Also, the rim composition is consistent with that observed in other crystals that have not experienced dissolution. Therefore, we consider the timescale reasonable within the dataset.

Fig1: Modelling of group C crystals: Can these crystals really be modelled with this step-like initial profile given the wavy nature of the data? In other words, there does not seem to be any evidence for a compositional plateau in the middle of the profile.

The other option is a homogenous crystal composition. Modelling such a composition would not result in a wavy profile, therefore the step function is the most suitable option.

Fig2: It would be very useful to present panels b, c and d with common x-axes, so that the different compositions can be readily compared with each other.

Noted and fixed.

Fig3: Main plot: It is very difficult to see which error bars correspond to which element. Perhaps you could offset the different elements very slightly for each crystal. It would also help if different elements were plotted with different symbols.

We have revised figure 3 into a 1:1 graph showing each element explicitly.

Fig3: Inset plot: Is “discrepancy” some kind of relative summed misfit? The inset plot would benefit greatly from more explanation; I’m not sure what these discrepancy values mean. Are they good values or bad values?

The discrepancy is a goodness of fit value (see Girona and Costa, 2014) which compares the model data with the natural data. The smaller the number (in %), the better the fit. For Fo, a fit was considered good if the discrepancy value was less than 10%. For the minor elements, we expect more noise and allow up to 20% difference in fit. We acknowledge this is an arbitrary setting; although it is consistent with the statistical use of R² values of 0.8 being considered a good fit for linear data.

We have removed this figure, but the data are presented in table 2.

Fig4: Is it meaningful to present the timescale data without their errors? Also, is it realistic to discuss the presence of absence of a mode with so few data points? If uncertainties in the timescales are considered, a mode may appear. See Fig. 4 in Hartley et al. (2016; EPSL).

We show this to provide a distribution of the dataset. We do provide the uncertainty in Figure 3, Figure 6 and in table 2.

Fig5: See my comments on the main text about the complexities of interpreting melt inclusion data, particularly with respect to H⁺ exchange and CO₂ loss.

We indicate that all values for CO₂ are minimum values. We have also added text about decrepitation and H⁺ gain.

Thoughts arising from reading the response to reviews

Rev #1, Pt 1: Many uncertainties are remain inadequately explained. In particular, the meaning of the oft-mentioned ± 30 °C uncertainty. While I accept that the anhydrous plagioclase-liquid thermometer may be sufficiently accurate for the purposes, why draw attention to it? Why not just use a hydrous model? Also, while I buy the argument that diffusion signals dominate over growth signals, the in-text discussion is still inadequate

With the addition of the olivine-melt thermometer, we have removed discussion of the plagioclase thermometer. We believe this reduces confusion.

Rev #1, Pt 2: This is the same as my Pt 2! Although the authors have incorporated most of Rev #1’s comments, I fear they still overstate how faithfully melt inclusion may record magma storage pressures.

We have clearly indicated that the storage pressures in our melt inclusions are minimum values. We have revised the text to indicate they are likely re-equilibration pressure, which is expected given the long residence times shown by timescales.

Rev #2: While the authors have clearly toned down their remarks about open- and closed system degassing trends, I think they still overstate the syn-eruptive degassing argument that Rev #2 criticises them for. It is possible to simply present their interpretation without having to build up a position to knock down first.

We have revised the framing of our research question. Although we do still have a section in the discussion concerning the interpretation of degassing curves (Lines 749-771), it is no longer the main focus of the paper.

Rev #3: I share Rev #3’s concerns about whether 22 crystals can be considered to be representative. 20 crystals may be sufficient to robustly define a single monodisperse population, but, as the authors convincingly demonstrate, the Llaima olivines tell a much more complex story!

See response to L370. Recent work by Leeds diffusion group found that 30 crystals is sufficient to define a diffusion dataset. Ideally, we would collect more data but that is not possible at this time.

Reviewers' comments:

Reviewer #2 (Remarks to the Author):

I have now reviewed this manuscript three times and while I still find this study very valuable, it has not changed as much in this last version as I would have expected despite a productive conversation at a recent meeting. I remain somewhat confused of how the seemingly shallowing of the magma storage depth recorded in the melt inclusions and the shorter diffusion times is interpreted (Fig. 6). The seismicity and ash emissions are preceding the magma ascent into the edifice by weeks to months (for most of the olivines studied). Would not you expect that to be more coeval? Also, the main assembly (bringing all the different crystals together from different depths) and finally eruption of the different olivines seems not to be correlated with any significant increase in seismicity. In ll. 424 the authors state that the "final stage of magma ascent was near coincident in time with increased seismic activity and ash emissions". This is confusing given that clearly the FINAL ascent is the eruption and assembly of all the different crystals in January 2008, during that time seismicity does not appear to be significantly increased, in fact, the 6 months before the eruption look rather like a background signal. Either I am not getting it, or the authors are not clearly stating it that all these crystals resided at different depths and then were collected by a magma coming from much greater depth (such like we have suggested for Irazu volcano from the lower crust or even the mantle in Ruprecht and Plank, 2013) to bring them up to the surface. This last assembly is essential when trying to bring together crystals of different "ages" from different depths (MI record) during the eruption.

While I now follow better Fig. 5, I still do not understand why the degassing path of these magmas has to be at 2 wt% H₂O. There is sufficient data at higher H₂O that the degassing history is more complex. I understand that it is a specific point of the authors that melt inclusion records have flaws. But, those issues have been recognized and are usually discussed and therefore they should be acknowledged here, too, ie. the closed system degassing path should not come down from 2 wt% H₂O, but higher. Of course, we do not know whether these arc magmas at depth are uniform in max H₂O, so potentially there could be a range in H₂O values entering the middle and upper crust, but given this study that residence is long one can easily reconcile the range in H₂O as a result of magma stalling and re-equilibration. However, that would mean that the degassing paths should be at much higher H₂O than shown.

The authors' argument (as a response to my review) that more complex mixing histories should impart themselves as more complex zoning patterns in olivine is not convincing to me, given that they argue at the same time that Llaima magmas are remarkably uniform in composition (e.g. ll. 253ff) mixing may not express itself in the olivine Fo content and the argument that more evolved melts have to occupy the shallowest part of the edifice is an assumption that in the case of Llaima seems less warranted, as magma compositions in general are rather primitive and quite restricted in composition (quick EarthChem look up suggested 52-56 wt% SiO₂).

Best regards,
Philipp Ruprecht

Reviewer #4 (Remarks to the Author):

This is the second time that I have been this manuscript from Ruth et al. on the timescales and depth of magma assembly under Llaima volcano.

They authors have done a very thorough job of responding to my comments and to the comments of the other reviewers. The text is now clearer and the authors' arguments are much easier to follow as a result. The figures are also much easier to interpret.

While I still have some reservations about the authors' general approach, they are transparent about its limitations, which makes it possible for readers to come to their own conclusions.

The manuscript is now ready for publication and will make a useful contribution to the literature on magmatic timescales. I did notice a few formatting glitches in the compiled pdf file, so watch out for these in the proof.

Reviewer #2 (Remarks to the Author):

This first review statement is involved. We have split it to address each question individually.

I have now reviewed this manuscript three times and while I still find this study very valuable, it has not changed as much in this last version as I would have expected despite a productive conversation at a recent meeting.

I remain somewhat confused of how the seemingly shallowing of the magma storage depth recorded in the melt inclusions and the shorter diffusion times is interpreted (Fig. 6). The seismicity and ash emissions are preceding the magma ascent into the edifice by weeks to months (for most of the olivines studied). Would not you expect that to be more coeval?

The reviewer brings up an important point about how geophysical and geochemical data are/should be compared and interpreted. We do not think that seismic events and magma mixing are going to be perfectly coeval. Individual seismic events are short term in nature, on the order of seconds to minutes (Rosenau et al., 2017 – Solid Earth). In contrast whereas magma mixing takes longer. Numerical models suggest, for crystal free systems, mixing can occur on the order of hours to a few days (e.g. Morgavi et al., 2017 – Advances in Volcanology). However, many volcanic systems are crystal-rich, which would necessarily increasing the time between the initial injection (and assumed seismic event) and complete mixing. We acknowledge this delay between the seismic activity and our timescale dataset and provide several possible explanations (Lines 641-643).

Also, the main assembly (bringing all the different crystals together from different depths) and finally eruption of the different olivines seems not to be correlated with any significant increase in seismicity.

This is not unreasonable given that eruptions at open vent systems like Llaima are not always preceded by obviously apparent geophysical unrest (e.g. Chaussard et al., 2013; Delgado et al., 2017), especially if they are not well instrumented. No change.

In ll. 424 the authors state that the “final stage of magma ascent was near coincident in time with increased seismic activity and ash emissions”. This is confusing given that clearly the FINAL ascent is the eruption and assembly of all the different crystals in January 2008, during that time seismicity does not appear to be significantly increased,

in fact, the 6 months before the eruption look rather like a background signal. Either I am not getting it, or the authors are not clearly stating it that all these crystals resided at different depths and then were collected by a magma coming from much greater depth (such like we have suggested for Irazu volcano from the lower crust or even the mantle in Ruprecht and Plank, 2013) to bring them up to the surface. This last assembly is essential when trying to bring together crystals of different “ages” from different depths (MI record) during the eruption.

We do not think the diffusion is recording the final ascent, i.e. the eruption related ascent. Based on ours and other datasets, the resolution for Fo diffusion is ≥ 1 day; thus, Fo diffusion likely cannot resolve eruption-related ascent timing. Therefore, we propose that Fo diffusion is best used to assess for long term processes and that other chronometers seem to be better suited to record the final, eruption-related ascent (e.g. H₂O diffusion, Li diffusion in olivine). We explicitly acknowledge this in the text (Lines 541-565 pg. 16). We have revised the text to indicate that Fo diffusion is recording the final stages of magma accumulation prior to the eruption (Lines 649-651 pg. 19).

While I now follow better Fig. 5, I still do not understand why the degassing path of these magmas has to be at 2 wt% H₂O. There is sufficient data at higher H₂O that the degassing history is more complex. I understand that it is a specific point of the authors that melt inclusion records have flaws. But, those issues have been recognized and are usually discussed and therefore they should be acknowledged here, too, i.e. the closed system degassing path should not come down from 2 wt% H₂O, but higher. Of course, we do not know whether these arc magmas at depth are uniform in max H₂O, so potentially there could be a range in H₂O values entering the middle and upper crust, but given this study that residence is long one can easily reconcile the range in H₂O as a result of magma stalling and re-equilibration. However, that would mean that the degassing paths should be at much higher H₂O than shown.

The degassing paths were based on the larger melt inclusion dataset reported in Bouvet de Maisonneuve et al., 2012 and Ruth et al., 2016 (see Ruth et al., 2016, figure 12). Although we do observe a few melt inclusions with H₂O contents up to 4 wt%, the majority exhibit H₂O contents ≤ 2 wt%. Moreover, the melt inclusions with the highest CO₂ contents had ~ 2 wt% H₂O. These observed values served as the guide for our degassing models.

However, we acknowledge in the text that post-entrapment H₂O loss is likely. We have also added an open-system degassing curve at 3wt% H₂O (based on observed H₂O contents in the larger melt inclusion dataset) for reference in figure 5.

The authors' argument (as a response to my review) that more complex mixing histories should impart themselves as more complex zoning patterns in olivine is not convincing to me, given that they argue at the same time that Llaima magmas are remarkably uniform in composition (e.g. ll.253ff).

Mixing may not express itself in the olivine Fo content...

We agree that not every olivine may record mixing events. We also acknowledge that %Fo diffusion profiles that do record mixing may be re-equilibrated and thus erased (e.g. Bouvet de Maisonneuve et al., 2016). However, we show complex diffusion profiles, with troughs and hooks. Such profiles have been observed elsewhere (e.g. Moore and Erlank, 1978; Kohn et al., 1989; Bouvet de Maisonneuve et al., 2012; Kahl et al., 2013; K. Lynn, personal communication). At least for the Llaima samples, the complex zoning exhibited a wide range of %Fo, and concomitant changes in Ni abundance. These observations rule out other processes and indicate the complex zoning is related to multiple magma injection and mixing events.

...and the argument that more evolved melts have to occupy the shallowest part of the edifice is an assumption that in the case of Llaima seems less warranted, as magma compositions in general are rather primitive and quite restricted in composition (quick EarthChem look up suggested 52-56 wt% SiO₂).

Although the bulk chemistry for recent and historical eruptions has been consistent at 52 wt% SiO₂, more evolved magmas have been erupted from Llaima.

- 1. The full range of magma chemistry at Llaima has varied between 51-67 wt% SiO₂ (GeoRoc database, accessed 26 Jan 2018); note that the higher SiO₂ magmas were produced during large plinian eruptions at 9 ka and ~13 ka.**
- 2. Melt inclusions from the recent eruptions exhibit compositions from 48-58 wt% SiO₂ (Bouvet de Maisonneuve et al., 2012; Wehrmann et al., 2014; Ruth et al., 2016). Several of the evolved inclusions have minimum apparent pressures <100 MPa**

(see included figure). This suggests that there could be regions of the shallow plumbing system that are able to evolve beyond the bulk chemistry.

The fact that Llaima erupts compositionally consistent lavas is an interesting problem that could be related to two processes.

- 1. We do not observe distinctly different injecting magma xenoliths that have been observed in other volcanic systems. The main line of evidence for magma mixing is the chemical zoning in the crystals. So, one hypothesis is that the injecting magma is not that much different than the resident magma.**
- 2. In addition, mixing might be very efficient, effectively erasing the bulk evidence for the mafic, injecting magma.**

Despite this interesting question, it is beyond the scope of this paper to address this in great detail.

Reviewer #4 (Remarks to the Author):

This is the second time that I have been this manuscript from Ruth et al. on the timescales and depth of magma assembly under Llaima volcano.

The authors have done a very thorough job of responding to my comments and to the comments of the other reviewers. The text is now clearer and the authors' arguments are much easier to follow as a result. The figures are also much easier to interpret.

While I still have some reservations about the authors' general approach, they are transparent about its limitations, which makes it possible for readers to come to their own conclusions.

The manuscript is now ready for publication and will make a useful contribution to the literature on magmatic timescales. I did notice a few formatting glitches in the compiled pdf file, so watch out for these in the proof.

Thank you for letting us know about the formatting issues. We have carefully reviewed the text to correct any formatting issues.

REVIEWERS' COMMENTS:

Reviewer #2 (Remarks to the Author):

I have reviewed this manuscript four times. I congratulate the authors on a very interesting manuscript. I have some minor reservations about the manuscript and the responses regarding my latest comments, but I understand that there are differences in opinion that do not have to be resolved in this process. Overall, I agree with the approach and find this manuscript highly informative and valuable to our community.

One final comment that I hope the authors review briefly prior to publication is the discrepancy in their manuscript and the corresponding figures regarding bubbles within the olivine-hosted melt inclusions (ll. 95 in the pdf version). The text states that there are three crystals that have vapor bubbles. A quick look at the images in the supplementary figure 1 suggests that many more have CO₂ bubbles (more than 10 crystals?). So, the CO₂ estimates are clearly a minimum estimate as no CO₂ correction was performed (which is difficult after the fact, especially if the Aster et al approach does not seem to work here). A brief statement about this in the published version may help the reader to assess pressure-time relationships and how pressures have been determined.

Lastly just a quick note for future submissions: The line references in the response to the reviewers did not line up with the pdf I had available for review. Presumably there was some discrepancy due to "tracked changes" in a word document I did not have access to.

Again, congrats on a very interesting study.
Philipp Ruprecht

Reviewer #4 (Remarks to the Author):

This is the third time I have seen this manuscript and I have been asked to comment on the authors' response to the last round of comments to reviewer #2.

The line references provided by the authors do not match the sections of text i believe the authors are referring to; the manuscript text extends only to line 466. Nevertheless, I think I can find all the sections of the text referenced in the review and am generally convinced by the authors' responses to reviewer #2.

Most of the points raised in the last round of review relate to the precision of the language used, which has improved substantially over the various iterations I have seen. I find the authors arguments about progressive magma assembly sensible and while entirely reasonable in content, I'm not sure that the concerns of reviewer #2 can definitively resolved within the scope of either this study or a much, much larger one. Moreover, it's important to remember that petrology and geophysics provide fundamentally different records of magmatic processes, so an expectation of perfect correspondence seems erroneous to me.

Only two points remain:

I worry that the text will be illegible in some figures.

The Pers Comm from Kendra Lynn should be replaced with Dohmen et al. (2010; GCA). I'm aware of Kendra's exciting work and I think it shows much promise indeed, but a proper citation is always better than a Pers Comm.

Reviewer #2 (Remarks to the Author):

I have reviewed this manuscript four times. I congratulate the authors on a very interesting manuscript. I have some minor reservations about the manuscript and the responses regarding my latest comments, but I understand that there are differences in opinion that do not have to be resolved in this process. Overall, I agree with the approach and find this manuscript highly informative and valuable to our community.

One final comment that I hope the authors review briefly prior to publication is the discrepancy in their manuscript and the corresponding figures regarding bubbles within the olivine-hosted melt inclusions (ll. 95 in the pdf version). The text states that there are three crystals that have vapor bubbles. A quick look at the images in the supplementary figure 1 suggests that many more have CO₂ bubbles (more than 10 crystals?). So, the CO₂ estimates are clearly a minimum estimate as no CO₂ correction was performed (which is difficult after the fact, especially if the Aster et al approach does not seem to work here). A brief statement about this in the published version may help the reader to assess pressure-time relationships and how pressures have been determined.

Thanks for noting this. We have removed the specific reference. We do explicitly state in several places that the volatile contents are minima values and therefore reflect re-equilibration depths (Results section- Assessing post entrapment modification; starting at line 267-275). Moreover, we have changed the wording throughout the paper to reflect this difference.

Lastly just a quick note for future submissions: The line references in the response to the reviewers did not line up with the pdf I had available for review. Presumably there was some discrepancy due to “tracked changes” in a word document I did not have access to.

Thanks for letting us know. The line-number discrepancy has been an issue that I’m still working on fixing.

Again, congrats on a very interesting study.

Philipp Ruprecht

Reviewer #4 (Remarks to the Author):

This is the third time I have seen this manuscript and I have been asked to comment on the authors' response to the last round of comments to reviewer #2.

The line references provided by the authors do not match the sections of text i believe the authors are referring to; the manuscript text extends only to line 466. Nevertheless, I think I can find all the sections of the text referenced in the review and am generally convinced by the authors' responses to reviewer #2.

Thanks for letting us know. The line-number discrepancy has been an issue that I'm still working on fixing.

Most of the points raised in the last round of review relate to the precision of the language used, which has improved substantially over the various iterations I have seen. I find the authors arguments about progressive magma assembly sensible and while entirely reasonable in content, I'm not sure that the concerns of reviewer #2 can definitively resolved within the scope of either this study or a much, much larger one. Moreover, it's important to remember that petrology and geophysics provide fundamentally different records of magmatic processes, so an expectation of perfect correspondence seems erroneous to me.

Only two points remain:

I worry that the text will be illegible in some figures.

I have adapted the image to Nature editorial standards and selected fonts sizes that should be readable at those sizes.

The Pers Comm from Kendra Lynn should be replaced with Dohmen et al. (2010; GCA). I'm aware of Kendra's exciting work and I think it shows much promise indeed, but a proper citation is always better than a Pers Comm.

Noted and changed.